# The Nedd4L ubiquitin ligase is activated by FCHO2-generated membrane curvature

Yasuhisa Sakamoto [1], Akiyoshi Uezu[1], Koji Kikuchi [1], Jangmi Kang[2], Eiko Fujii[2], Toshiro Moroishi [3], Shiro Suetsugu [4] & Hiroyuki Nakanishi [1,2]✉

## Abstract

The C2-WW-HECT domain ubiquitin ligase Nedd4L regulates membrane sorting during endocytosis through the ubiquitination of cargo molecules such as the epithelial sodium channel (ENaC). Nedd4L is catalytically autoinhibited by an intramolecular inter-action between its C2 and HECT domains, but the protein's acti-vation mechanism is poorly understood. Here, we show that Nedd4L activation is linked to membrane shape by FCHO2, a Bin-Amphiphysin-Rsv (BAR) domain protein that regulates endocy-tosis. FCHO2 was required for the Nedd4L-mediated ubiquitination and endocytosis of ENaC, with Nedd4L co-localizing with FCHO2 at clathrin-coated pits. In cells, Nedd4L was specifically recruited to, and activated by, the FCHO2 BAR domain. Furthermore, we reconstituted FCHO2-induced recruitment and activation of Nedd4L in vitro. Both the recruitment and activation were mediated by membrane curvature rather than protein–protein interactions. The Nedd4L C2 domain recognized a specific degree of membrane curvature that was generated by the FCHO2 BAR domain, with this curvature directly activating Nedd4L by relieving its autoinhibition. Thus, we show for the first time a specific function (i.e., recruit-ment and activation of an enzyme regulating cargo sorting) of membrane curvature by a BAR domain protein.

**Keywords** Nedd4L; FCHO2; Clathrin; Endocytosis; Membrane Curvature
**Subject Categories** Membranes & Trafficking; Signal Transduction

## Introduction

Posttranslational modification of proteins by covalent attachment of ubiquitin (Ub) is catalyzed by three enzymes: a Ub-activating enzyme (E1), a Ub-conjugating enzyme (E2), and a Ub-protein ligase (E3) that determines substrate specificity (Hershko and Ciechanover, 1998). E3 Ub ligases are classified into two categories based on their Ub transfer mechanisms: RING finger/U-box E3 and HECT-type E3. The Nedd4 family belongs to HECT-type E3 Ub ligases and consists of nine members, including Nedd4/Nedd4-1, Nedd4L/Nedd4-2, Itch, Smurf1, Smurf2, WWP1, WWP2, NEDL1, and NEDL2 (Ingham et al, 2004; Rotin and Kumar, 2009). They are characterized by a common modular organization, with an N-terminal C2 domain, two to four WW domains, and a C-terminal catalytic HECT domain. While the C2 domain was originally identified in protein kinase C (PKC) as a $Ca^{2+}$-dependent phosphatidylserine (PS)-binding domain (Nishizuka, 1992), it shows significant diversity in the binding partners, including intracellular proteins and other phospho-lipids, such as phosphatidylinositol (4,5)-bisphosphate [$PI(4,5)P_2$] (Lemmon, 2008). The WW domains recognize proline-rich motifs, such as PPxY (PY motif, where x is any residue), of substrates or adaptor proteins. The HECT domain catalyzes the isopeptide bond formation between the Ub C terminus and the substrate lysine residues. At least some of the Nedd4 family members, including Nedd4L, are catalytically autoinhibited by an intramolecular interac-tion between the C2 and HECT domains (Wang et al, 2010; Wiesner et al, 2007; Zhu et al, 2017). However, the activation mechanism of the Nedd4 family is poorly understood.

The Nedd4 family is implicated in a wide variety of cellular processes by regulating membrane trafficking and degradation of components of various signaling pathways (Ingham et al, 2004; Rotin and Kumar, 2009). Among a diversity of substrates, the best-characterized is the epithelial sodium channel (ENaC). This channel is composed of three subunits, α-, β-, and γENaC, each of which possesses two transmembrane segments and intracellular N- and C-termini with a large extracellular loop (Butterworth, 2010; Rotin and Staub, 2011; Rossier, 2014). Each subunit has a PY motif at the cytosolic C-terminus. Nedd4L Ub ligase specifically binds to the PY motif and mediates the ubiquitination of lysine residues on the N-terminus of at least α- and γENaC at the plasma membrane. Ub serves as an internalization signal for recognition by adaptor proteins, such as epsin and Eps15, for clathrin-mediated endocytosis (CME) (Traub, 2009). Consequently, ENaC is con-stitutively internalized through CME. Heterozygous mutations of the PY motif of β- or γENaC lead to impaired ubiquitination and endocytosis, resulting in persistence of the channels at the cell surface and increased $Na^+$ absorption in Liddle syndrome, an autosomal dominant form of severe hypertension.

[1]Department of Molecular Pharmacology, Faculty of Life Sciences, Kumamoto University, 1–1–1 Honjyo, Kumamoto 860-8556, Japan. [2]Faculty of Clinical Nutrition and Dietetics, Konan Women's University, 6–2–23 Morikita-machi, Kobe 658-0001, Japan. [3]Department of Molecular and Medical Pharmacology, Faculty of Life Sciences, Kumamoto University, 1–1–1 Honjyo, Kumamoto 860-8556, Japan. [4]Division of Biological Science, Graduate School of Science and Technology, Nara Institute of Science and Technology, 8916-5 Takayama-cho, Ikoma 630-0192, Japan. ✉E-mail: hnakanis@gpo.kumamoto-u.ac.jp

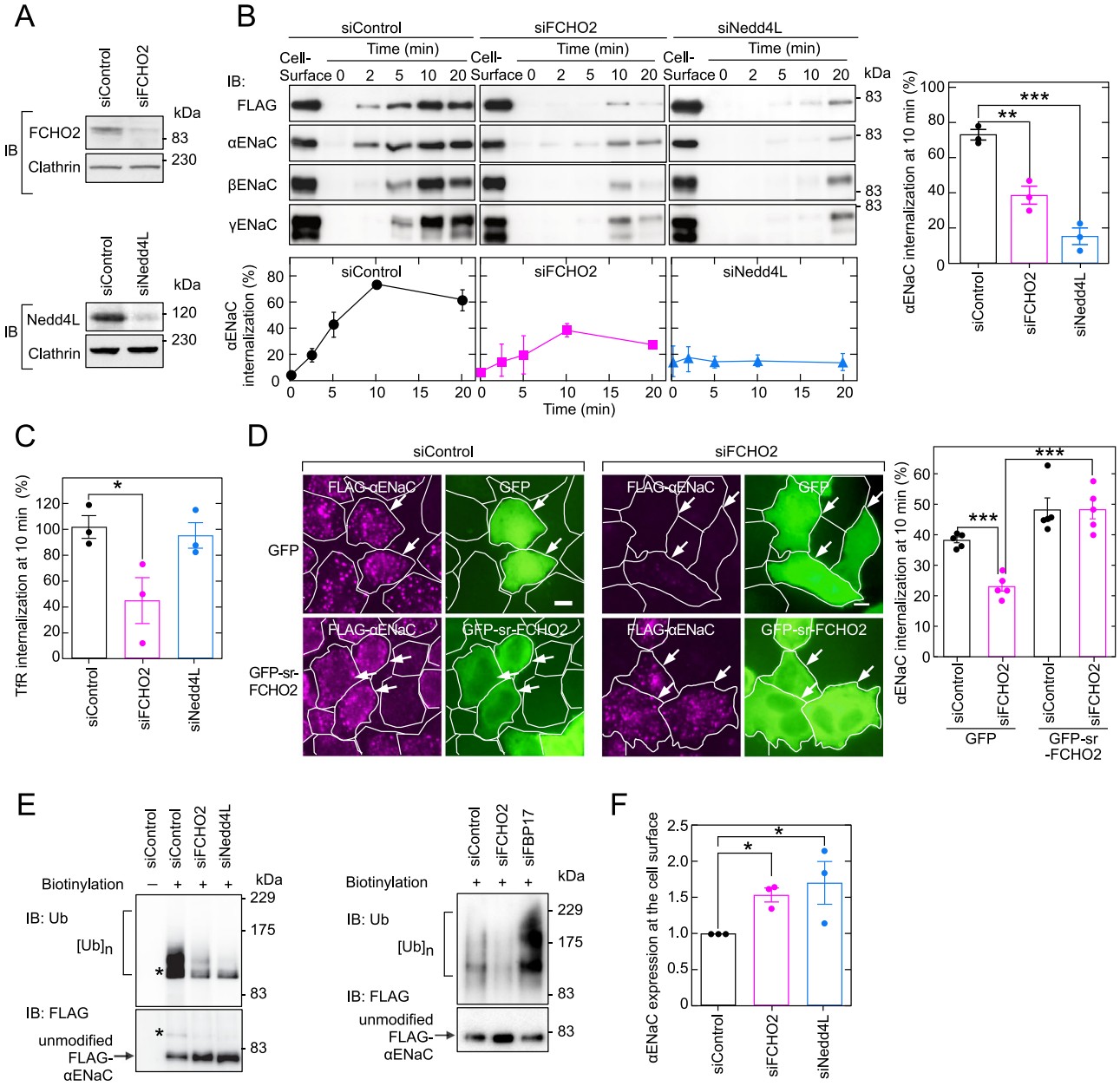

The process of CME begins at the cytoplasmic surface of the plasma membrane with the assembly of clathrin with adaptor proteins and transmembrane cargo molecules (Doherty and McMahon, 2009; Traub, 2009). In parallel with this assembly, membrane curvature is generated to form hemispherical clathrin-coated pits (CCPs). This process leads to deep membrane invaginations. Subsequently, the neck of invaginated pits is constricted and severed to separate the newly formed clathrin-coated vesicles (CCVs) from the plasma membrane. These membrane curvatures are considered to be generated, sensed, and/or maintained by Bin-Amphiphysin-Rsv (BAR) domain superfamily proteins. BAR domains form a crescent-shaped dimer characterized by a specific degree of intrinsic curvature. They bind to negatively charged phospholipids, such as PS, and force membranes to bend according to their intrinsic curvatures,

thereby inducing the formation of membrane tubules with specific diameters (curvatures). For example, the diameters of membrane tubules generated by the amphiphysin BAR domain, FCHO2 BAR domain, and FBP17 BAR domain are 20–50 nm (sharper curvature) (Peter et al, 2004; Takei et al, 1999; Yin et al, 2009), 50–80 nm (intermediate curvature) (Henne et al, 2007), and ~150 nm (Takano et al, 2008) (shallower curvature), respectively.

The FCHO2 BAR domain has been proposed to trigger the initial invagination for CCP formation (Henne et al, 2010). FCHO2 binds to the flat surface of the plasma membrane at sites where CCPs will form. FCHO2 then generates the initial membrane curvature and recruits the clathrin machinery for CCP formation. Here, we report the role of FCHO2 in Nedd4L-mediated ENaC endocytosis and show that FCHO2 BAR domain-generated

**Figure 1.  FCHO2 is required for Nedd4L-mediated ubiquitination and endocytosis of ENaC.**

(A) Knockdown of FCHO2 or Nedd4L by siRNA. αβγENaC-HeLa cells treated with each siRNA were analyzed by immunoblotting (IB). (B) Inhibition of ENaC endocytosis by knockdown of FCHO2 or Nedd4L. After cells were transfected with each siRNA, cell-surface ENaC was labeled by incubating at 4 °C with anti-FLAG antibody. Endocytosis was started by incubating cells at 37 °C for the indicated periods of time and stopped by chilling cells on ice. Antibody bound to the cell surface but not internalized was removed by acid stripping. After cells were solubilized, the antibody-labeled internalized ENaC subunits were precipitated and analyzed by IB (upper panel) and quantified (bottom and right panels). Internalization was expressed as a percentage of the initial amount of cell-surface αENaC determined by lysing cells without incubation at 37 °C or acid stripping. Data are shown as the mean ± SEM of three independent experiments. **$P < 0.01$; ***$P < 0.001$ (Student's $t$ test). (C) Inhibition of transferrin receptor (TfR) endocytosis by FCHO2 knockdown. After cells were transfected with each siRNA, cell-surface proteins were biotinylated at 4 °C. Cells were then incubated at 37 °C for 10 min. Biotin was removed from the cell surface, and internalized biotinylated proteins were precipitated with avidin beads. Samples were subjected to IB with anti-TfR antibody and quantified. Internalization was expressed as a proportion of the initial amount of TfR on the cell surface. Data are shown as the mean ± SEM of three independent experiments. *$P < 0.05$ (Student's $t$ test). (D) Restoration of αENaC endocytosis by FCHO2 re-expression in knockdown cells. After cells were transfected with the indicated siRNA and GFP construct, cell-surface αENaC was labeled with anti-FLAG antibody at 4 °C. Cells were then incubated at 37 °C for 10 min. After acid stripping, cells were subjected to immunofluorescence microscopy (left panels) and quantified (right panel). The antibody-labeled, internalized αENaC was visualized with a fluorescence-conjugated secondary antibody. The border of each cell is delineated by a solid line. Arrows indicate cells expressing GFP-tagged proteins. Scale bars, 10 μm. The proportion of cells with internalized αENaC among cells expressing control GFP or GFP-siRNA-resistant (sr) -FCHO2 is shown. Data are shown as the mean ± SEM of five independent images (31–71 cells expressing GFP-tagged proteins per image). ***$P < 0.001$ (two-way analysis of variance with post hoc test). (E, F) Effects of FCHO2, Nedd4L, and FBP17 knockdown on ENaC ubiquitination and expression at the cell surface. After cells were transfected with each siRNA, cell-surface αENaC was biotinylated and sequentially precipitated with anti-FLAG antibody and then avidin beads. Samples were subjected to IB with anti-FLAG and anti-Ub antibodies (E) and quantified (F). (E) ENaC ubiquitination, Asterisks indicate the same band of ubiquitinated FLAG-αENaC. (F) ENaC expression at the cell surface. Data are shown as the mean ± SEM of three independent experiments. *$P < 0.05$ (Student's $t$ test). Source data are available online for this figure.

## Results

### FCHO2 is required for Nedd4L-mediated ubiquitination and endocytosis of ENaC

To study the role of FCHO2 in ENaC endocytosis, we established a HeLa cell line stably expressing all three subunits, α-, β-, and γENaC (αβγENaC-HeLa cells) (Fig. EV1A–C). A FLAG tag was introduced into the extracellular region of αENaC in a position previously shown not to affect channel activity (Firsov et al, 1996). Upon FCHO2 knockdown by small interfering RNA (siRNA) methods, ENaC internalization was reduced, as was upon Nedd4L knockdown (Zhou et al, 2007) (Fig. 1A,B). Transferrin receptor (TfR) internalization was reduced upon FCHO2 knockdown but not upon Nedd4L knockdown (Fig. 1C). The phenotype of FCHO2 knockdown in ENaC internalization was rescued by an siRNA-resistant (sr) form of FCHO2 (Fig. 1D; Appendix Fig. S1). Furthermore, αENaC ubiquitination was reduced upon FCHO2 or Nedd4L knockdown but not upon FBP17 knockdown (Figs. 1E and EV2). FCHO2 or Nedd4L knockdown increased αENaC expression at the cell surface (Fig. 1E,F). These results suggest that FCHO2 is required for Nedd4L-mediated ubiquitination and endocytosis of ENaC.

### Recruitment and activation of Nedd4L by FCHO2 in cells

Immunofluorescence microscopy revealed that in αβγENaC-HeLa cells, cell-surface αENaC co-localized with FCHO2 at clathrin-coated structures (Fig. 2A). However, its co-localization with Nedd4L was not detected (Appendix Fig. S2A), because exogenous expression of Nedd4L-green fluorescent protein (GFP) stimulated ENaC internalization, resulting in the disappearance of αENaC from the cell surface. Cell-surface αENaC was observed in cells lacking Nedd4L-GFP expression (non-transfected cells). When a catalytically inactive Nedd4L mutant, C922A, was expressed, it co-localized with cell-surface αENaC and endogenous FCHO2 (Fig. 2B). The mutant also formed FCHO2-negative spots. In wild-type HeLa cells, Nedd4L accumulated at clathrin-coated structures where FCHO2 localized (Fig. 2C). Additionally, Nedd4L formed clathrin-negative spots, suggesting that they may be involved in clathrin-independent endocytosis.

Using total internal reflection fluorescence (TIRF) microscopy, we investigated whether Nedd4L is recruited to CCPs in wild-type HeLa cells. Similar to FCHO2 described previously (Henne et al, 2010; Lehmann et al, 2019; Zaccai et al, 2022), Nedd4L showed dynamic co-localization with clathrin during pit formation (Fig. 3). When both fluorescent protein-tagged FCHO2 and Nedd4L were expressed, the accumulation of Nedd4L at FCHO2-positive spots was not clearly observed (Appendix Fig. S2B). The reason behind this remains unknown, but it has been shown that clathrin coat formation, measured in time-lapse images of cells expressing fluorescent proteins, is affected by several factors, such as overexpression level, choice of the proteins, and choice of the fluorescent tag (Aguet et al, 2013). These factors may contribute to the failure to detect the accumulation of Nedd4L at FCHO2-positive spots.

While Nedd4L shows a punctate staining pattern at the cell surface (see Figs. 2C and 3), it exhibited diffuse distribution throughout the cytoplasm (Fig. 4A,B). In COS7 cells, co-expression of the FCHO2 BAR domain [amino acid (aa) 1–302] recruited Nedd4L to membrane tubules (Fig. 4A). The Nedd4L C2 domain (Myc-Nedd4L C2, aa 1–160) was also concentrated to membrane tubules, whereas C2 domain-deleted Nedd4L (Myc-Nedd4L ΔC2) was not. In contrast to FCHO2, the FBP17 BAR domain did not recruit Nedd4L to membrane tubules. Additionally, the BAR domain protein amphiphysin1 did not recruit Nedd4L to membrane tubules. Thus, Nedd4L is selectively recruited through the C2 domain to membrane tubules generated by the FCHO2 BAR domain. Similar results were obtained in HEK293 cells (Fig. 4B). In these cells, membrane tubules generated by amphiphysin1 were hardly detected. It showed a punctate staining pattern and did not co-localized with Nedd4L.

membrane curvature induces the recruitment and activation of Nedd4L.

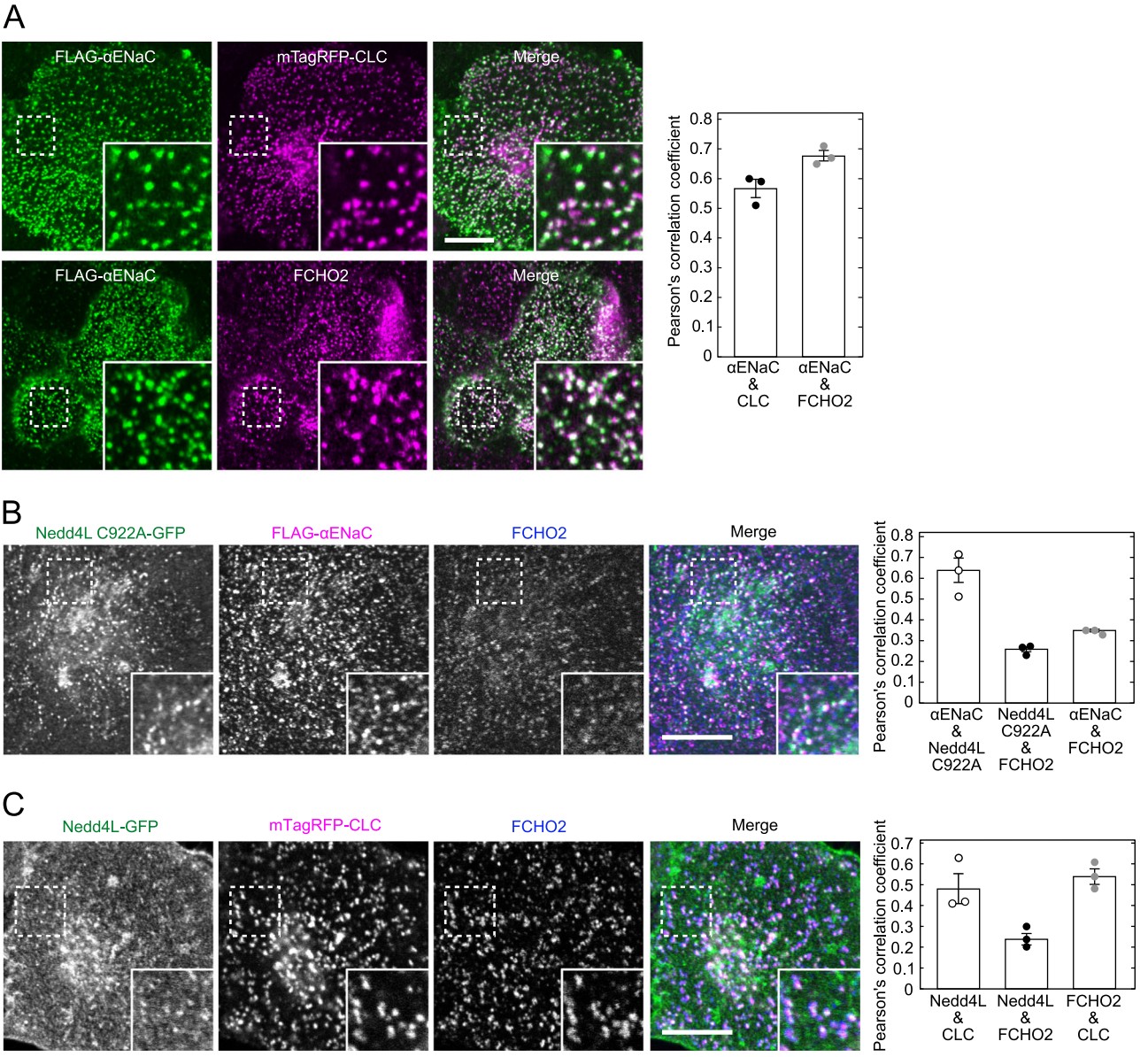

**Figure 2. Localization of cell-surface ENaC, FCHO2, and Nedd4L at clathrin-coated structures.**

(A, B) Localization of cell-surface ENaC. mTagRFP-clathrin light chain (CLC) (A) or Nedd4L C922A (catalytic inactive mutant)-GFP (B) was expressed in αβγENaC-HeLa cells. Cell-surface αENaC was labeled with anti-FLAG antibody at 4 °C. Cells were then fixed and stained with anti-FCHO2 antibody. Samples were subjected to immunofluorescence microscopy (left panel). Insets represent enlarged images of the dashed boxes. Scale bar, 10 μm. Co-localization was analyzed (right panel). The mean values of Pearson's correlation coefficient from three cells and the SEM are shown. (A) Co-localization of cell-surface ENaC with FCHO2 at clathrin-coated structures. (B) Co-localization of cell-surface ENaC with Nedd4L C922A mutant and FCHO2. (C) Co-localization of Nedd4L with FCHO2 at clathrin-coated structures. Nedd4L-GFP and mTagRFP-CLC were co-expressed in wild-type HeLa cells. Cells were then fixed and stained with anti-FCHO2 antibody (left panel). Insets represent enlarged images of the dashed boxes. Scale bar, 10 μm. Co-localization was analyzed (right panel). The mean values of Pearson's correlation coefficient from three cells and the SEM are shown. Source data are available online for this figure.

We also investigated whether the FCHO2 BAR domain not only recruits but also activates Nedd4L in cells. For this purpose, we performed an in vivo ubiquitination assay using HEK293 cells, which formed membrane tubules upon FCHO2 expression more efficiently than did COS7 cells: 91 ± 2% of HEK293 cells vs. 32 ± 2% of COS7 cells [the mean ± standard error of the mean (SEM) of four independent experiments]. As the catalytic activity of E3 Ub ligases is typically reflected in their autoubiquitination (Wiesner et al, 2007), we determined activity by measuring the ubiquitination level of Nedd4L itself. Expression of the FCHO2 BAR domain in cells remarkably enhanced Nedd4L autoubiquitination, whereas FBP17 or amphiphysin1 did not (Fig. 5A). Tubulation-deficient FCHO2 mutants (K146E + R152E and L136E) (Henne et al, 2010) did not enhance Nedd4L activity (Figs. 5A–C and EV3). A catalytically inactive Nedd4L mutant, C922A, which was recruited to FCHO2-generated membrane tubules, showed little

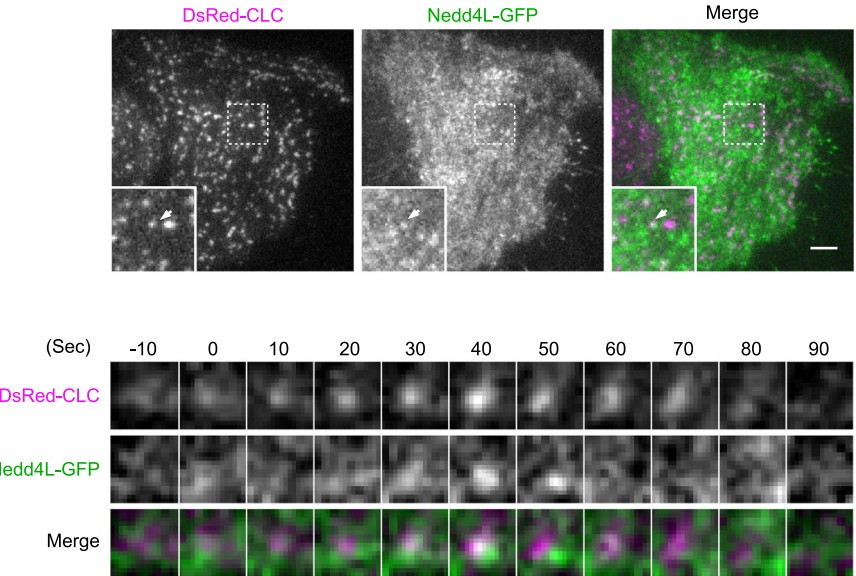

**Figure 3.  Recruitment of Nedd4L to CCPs.**

DsRed-CLC were co-expressed with Nedd4L-GFP in wild-type HeLa cells. Images were acquired at 1 s intervals through TIRF microscopy. Upper panel, TIRF images of HeLa cells. Insets represent enlarged images of the dashed boxes. Bottom panel, selected snapshots (every 10 s) from a time series were obtained from a single CCP indicated by the arrowhead. The time point at which Nedd4L started to accumulate was set 0. Scale bar, 2 μm. Source data are available online for this figure.

enhancement of ubiquitination upon FCHO2 expression (Fig. 5D,E), indicating that Nedd4L ubiquitination is self-induced. Using αENaC as a substrate, we further confirmed that Nedd4L is activated upon FCHO2 expression in cells (Fig. 5F).

## Preference of Nedd4L for FCHO2-generated membrane curvature

The aforementioned results prompted us to investigate whether FCHO2 and Nedd4L interact through membrane tubules rather than through stereotyped protein–protein interactions. Membrane tubules generated by BAR domains in cells have specific diameters (curvatures) (Henne et al, 2007; Peter et al, 2004; Takano et al, 2008; Takei et al, 1999; Yin et al, 2009). Nedd4L may preferentially bind to the specific degree of membrane curvature generated by FCHO2. To investigate this possibility, we first examined the biochemical properties of Nedd4L. Using a co-sedimentation assay with liposomes containing various percentages of PS, we found that the C2 domain of Nedd4L is a $Ca^{2+}$-dependent PS-binding domain (Fig. 6A,B). Submicromolar $Ca^{2+}$ concentrations, which are comparable to intracellular levels in unstimulated cells, were sufficient for the Nedd4L C2 domain to bind to PS-containing liposomes. The C2 domain alone and full-length Nedd4L showed different $Ca^{2+}$ requirements. The higher $Ca^{2+}$ requirement of full-length Nedd4L may be due to the intramolecular interaction between the C2 and HECT domains. Increasing PS percentages in synthetic liposomes enhanced the liposome binding of the Nedd4L C2 domain, and PS requirements varied depending on $Ca^{2+}$ concentration.

We next examined whether Nedd4L preferentially binds to a specific degree of membrane curvature. For this purpose, liposomes of different sizes (curvatures) were prepared by stepwise extrusion through filter membranes with different pore sizes (0.8, 0.4, 0.2, 0.1, 0.05, and 0.03 μm) and sonication. Sonication or extrusion through a 0.03 μm pore-size filter was used to prepare the smallest liposomes. Transmission electron microscopy revealed that the actual liposome size was similar to the pore size for 0.2, 0.1, 0.05, and 0.03 μm pore-size liposomes, whereas it was significantly smaller for 0.8 and 0.4 μm pore-size liposomes (Fig. 7A). Nedd4L showed the strongest binding to 0.05 μm pore-size liposomes (Fig. 7B).

Synaptotagmin I (SytI) tandem C2 domains can both generate and sense membrane curvature in vitro by penetrating their hydrophobic residues into the lipid bilayer (Martens et al, 2007). As a sensor, the binding affinity is considered to be highest for membranes that have a degree of curvature consistent with that generated by SytI itself. We investigated whether, in addition to sensing membrane curvature, the Nedd4L C2 domain can also generate membrane curvature in vitro and, if so, what diameter of membrane tubules is generated. As expected, transmission electron microscopy revealed that the C2 domain deformed liposomes into extensive tubulation (Fig. 7C). It should be noted that Nedd4L does not generate membrane tubules in cells (see Fig. 4A,B). The tubule diameter generated by Nedd4L in vitro was mainly 40–80 nm, which is consistent not only with the diameter of 0.05 μm pore-size liposomes, to which Nedd4L preferentially binds, but also with the diameter of membrane tubules generated by FCHO2 (Fig. 7D). These results suggest that Nedd4L shows the highest binding affinity for the degree of membrane curvature generated by FCHO2. It is therefore likely that FCHO2-induced recruitment and activation of Nedd4L are mediated by the degree of membrane curvature.

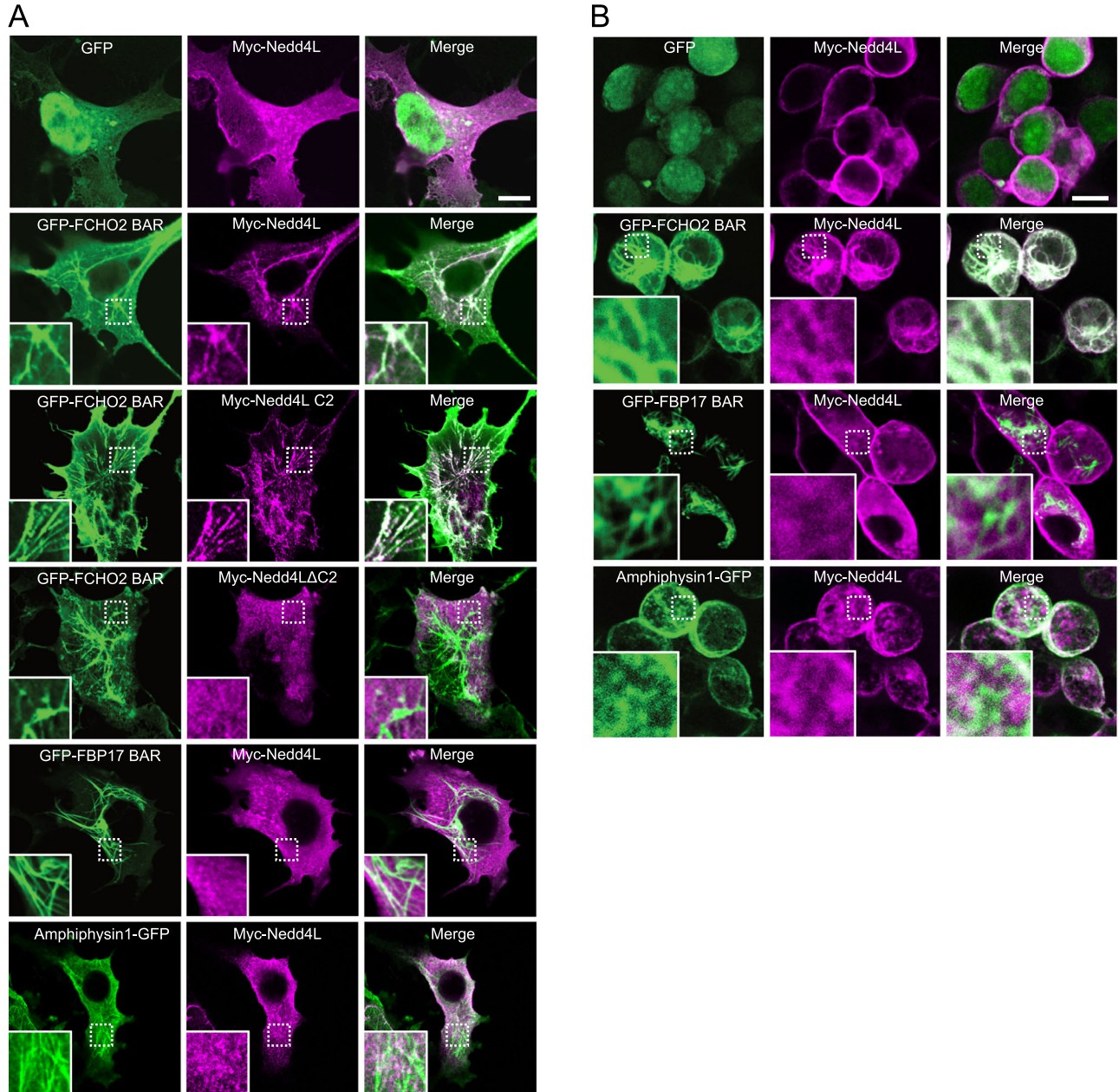

**Figure 4.   Recruitment of Nedd4L to membrane tubules generated by FCHO2 in cells.**

GFP-BAR domains were co-expressed with Myc-Nedd4L proteins in COS7 cells (**A**) and HEK293 cells (**B**). Cells were serum-starved and subjected to immunofluorescence microscopy. Insets represent enlarged images of the dashed boxes. Scale bar, 10 μm. Source data are available online for this figure.

## Roles of conserved hydrophobic residues in the Nedd4L C2 domain

We compared the C2 domain sequence of Nedd4L to those of other Nedd4 family members, SytI, and PKCγ (Fig. 8A). SytI C2 domains possess hydrophobic residues (M173, F234, V304, and I367) for penetration into the lipid bilayer to generate and sense membrane curvature (Martens et al, 2007). The Nedd4L C2 domain also has three hydrophobic residues (I37, F38, and L99), which are conserved among Nedd4 family members except for WWP2. We mutated these residues to alanine to produce six mutants, I37A,

F38A, L99A, I37A + F38A, I37A + L99A, and I37A + F38A + L99A (triple-A). Any single mutation did not inhibit Nedd4L recruitment to FCHO2-generated membrane tubules (Fig. 8B). However, double and triple mutations severely impaired this recruitment. Furthermore, the I37A + F38A and triple-A mutants showed little liposome binding and lost curvature generation (Fig. 8C,D). Thus, these hydrophobic residues (I37, F38, and L99) are critical for both membrane binding and curvature sensing and generation. These residues are likely inserted into the lipid bilayer, thereby strengthening the C2 domain interaction with membranes and enabling the sensing and generation of membrane curvature. It

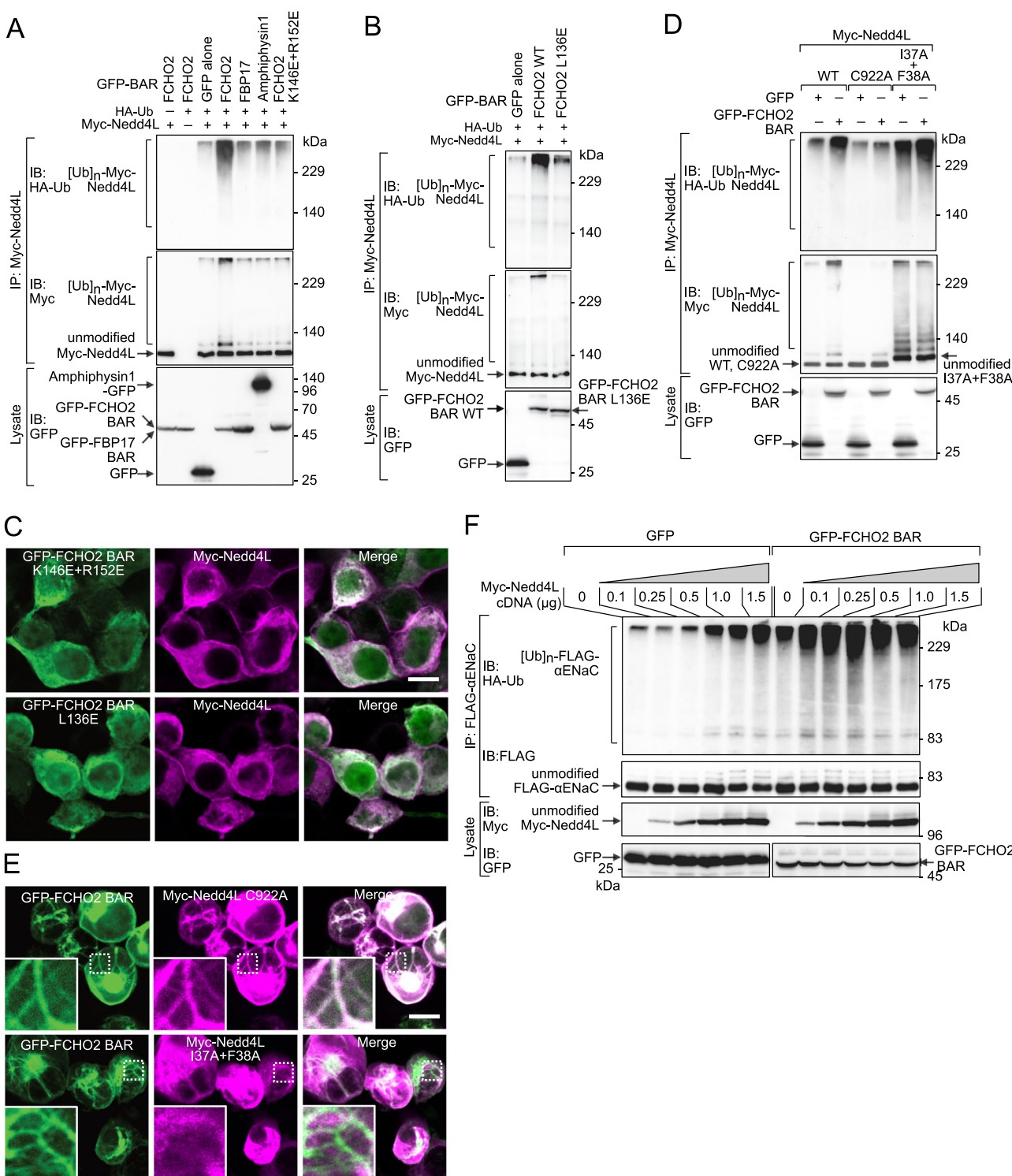

**Figure 5. Activation of Nedd4L by membrane tubules generated by FCHO2 in cells.**

(A–E) Nedd4L autoubiquitination. An in vivo ubiquitination assay was performed with various BAR domains and FCHO2 mutants (A, B) and Nedd4L mutants (D). GFP-BAR domains were co-expressed with Myc-Nedd4L proteins and HA-tagged Ub in HEK293 cells. Cell lysates were subjected to immunoprecipitation (IP) and analyzed by IB (A, B, D). WT, wild type. Cells were also subjected to immunofluorescence microscopy (C, E). Scale bar, 10 μm. (F) αENaC ubiquitination. An in vivo ubiquitination assay was performed with various concentrations of Myc-Nedd4L using FALAG-αENaC as a substrate in the presence or absence of GFP-FCHO2 BAR domain. Cell lysates were subjected to IP. Samples were analyzed by IB. Source data are available online for this figure.

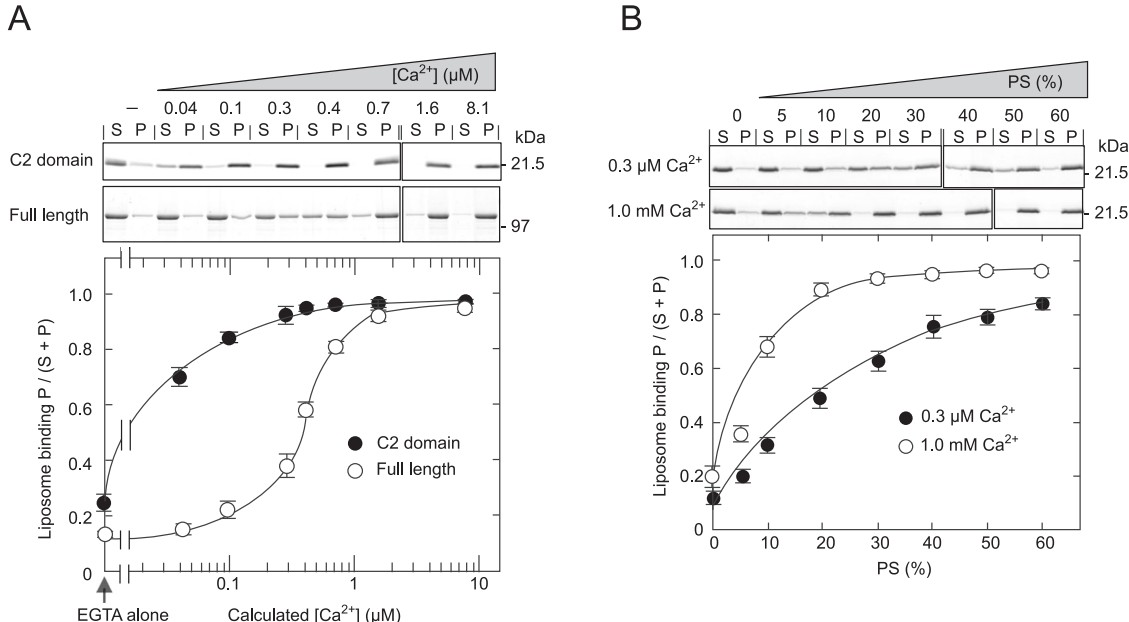

**Figure 6.    The Nedd4L C2 domain is a Ca²⁺-dependent PS binding domain.**

A co-sedimentation assay was performed with the C2 domain or full-length Nedd4L using brain-lipid liposomes ( ~ 50% PS) at various Ca²⁺ concentrations (**A**) and with the C2 domain using synthetic liposomes containing various percentages of PS at 0.3 μM or 1.0 mM Ca²⁺ (**B**). The supernatants (S) and pellets (P) were subjected to sodium dodecyl sulfate (SDS)-polyacrylamide gel electrophoresis (PAGE), followed by Coomassie brilliant blue (CBB) staining (upper panels). Bottom panels, quantitative analysis. ( − ) for [Ca²⁺] indicates EGTA alone. Data are shown as the mean ± SEM of three independent experiments. (**A**) Ca²⁺ dependency. (**B**) PS dependency. Source data are available online for this figure.

has been shown that I37 and F38 are directly involved in Ca²⁺ binding (Escobedo et al, 2014). The residues may be inserted into the lipid bilayer while binding Ca²⁺.

In the C2 domain of Smurf2, another Nedd4 family member, conserved hydrophobic residues (F29 and F30) are responsible for interactions with the HECT domain (Fig. 8A) (Wiesner et al, 2007). Mutation at these residues impairs the interaction between the C2 and HECT domains, resulting in enhanced Smurf2 autoubiquitination by relieving autoinhibition. Consistent with these previous findings, we found that the Nedd4L I37A + F38A mutant, which was not recruited to FCHO2-generated membrane tubules, enhanced autoubiquitination, irrespective of FCHO2 expression (Fig. 5D,E). These results suggest that I37 and F38 are critical for interactions with the HECT domain as well as for membrane binding and curvature sensing. Interestingly, the Nedd4L I37A + F38A mutant, as well as the C922A mutant, showed little αENaC ubiquitination (Fig. EV4), indicating that membrane binding of Nedd4L is critical for ENaC ubiquitination.

## In vitro activation of Nedd4L by FCHO2-generated membrane curvature

To examine the properties of Nedd4L catalytic activity, we carried out an in vitro ubiquitination assay using an αENaC peptide as a substrate. The αENaC peptide possesses a conserved PY motif for interaction with the Nedd4L WW domain, and it is fused with monomeric streptavidin (mSA) (Lim et al, 2013) to bind to liposomes containing biotinylated phospholipids (biotinylated

liposomes) (Sakamoto et al, 2017) (Fig. 9A,B). We found that Ca²⁺ and PS were required for Nedd4L catalytic activity (Fig. 9C,D), and that 0.05 μm pore-size liposomes were most effective in stimulating Ca²⁺- and PS-dependent Nedd4L activity (Fig. 9E). Thus, the properties of Nedd4L catalytic activity are similar to those of its liposome binding. Furthermore, we found that Nedd4L did not ubiquitinate PY motif-mutated (Y644A in αENaC) mSA-αENaC (Appendix Fig. S3). These findings are consistent with defects in ENaC ubiquitination observed in Liddle syndrome.

We next examined whether membrane localization of mSA-αENaC affects the membrane binding and catalytic activity of Nedd4L. Nedd4L bound only slightly to mSA-αENaC on PS-free (0% PS) liposomes, but the liposome binding was synergistically increased by elevating the PS percentage in liposomes (Fig. 9F). By contrast, when free biotin competitively inhibited the binding of mSA-αENaC to biotinylated liposomes, the PS-dependent liposome binding of Nedd4L was significantly reduced, resulting in remarkable inhibition of mSA-αENaC ubiquitination (Fig. 9D,F). Thus, membrane localization of the PY motif is critical for membrane binding and activation of Nedd4L. It is, therefore, likely that the PY motifs on membranes, such as ENaC, play a role in enhancing Nedd4L catalytic activity by potentiating membrane binding through interactions with the WW domain.

We then reconstituted in vitro FCHO2-induced recruitment and activation of Nedd4L. When liposomes (20% PS) were used, the FCHO2 BAR domain enhanced the liposome binding of Nedd4L (Fig. 10A) and mSA-αENaC ubiquitination (Fig. 10B). This enhancement was lost when the strength of interaction of Nedd4L

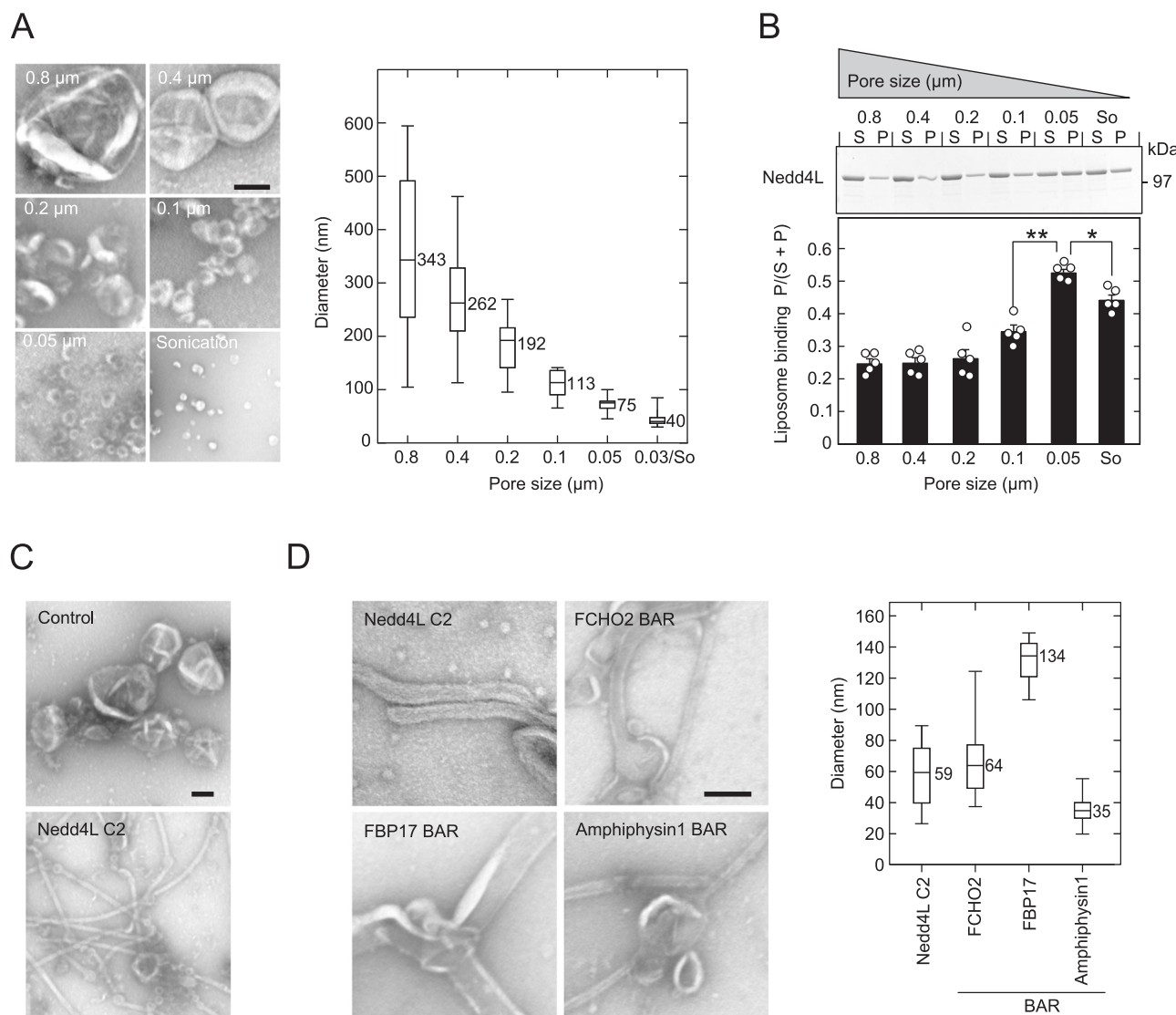

**Figure 7. Preference of Nedd4L for FCHO2-generated membrane curvature.**

(A) Distribution of liposome diameters. Brain-lipid liposomes (~50% PS) of different diameters were prepared by extrusion through the indicated pore sizes and sonication. Left panel, electron microscopic images. Scale bars, 200 nm. Right panel, distribution of liposome diameters shown by boxplots (number of observations per pore size = 25). The center line inside the box corresponds to the median, the bounds of the box encompass data points between the first and third quartiles, and the whiskers extend to the minimum and maximum values, including outliers. So, sonication. (B–D) Sensing and generation of membrane curvature by Nedd4L. Co-sedimentation (B) and in vitro tubulation (C, D) assays were performed using brain-lipid liposomes (~50% PS) at 0.1 μM (B) and 0.3 μM Ca$^{2+}$ (C, D). (B) Preference of Nedd4L for a specific degree of membrane curvature. Liposomes were supplemented with X-biotin-PE and rhodamine-PE. Full-length Nedd4L was incubated with liposomes of different sizes, which were prepared by extrusion through the indicated pore sizes. The sample was then incubated with avidin beads and ultracentrifuged. The proportion of precipitated liposomes of each size was calculated to be ~100% by measuring both total and supernatant fluorescence. The supernatants (S) and pellets (P) were subjected to SDS-PAGE, followed by CBB staining (upper panel). Bottom panel, quantitative analysis. So, sonication. Data are shown as the mean ± SEM of five independent experiments. *$P < 0.05$; **$P < 0.01$ (one-way analysis of variance with Tukey's post hoc test). (C) Curvature generation by Nedd4L. The assay was performed with the Nedd4L C2 domain. Samples were examined by electron microscopy. Scale bar, 100 nm. (D) The degree of membrane curvature generated by Nedd4L is consistent with that generated by FCHO2. The assay was performed with the indicated protein. Left panel, electron microscopic image. Scale bar, 100 nm. Right panel, distribution of tubule diameters shown by boxplots (number of observations per protein = 14–33). The center line inside the box corresponds to the median, the bounds of the box encompass data points between the first and third quartiles, and the whiskers extend to the minimum and maximum values including outliers. Source data are available online for this figure.

with liposomes was increased by elevating the PS percentage in liposomes (~50% PS) (Figs. 10C and EV5). In contrast to FCHO2, neither amphiphysin1 nor FBP17 enhanced Nedd4L activity (Figs. 10D and EV6A). Similarly, tubulation-deficient FCHO2 mutant (L136E) did not enhance Nedd4L activity (Figs. 10E and

EV6A). These results are in good agreement with those obtained from in vivo experiments on FCHO2-induced recruitment and activation of Nedd4L. Electron microscopic analysis revealed that mSA-αENaC-associated liposomes were deformed by membrane-sculpting proteins into tubules with diameters similar to those of

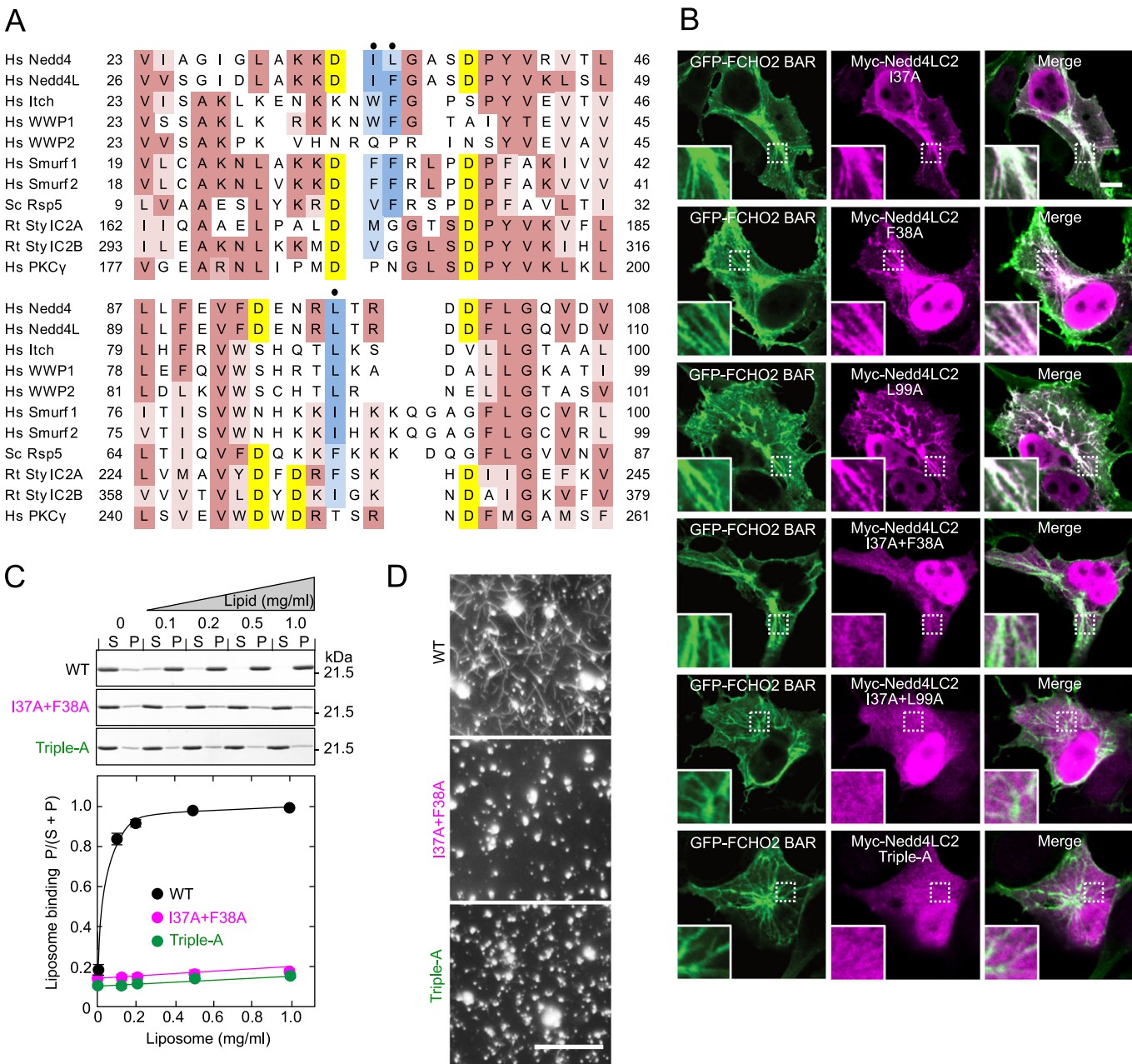

**Figure 8. Conserved hydrophobic residues of Nedd4L are responsible for sensing and generation of membrane curvature.**

(A) Sequence alignment of C2 domains. Conserved residues are highlighted with the following color code: yellow, aspartates for Ca$^{2+}$ binding; blue, very hydrophobic residues probably for penetration into the lipid bilayer; and brown, others. Dots show the mutation sites. Hs Homo sapiens, Sc *Saccharomyces cerevisiae*, Rt Rattus norvegicus. (B) Inability of Nedd4L C2 domain mutants to sense membrane curvature in cells. GFP-FCHO2 BAR domain and Myc-Nedd4L C2 domain mutants were co-expressed in COS7 cells. Insets are enlarged images of dashed boxes. Scale bar, 10 μm. (C, D) Inability of Nedd4L C2 domain mutants to bind to membranes and to generate membrane curvature in vitro. Co-sedimentation (C) and in vitro tubulation (D) assays were performed using brain-lipid liposomes (~50% PS) at 0.3 μM Ca$^{2+}$. (C) Membrane binding. The assay was performed with Nedd4L C2 domain mutants using various concentrations of liposomes. The supernatants (S) and pellets (P) were subjected to SDS-PAGE, followed by CBB staining (upper panel). Bottom panel, quantitative analysis. Data are shown as the mean ± SEM of three independent experiments. (D) Curvature generation. The assay was performed with Nedd4L C2 domain mutants. Samples were examined by fluorescence microscopy. Scale bar, 10 μm. Source data are available online for this figure.

mSA-αENaC-free liposomes (Fig. EV6B and see also Fig. 7D). The diameter of membrane tubules generated by FCHO2 is consistent with that of 0.05 μm pore-size liposomes which are most effective in stimulating Nedd4L activity.

To confirm whether FCHO2-induced recruitment and activation of Nedd4L are mediated by liposomes (i.e., membrane curvature) rather than by protein–protein interactions, we conducted two sets of experiments. Initially, we employed synthetic liposomes where 20% PS

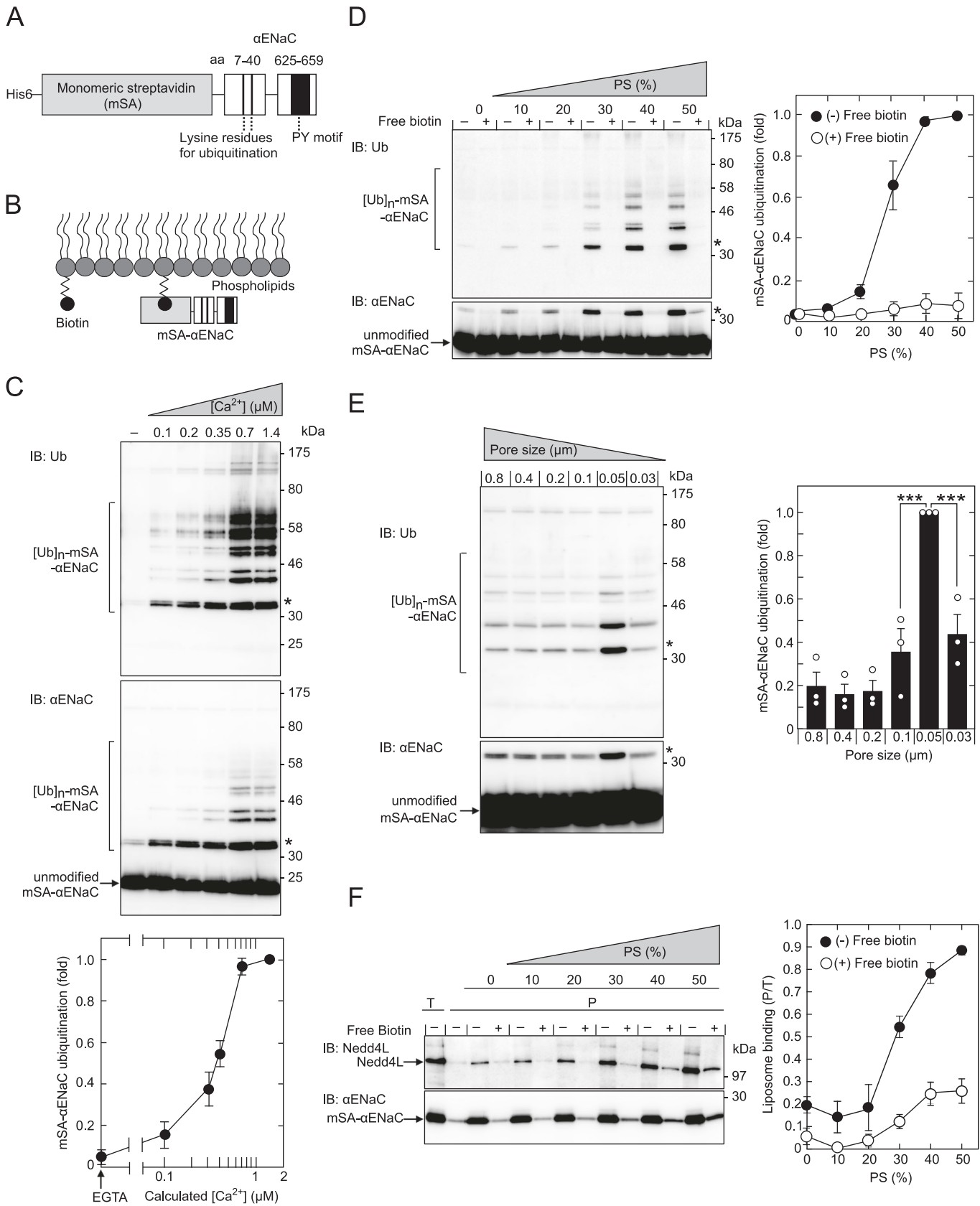

**Figure 9. Properties of the catalytic activity and membrane binding of Nedd4L.**

Liposomes supplemented with biotinylated phospholipids were incubated with mSA-αENaC (A, B). Samples were mixed with Nedd4L. Mixtures were subjected to ubiquitination (C–E) and co-sedimentation (F) assays. These assays were performed with brain-lipid liposomes (~50% PS) at various $Ca^{2+}$ concentrations (C), with synthetic liposomes (various percentages of PS) at 0.7 μM $Ca^{2+}$ (D, F), and with brain-lipid liposomes (20% PS) at 0.7 μM $Ca^{2+}$ (E). Ubiquitination and liposome binding were quantified by scanning and expressed as ratios to maximum levels. Asterisks indicate the same band of mono-ubiquitinated mSA-αENaC. (A, B) Schematic diagrams of mSA-αENaC. (A) Structure. (B) Binding to biotinylated liposomes. (C) $Ca^{2+}$-dependent Nedd4L activity. Ubiquitination was detected by IB (upper panel) and the intensity was quantified by scanning (bottom panel). Data are shown as the mean ± SEM of three independent experiments. (D) Membrane localization of mSA-αENaC enhances the PS-dependent catalytic activity of Nedd4L. Assays were performed in the presence or absence of 40 μM biotin. Ubiquitination was detected by IB (left panel). The intensities were quantified by scanning (right panel). Data are shown as the mean ± SEM of three independent experiments. (E) Nedd4L activity is predominantly stimulated by the specific degree of membrane curvature. The assay was performed using liposomes of different sizes which were prepared by extrusion through the indicated pore sizes. Ubiquitination was detected by IB (left panel) and quantified (right panel). Data are shown as the mean ± SEM of three independent experiments. ***$P < 0.001$ (one-way analysis of variance with Tukey's post hoc test). (F) Membrane localization of mSA-αENaC enhances the PS-dependent membrane binding of Nedd4L. Assays were performed in the presence or absence of 40 μM biotin. Liposome binding was detected by IB (left panel). The intensities were quantified by scanning (right panel). T total sample, P pellet. Data are shown as the mean ± SEM of three independent experiments. Source data are available online for this figure.

was substituted with 5% PI(4,5)P$_2$. The FCHO2 BAR domain bound to PI(4,5)P$_2$ as well as PS with high affinity, whereas the Nedd4L C2 domain did not (Fig. 11A,B). The FCHO2 BAR domain is known to induce the tubulation of liposomes [5% PI(4,5)P$_2$] (Henne et al, 2010). If Nedd4L could interact through protein–protein interactions with the FCHO2 BAR domain binding to liposomes [5% PI(4,5)P$_2$], the BAR domain would facilitate the co-precipitation of Nedd4L in a co-sedimentation assay, thereby stimulating mSA-αENaC ubiquitination. However, the FCHO2 BAR domain did not demonstrate such interactions (Fig. 11C,D). Therefore, it is unlikely that Nedd4L interacts with the FCHO2 BAR domain on liposomes through protein–protein interactions. Secondly, we employed 0.05 μm pore-size liposomes that are most effective in stimulating Nedd4L activity. The diameter of 0.05 μm pore-size liposomes is consistent with that of membrane tubules generated by the FCHO2 BAR domain. The BAR domain is, thus, expected not to induce the tubulation of 0.05 μm pore-size liposomes. The FCHO2 BAR domain showed similar binding activity toward 0.8 and 0.05 μm pore-size liposomes (Fig. 12A). When 0.8 μm pore-size liposomes were used, the FCHO2 BAR domain enhanced Nedd4L activity (Fig. 12B). However, in the case of 0.05 μm pore-size liposomes, curvature-stimulated Nedd4L activity was not potentiated by the addition of the FCHO2 BAR domain. If Nedd4L had interacted with the BAR domain, we would have expected potentiation of Nedd4L activity. These results strongly suggest that FCHO2-induced activation of Nedd4L is mediated by membrane curvature and not by protein–protein interactions.

# Discussion

Nedd4L represents the ancestral Ub ligase with strong similarity to yeast Rsp5 which plays a key role in the trafficking, sorting, and degradation of a large number of proteins (Yang and Kumar, 2010). Consistently, Nedd4L regulates a growing number of substrates, including membrane receptors, transporters, and ion channels. Although Nedd4L exists in a catalytically autoinhibited state due to an intramolecular interaction between the N-terminal C2 and C-terminal HECT domains, little is known about its activation mechanism. In this study, we clearly demonstrated that Nedd4L is activated by FCHO2 F-BAR domain-induced membrane curvature. It has been shown that the activity of synptojanin-1, a phosphoi-nositide phosphatase, is modulated by membrane curvature (Chang-Ileto et al, 2011). Taken together, it is conceivable that

membrane curvature serves as a signal for the activation of various enzymes involved in membrane trafficking.

## Unique properties of the Nedd4L C2 domain

Although both sytI and Nedd4L C2 domains are $Ca^{2+}$-dependent PS-binding domains that can generate and sense membrane curvature, their properties show a number of significant differences. An interesting difference exists in the degrees of membrane curvatures generated by sytI and Nedd4L: the diameters of their membrane tubules are about 15–20 nm and 40–80 nm, respectively (Hui et al, 2009; Martens et al, 2007). This difference is not simply due to that between tandem and single C2 domains, because it has been shown that the sytI C2B domain alone generates membrane tubules with a similar diameter (Hui et al, 2009). SytI and Nedd4L C2 domains must therefore have respective unique structures that determine the specific degrees of membrane curvatures that they sense and generate.

Another important difference between sytI and Nedd4L C2 domains exists in their $Ca^{2+}$ requirements. SytI C2 domains require $Ca^{2+}$ concentrations in the submillimolar range to generate membrane curvature (Hui et al, 2009). In contrast, submicromolar $Ca^{2+}$ concentrations, which are comparable to intracellular levels in unstimulated cells, are sufficient for the Nedd4L C2 domain to sense membrane curvature. This is consistent with the observation that Nedd4L is recruited and activated in serum-starved cells. The different $Ca^{2+}$ requirements between sytI and Nedd4L C2 domains are reflected in their functions. SytI C2 domains induce membrane curvature in response to an increase in intracellular $Ca^{2+}$ concentration, which is essential for $Ca^{2+}$-triggered membrane fusion (Hui et al, 2009; Martens et al, 2007). Thus, sytI C2 domains serve as a $Ca^{2+}$ sensor and a curvature generator. In contrast, the Nedd4L C2 domain recognizes a specific degree of curvature without an increase in intracellular $Ca^{2+}$ concentration. Therefore, the Nedd4L C2 domain serves as a curvature sensor to regulate the localization and activity of the E3 Ub ligase.

A striking feature that distinguishes the Nedd4L C2 domain from other C2 domain proteins (except for at least a subset of Nedd4 family members) is that the Nedd4L C2 domain interacts not only with membranes but also with the intramolecular catalytic domain for its autoinhibition. Mutational experiments have revealed that I37 and F38 residues in the C2 domain are critical for interactions with both membranes and the HECT domain,

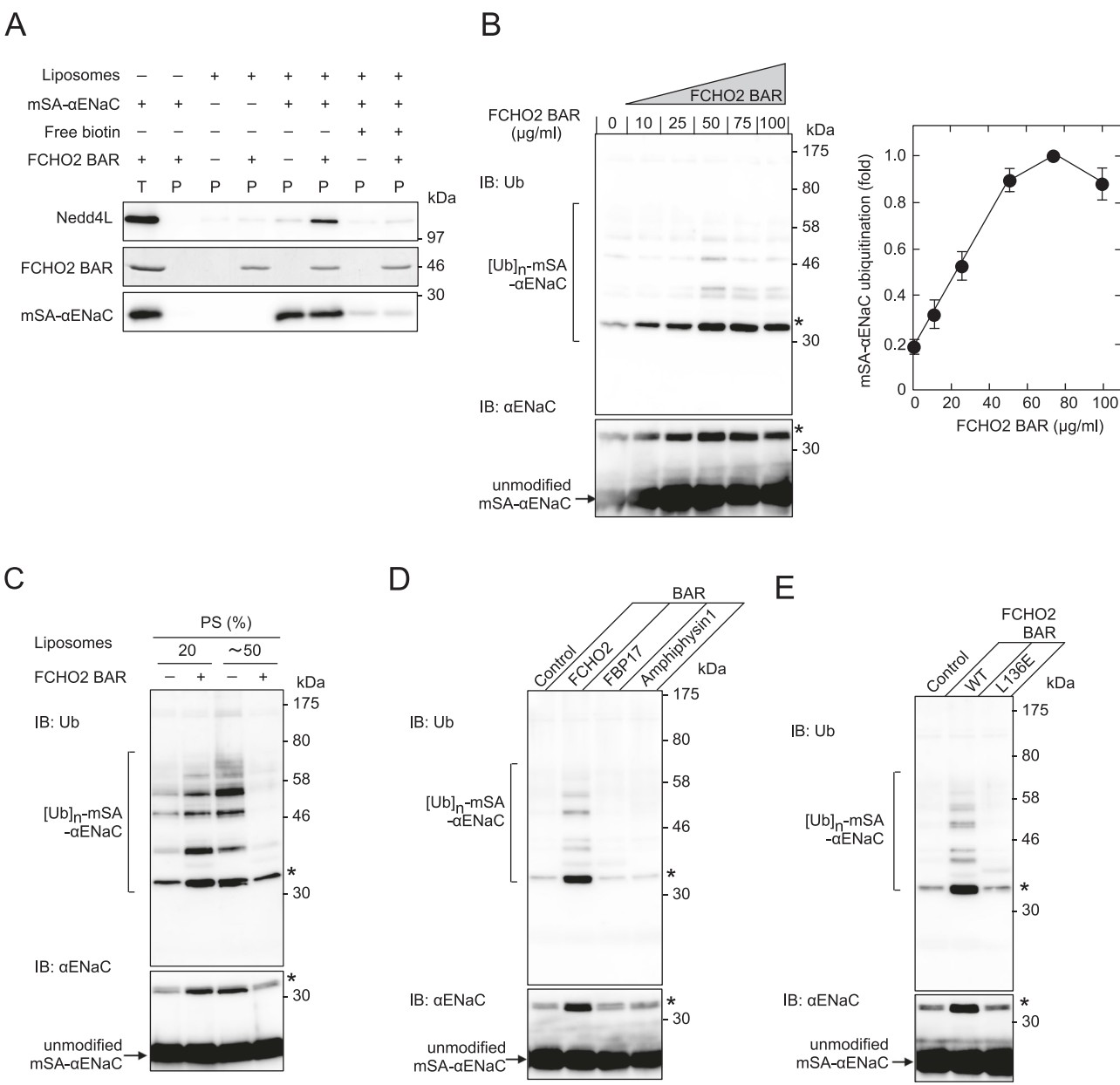

**Figure 10. Recruitment and activation of Nedd4L by FCHO2-generated membrane curvature in vitro.**

mSA-αENaC-associated brain-lipid liposomes (20% PS) were mixed with Nedd4L in the presence or absence of BAR domains. Mixtures were subjected to co-sedimentation (**A**) and ubiquitination (**B–E**) assays. These assays were performed at 0.7 μM $Ca^{2+}$ with brain-lipid liposomes (20%PS) (**A, B, D, E**) and with the indicated (20% PS or ~50% PS) brain-lipid liposomes (**C**). Ubiquitination and liposome binding were quantified by scanning and expressed as ratios to maximum levels. Asterisks indicate the same band of mono-ubiquitinated mSA-αENaC. (**A**) FCHO2-induced recruitment of Nedd4L. The assay was performed with the FCHO2 BAR domain (1.4 μM, 50 μg/ml) in the presence or absence of biotin. The total sample (T) and pellets (P) were subjected to SDS-PAGE, followed by CBB staining (FCHO2 BAR) and IB (Nedd4L and mSA-αENaC). (**B, C**) FCHO2-induced activation of Nedd4L. (**B**) Effects of various doses of the FCHO2 BAR domain. The assay was performed with various concentrations of the FCHO2 BAR domain. Ubiquitination was detected by IB (left panel) and the intensity was quantified by scanning (right panel). (**C**) Effects of the PS percentage in liposomes. The assay was performed with the FCHO2 BAR domain (1.4 μM). Samples were analyzed by IB. Data are shown as the mean ± SEM of three independent experiments. (**D, E**) Specificity of BAR domains and effect of an FCHO2 mutant. The assay was performed with various BAR domains (**D**) and an FCHO2 mutant (**E**) (1.4 μM each). Samples were analyzed by IB. Source data are available online for this figure.

suggesting that these two interactions are competitive. This idea is supported by the observation that the $Ca^{2+}$ $EC_{50}$ value of full-length Nedd4L is much higher than that of the C2 domain. The HECT domain in full-length Nedd4L may competitively inhibit the interaction of the C2 domain with membranes. It is likely that membranes displace the HECT domain from the C2 domain when the strength of the C2 domain interaction with membranes is increased by the preferred membrane curvature. This displacement relieves the autoinhibition, resulting in Nedd4L activation. Thus, the membrane binding of Nedd4L is coupled with its activation.

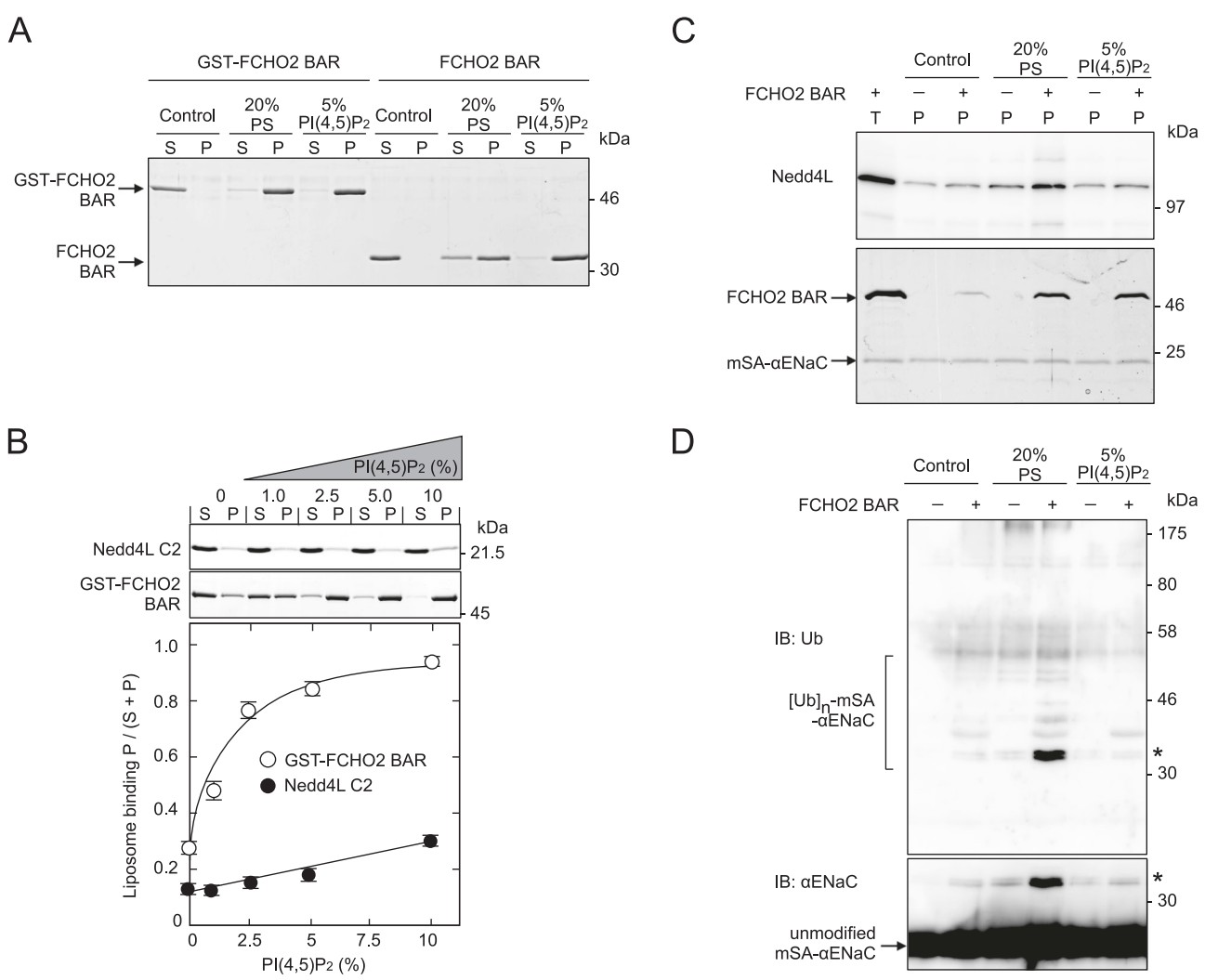

**Figure 11. Effects of PI(4,5)P₂ on FCHO2-induced recruitment and activation of Nedd4L.**

(A, B) Binding to PI(4,5)P₂. A co-sedimentation assay was performed at 0.7 μM Ca²⁺ with the FCHO2 BAR or Nedd4L C2 domain using mSA-αENaC-associated synthetic liposomes containing 20% PS, 5% PI(4,5)P₂ (A), or various percentages of PI(4,5)P₂ (B). The supernatants (S) and pellets (P) were subjected to SDS-PAGE, followed by CBB staining (A and upper panel in B). Bottom panel in (B), quantitative analysis. Control, synthetic liposomes [0% PS and 0% PI(4,5)P₂]. Data are shown as the mean ± SEM of three independent experiments. (C, D) Effects on FCHO2-induced recruitment and activation of Nedd4L. Co-sedimentation (C) and ubiquitination (D) assays were performed at 0.7 μM Ca²⁺ with mSA-αENaC-associated synthetic liposomes containing 20% PS or 5% PI(4,5)P₂ in the presence or absence of the FCHO2 BAR domain. (C) Nedd4L recruitment. The total sample (T) and pellets (P) were subjected to SDS-PAGE, followed by IB with anti-Nedd4L antibody (upper panel) and CBB staining (lower panel). (D) Nedd4L activation. Samples were analyzed by IB. Asterisks indicate the same band of mono-ubiquitinated mSA-αENaC. Source data are available online for this figure.

## Stimulatory and inhibitory effects of the FCHO2 BAR domain on the liposome binding and catalytic activity of Nedd4L

The FCHO2 BAR domain enhances both the liposome binding and catalytic activity of Nedd4L when the interaction strength of Nedd4L with liposomes is weak (20% PS) (Fig. 10A,B). Conversely, the BAR domain inhibits the binding and activity of Nedd4L when the interaction of Nedd4L with liposomes is increased by elevating the PS percentage in liposomes (~50% PS) (Figs. 10C and EV5). Thus, the FCHO2 BAR domain exerts both stimulatory and inhibitory effects on Nedd4L, depending on the PS percentage in liposomes. However, given that 20% PS is comparable to the concentration in the inner leaflet of

the plasma membrane, it is plausible that FCHO2 exhibits only a stimulatory effect on Nedd4L in cells. The mechanism underlying the stimulatory and inhibitory effects in vitro is as follows: when using liposomes (20% PS), Nedd4L hardly binds to liposomes through PS (Figs. 9F and 10A). The addition of the FCHO2 BAR domain enhances the strength of Nedd4L's interaction with PS by inducing membrane curvature. Consequently, Nedd4L gains a new binding to liposomes through PS, which augments its catalytic activity. In contrast, when using liposomes (~50% PS), Nedd4L binds to liposome through PS. The addition of the FCHO2 BAR domain hinders the PS-mediated liposome binding of Nedd4L, because FCHO2 and Nedd4L are PS-binding proteins and compete for PS binding in liposomes. Consequently, the BAR domain inhibits the catalytic activity of Nedd4L. This

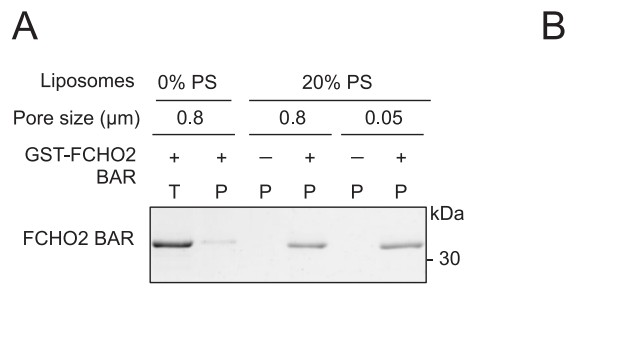

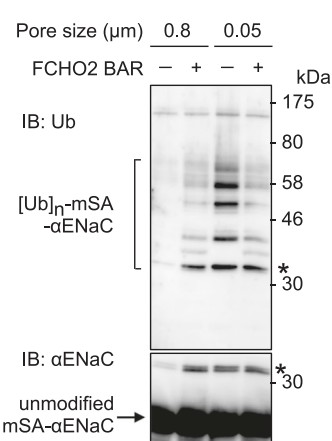

**Figure 12. Effects of liposome size on FCHO2-induced activation of Nedd4L.**

Co-sedimentation (**A**) and ubiquitination (**B**) assays were performed at 0.7 μM Ca²⁺ with the FCHO2 BAR domain using brain-lipid liposomes (20% PS, 0.8 μm or 0.05 μm pore-size) that were associated with mSA-αENaC. (**A**) Binding of the FCHO2 BAR domain to 0.8 μm and 0.05 μm pore-size liposomes. The co-sedimentation assay was performed using synthetic (0% PS) or brain-lipid liposomes (20% PS) containing rhodamine-PE and fluorescein-PE. Liposomes were precipitated with anti-fluorescein antibody. The total sample (T) and pellets (P) were subjected to SDS-PAGE, followed by CBB staining. The proportion of precipitated liposomes of each size was calculated to be ~100% by measuring both total and supernatant fluorescence. (**B**) Effects of liposome size on FCHO2-induced activation of Nedd4L. Samples were analyzed by IB. Asterisks indicate the same band of mono-ubiquitinated mSA-αENaC. Source data are available online for this figure.

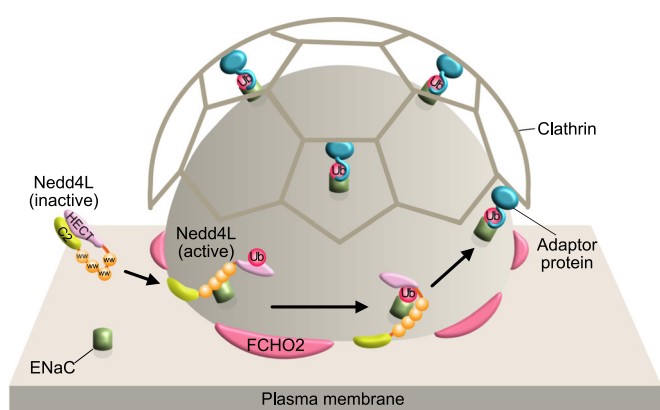

**Figure 13. Model of ENaC endocytosis.**

FCHO2 binds to the plasma membrane and generates a specific degree of membrane curvature according to the intrinsic curvature of its BAR domain. Nedd4L exists in a catalytically autoinhibited state due to an intramolecular interaction between the N-terminal C2 and C-terminal HECT domains. Nedd4L is recruited to FCHO2-generated membrane curvature at the rim of nascent CCPs that ENaC enters. Membrane binding of the C2 domain causes catalytic activation by relieving autoinhibition. ENaC is ubiquitinated and captured by adaptor proteins with Ub-interacting motifs, such as Eps15.

is consistent with the results obtained with 0.05 μm pore-size liposomes (Fig. 12B). These 0.05 μm pore-size liposomes are comparable to liposomes with a higher PS percentage, as the interaction of Nedd4L with liposomes through PS is increased by membrane curvature. Therefore, when using 0.05 μm pore-size liposomes, the FCHO2 BAR domain inhibits the catalytic activity of Nedd4L.

Based on a model in which a subset of BAR domain proteins, such as FCHO2 and FBP17, drive liposome tubulation in vitro (Frost et al, 2008; Shimada et al, 2007), the FCHO2 BAR domain is considered to polymerize on the liposome surface into spiral dense coats by lateral and tip-to-tip interactions. When using liposomes (~50% PS), it is conceivable that the FCHO2 BAR domain densely packs the liposomes, leaving little free space for Nedd4L binding. Conversely, when using liposomes (20% PS), it is likely that the liposomes are loosely packed, providing ample surface space for Nedd4L binding.

## Model for ENaC endocytosis

Based on the present study, we propose a model for ENaC endocytosis through ubiquitination (Fig. 13). First, FCHO2 binds to the plasma membrane and generates a specific degree of membrane curvature. FCHO2 interacts with clathrin adaptor proteins, such as Eps15, and recruits the clathrin machinery for CCP formation (Uezu et al, 2011; Henne et al, 2010). Eps15 has Ub-interacting motifs (Di Fiore et al, 2003). Nedd4L is subsequently recruited to and activated by FCHO2-generated curved membrane at the rim of nascent CCPs that ENaC enters. ENaC is then ubiquitinated and captured by Eps15 which FCHO2 interacts with. Thus, ubiquitinated ENaC is spatially restricted to CCPs and separated from unmodified ENaC. Overall, our findings suggest that in the early stage of CME, a membrane deformation induced by FCHO2 can program, through both the recruitment and activation of Nedd4L, the gathering of a subclass of cargo molecules in CCPs.

## Methods

### Construction

Human αENaC, βENaC and γENaC cDNAs were kindly supplied by Dr. H. Kai (Kumamoto University, Kumamoto, Japan)

(Sugahara et al, 2009). FLAG-αENaC was constructed by replacing the sequence [amino acid (aa) 169–174, TLVAGS] in the extracellular region with the FLAG sequence (DYKDDDK) as described previously (Firsov et al, 1996). This introduction of the epitope tag was carried out by polymerase chain reaction (PCR). FLAG-αENaC, βENaC, and γENaC were subcloned into pTRE-Tight-Hygro, pCAGI-Puro (Miyahara et al, 2000), and pCAGI-Bst (Umeda et al, 2006), respectively. pTRE-Tight-Hygro was constructed by inserting the hygromycin-resistance gene into pTRE-Tight (Clontech). FLAG-αENaC was also subcloned into pCAGI-Hygro, which was constructed by replacing the puromycin-resistance gene of pCAGI-Puro with the hygromycin-resistance gene. Human Nedd4L (NM_015277, isoform 3) cDNA was obtained by reverse transcription (RT)-PCR and cloned in pCMV-Myc (Miyahara et al, 2000). Nedd4L was also cloned in pCAGI-puro-EGFP and pGEX-6P-hexahistidine (His6). pGEX-6P-His6 was constructed from pGEX-6P (GE Healthcare) to express an N-terminal GST-tagged, C-terminal His6-tagged protein. pCAGI-puro-EGFP was constructed by subcloning EGFP cDNA into pCAGI-puro to express a C-terminal GFP-tagged protein. The Nedd4L C2 domain (C2, aa 1–160) and C2 domain-deleted mutant (ΔC2, aa 150–955) were subcloned into pCMV-Myc and/or pGEX-6P. Mouse FCHO2 (full length, aa 1–809; BAR domain, aa 1–302) and human FBP17 (BAR domain, aa 1–300) were subcloned into pEGFP-C1 and/or pGEX-6P (Uezu et al, 2011; Tsujita et al, 2006). FCHO2 (aa 1–327) were subcloned into pGEX-6P. Human amphiphysin1 cDNA was obtained by RT-PCR and cloned in pEGFP-N3. Amphiphysin1 (BAR domain, aa 1–377) was then subcloned into pGEX-6P. mTagRFP-T-Clathrin-15 was a gift from Michael Davidson (Addgene plasmid # 58005; http://n2t.net/addgene:58005; RRID:Addgene_58005) (Shaner et al, 2008). Human hemagglutinin (HA)-tagged Ub cDNA in pCGN was kindly donated by Dr. M. Nakao (Kumamoto University, Kumamoto, Japan). Ub cDNA was subcloned into pET3a. Human UbcH7 cDNA was obtained by RT-PCR and cloned in pGEX-6P. pET21d-Ube1 was a gift from Cynthia Wolberger (Addgene plasmid # 34965; http://n2t.net/addgene:34965; RRID:Addgene_34965) (Berndsen and Wolberger, 2011). pRSET-mSA was a gift from Sheldon Park (Addgene plasmid # 39860; http://n2t.net/addgene:39860; RRID:Addgene_39860) (Lim et al, 2013). mSA-αENaC constructed in pRSET-mSA to express a fusion protein of the N-terminal (aa 7–40) and C-terminal (aa 625–659) intracellular regions of αENaC (see Fig. 9A). Mutagenesis was performed by PCR using mutated primers and a site-directed mutagenesis kit (Stratagene).

## Protein expression and purification

Expression and purification of GST-tagged proteins were performed using standard protocols. His6-tagged proteins were purified with Talon metal affinity resin (Clontech) using the manufacturer's protocol. The GST tag was cleaved off by PreScission protease (GE Healthcare). The GST tag of BAR domains did not inhibit liposome binding or tubulation, whereas the GST tag of Nedd4L inhibited these functions. As FCHO2 (BAR domain, aa 1–302) was easily aggregated by the removal of the GST tag, FCHO2 (aa 1–327) was used instead as a GST-cleaved form of the FCHO2 BAR domain. Full-length Nedd4L (N-terminal GST-tagged, C-terminal His6-tagged) was purified by

glutathione–Sepharose (GE Healthcare) column chromatography. The GST tag was cleaved off and the Nedd4L sample was then subjected to Talon metal affinity column chromatography. After the eluate was passed through glutathione–Sepharose beads, the sample was dialyzed against buffer A (20 mM Tris-HCl at pH 7.5, 100 mM KCl, 0.5 mM EDTA, 1 mM DTT, and 20% sucrose). The Nedd4L sample was stored at −80 °C until use. Ub was expressed in E. coli strain BL21 (DE3) pJY2 and purified as previously described (Pickart and Raasi, 2005). His6-tagged mSA-αENaC was expressed in E. coli strain BL21 (DE3) pLysS and purified as previously described (Lim et al, 2013) with slight modifications. Briefly, the inclusion body was isolated and solubilized in buffer B (50 mM Tris-HCl at pH 8.0 and 6 M guanidine hydrochloride). The sample was then subjected to Talon metal affinity column chromatography. The eluate was added drop by drop to ice-cold refolding buffer (50 mM Tris-HCl at pH 8.0, 150 mM NaCl, 0.15 mg/ml D-biotin, 0.2 mg/ml oxidized glutathione, and 1 mg/ml reduced glutathione) under rapid stirring. The sample was then concentrated using a Centriprep centrifugal filter unit (Millipore) and centrifuged to remove aggregates. The supernatant was dialyzed against buffer C (20 mM Tris-HCl at pH 8.0). The mSA-αENaC sample was further purified by Mono Q anion exchange column chromatography as follows. The sample was applied to a Mono Q 5/50 GL column (GE Healthcare) equilibrated with buffer C. Elution was performed with a 50 ml linear gradient of NaCl (0–200 mM) in buffer C. Fractions of 1 ml each were collected. The active fractions (fraction 18–22) were collected and dialyzed against buffer D (10 mM Tris-HCl at pH 7.5, 100 mM KCl, and 0.5 mM EDTA). The mSA-αENaC sample was stored at −80 °C until use.

## Antibodies

Rabbit anti-Nedd4L antibody was raised against GST-Nedd4L C2 (aa 1–160). Rabbit anti-FCHO2 antibody was obtained as previously described (Uezu et al, 2011). The following antibodies were purchased from commercial sources: mouse anti-Myc (9E10) (American Type Culture Collection); mouse anti-FLAG (M2) and mouse anti-α-tubulin (clone DM1A) (Sigma-Aldrich); mouse anti-clathrin heavy chain (BD Biosciences); rabbit anti-GFP (MBL Co.); rabbit anti-αENaC (Calbiochem); rabbit anti-βENaC (Proteintech); rabbit anti-γENaC (Abcam); mouse anti-transferrin receptor H68.4 (Santa Cruz); mouse anti-Ub antibody P4D1 (Santa Cruz); and secondary antibodies conjugated with Alexa Fluor 488, 594, and 647 (Invitrogen).

## Cell culture and transfection

HeLa, HEK293, and COS7 cells were cultured in Dulbecco's modified Eagle medium (DMEM) supplemented with 10% fetal calf serum. HEK293 cells were grown on poly-L-lysine-coated dishes. Transfection was performed by using Lipofectamine 2000 (Invitrogen) according to the manufacturer's protocol.

## Generation of stable HeLa cells expressing all three ENaC subunits

HeLa cells were transfected with the pTet-On Advanced vector (Clontech) to develop stable Tet-On cell lines for a doxycycline (Dox)-inducible expression system according to the manufacturer's

protocol. Tet-On cells were selected in 0.5 mg/ml G418. Cloned Tet-On cells were then transfected with pCAGI-Bst-γENaC and selected in 4 μg/ml blasticidin (in the presence of G418). Blasticidin-resistant cells were transfected with pCAGI-Puro-βENaC and selected in 1 μg/ml puromycin (in the presence of G418 and blasticidin). Puromycin-resistant cells were finally transfected with pTRE-Tight-Hygro-FLAG-αENaC and cultured in 0.2 mg/ml hygromycin (in the presence of G418, blasticidin, and puromycin). After selection, cells were cloned and maintained in 0.5 mg/ml G418, 0.2 mg/ml hygromycin, 1 μg/ml puromycin, and 2.5 μg/ml blasticidin. FLAG-tagged αENaC expression was induced by treating cells with 1 μg/ml Dox in the presence of 40 μM amiloride overnight. Amiloride was added to prevent cellular $Na^+$ overload.

## Immunofluorescence microscopy

Cells were fixed with 3.7% formaldehyde in phosphate-buffered saline (PBS) at room temperature for 15 min, and then permeabilized with 0.2% Triton X-100 in PBS for 10 min. After blocking with 1% bovine serum albumin (BSA) in PBS, cells were sequentially incubated with primary antibodies and fluorescence-conjugated secondary antibodies at room temperature for 1 h, respectively. Cells were then analyzed with either a fluorescence microscope (BX51; Olympus) or a confocal microscopy system (BX50, Fluoview FV300, and Fluoview FV1200; Olympus). A 60× oil immersion objective (NA = 1.40, Olympus) was used. All comparable images were acquired under identical parameters. To visualize cell-surface FLAG-αENaC in αβγENaC-HeLa cells, cells were incubated with 5 μg/ml anti-FLAG M2 antibody at 4 °C for 1 h to prevent internalization. Cells were then fixed, permeabilized, and stained with a secondary antibody. Cells were also sequentially incubated with primary (anti-FCHO2) and secondary antibodies. To quantify co-localization, the Pearson's correlation coefficient was calculated from three cells using the Fiji (ImageJ) Coloc2 plug-in.

## RNA interference

Stealth™ double-stranded RNAs were purchased from Invitrogen. siRNA sequences of human FCHO2, Nedd4L, and FBP17 were 5′-CCA CAG AUC UUA GAG UGG AUU AUA A-3′ (corresponding to nucleotides 2048–2072 relative to the start codon), 5′-GAA GAG UUG CUG GUC UGG CCG UAU U-3′ (corresponding to nucleotides 1778–1802 relative to the start codon), and 5′-CCC ACU UCA UAU GUC GAA GUC UGU U-3′ for human FBP17 (corresponding to nucleotides 1804–1828 relative to the start codon), respectively. A double-stranded RNA targeting luciferase (GL-2, 5′-CGU ACG CGG AAU ACU UCG AAA UGU C-3′) was used as a control. HeLa cells were transfected with 20 nM siRNA using Lipofectamine 2000 according to the manufacturer's protocol. After 24 h, a second transfection was performed, and cells were cultured for 3 days and subjected to various experiments. For rescue experiments, cells were transfected with the intended plasmid at 36 h after the second transfection. After 18 h, cells were fixed and analyzed.

## Quantitative real-time PCR analysis

Total RNA was isolated using ISOGEN (Nippon Gene, 319-90211), and cDNA was synthesized using ReverTra Ace qPCR RT Master Mix with gDNA Remover (Toyobo, FSQ-301), following manufacturer's instruction. Real time PCR was carried out using Thunderbird Next SYBR (Toyobo, QPX-201) on StepOnePlus Real-Time PCR Systems (Applied Biosystems). The sequences of qPCR primers are 5′-TCT CCC ATC GCT TCA ACG AG-3′ and 5′-GTT GCA CCC AGC TTG AGA GA-3′ for human FBP17; 5′-GTC TCC TCT GAC TTC AAC AGC G-3′ and 5′-ACC ACC CTG TTG CTG TAG CCA A-3′ for human GAPDH. The expression of target genes was normalized to GAPDH mRNA levels.

## Endocytosis assay

αβγENaC-HeLa cells at ~90% confluence in a 10-cm dish were starved with serum-free medium (DMEM containing 1% BSA, 1 μg/ml Dox, and 40 μM amiloride) at 37 °C for 4 h. For ENaC endocytosis, cells were then incubated with 5 μg/ml anti-FLAG M2 in 2 ml of DMEM containing 1% BSA at 4 °C for 1 h to prevent internalization. After three washes with ice-cold PBS containing 1 mM $MgCl_2$ and 0.1 mM $CaCl_2$ (PBS-CM), endocytosis was allowed by incubating cells in prewarmed DMEM containing 1% BSA at 37 °C for the indicated periods of time. Endocytosis was stopped by placing cells on ice and washing them with ice-cold PBS-CM. Antibody bound to the cell surface but not internalized was removed by acid stripping with 0.2 M acetic acid and 0.5 M NaCl solution (pH 2.5). After washing with ice-cold PBS-CM, internalized ENaC was detected by immunoblotting (IB) or immunofluorescence microscopy. For IB, cells were scraped in 1 ml of lysis buffer A (50 mM Tris-HCl at pH 8.0, 150 mM NaCl, 5 mM EDTA, 1% Triton X-100, 1 mM PMSF, 20 μg/ml leupeptin, and 1 μg/ml pepstatin A). Cells were solubilized at 4 °C for 1 h on a rotating wheel, and centrifuged at 20,000 × g at 4 °C for 15 min. Protein G-Sepharose beads (GE Healthcare) were added to the supernatant and incubated at 4 °C for 3 h. The beads were then thoroughly washed with lysis buffer A and boiled in sodium dodecyl sulfate (SDS) sample buffer. The sample was subjected to SDS-polyacrylamide gel electrophoresis (PAGE), followed by IB with anti-ENaC subunit antibodies. Internalization was expressed as a percentage of the initial amount of cell-surface ENaC subunit determined by incubating cells with anti-FLAG antibody at 4 °C for 1 h and lysing them without incubation at 37 °C or acid stripping. For immunofluorescence microscopy, cells were transfected with control or FCHO2 siRNA and then with GFP-siRNA-resistant (sr) FCHO2 mutant. After cells were labeled with anti-FLAG antibody on 4 °C, they were incubated at 37 °C for 10 min to allow ENaC endocytosis. After acid stripping, cells were fixed with 3.7% formaldehyde and permeabilized with 0.2% Triton X-100. The antibody-labeled, internalized αENaC was visualized with a fluorescence-conjugated secondary antibody. The percentage of cells with internalized αENaC among cells expressing GFP-sr-FCHO2 was determined using MetaMorph imaging System software. All images were acquired under identical parameters. For TfR endocytosis, a cell surface biotinylation assay was carried out as previously described (Roberts et al, 2001). Briefly, the cell surface was labeled with Sulfo-NHS-SS-Biotin (Thermo Fisher Scientific) at 4 °C. After cells were incubated with ice-cold quenching buffer (50 mM $NH_4Cl$ in PBS-CM) at 4 °C, they were incubated with serum-free medium for 10 min at 37 °C. Biotin was removed from the cell surface by incubation with ice-cold MesNa buffer (50 mM MesNa, 100 mM NaCl, and 100 mM Tris-HCl, pH 8.8). Cells were then incubated with iodoacetamide at 4 °C. Cell lysates were then incubated with NeutrAvidin beads

(Thermo Fisher Scientific). The beads were boiled in SDS sample buffer. Proteins were analyzed by IB with TfR antibody.

## ENaC expression and ubiquitination at the cell surface

αβγENaC-HeLa cells at approximately 90% confluence in a 10-cm dish were starved with serum-free medium (DMEM containing 1% BSA, 1 μg/ml Dox, and 40 μM amiloride) at 37 °C for 4 h, washed three times with ice-cold PBS, and incubated at 4 °C for 45 min with 2 ml of 1 mg/ml EZ-link Sulfo-NHS-SS-Biotin in PBS with gentle shaking. Subsequently, cells were extensively washed with ice-cold Tris-buffered saline, and then scraped into 1 ml of lysis buffer B (50 mM Tris-HCl at pH 7.5, 150 mM NaCl, 5 mM EDTA, 1% Triton X-100, 10 mM N-ethylmaleimide, 1 mM PMSF, 10 μg/ml leupeptin, and 1 μg/ml pepstatin A) containing 40 μM MG132. Cells were solubilized at 4 °C for 1 h on a rotating wheel, and centrifuged at $20,000 \times g$ at 4 °C for 15 min. The supernatant (4 mg of protein) was incubated at 4 °C for 3 h with 10 μg of anti-FLAG M2 antibody. Protein G-Sepharose beads (50 μl of 50% slurry) were then added to the sample, which was incubated at 4 °C for 3 h. After beads were thoroughly washed with lysis buffer B containing 40 μM MG132, FLAG-tagged proteins were eluted from the beads with 200 μg/ml FLAG peptide (Sigma-Aldrich) in 0.5 ml of lysis buffer B. After centrifugation, the supernatant was incubated with 25 μl of NeutrAvidin beads at 4 °C overnight. After the beads were thoroughly washed with lysis buffer B, bound proteins were eluted by incubating the beads at 4 °C for 1 h with 150 μl of lysis buffer B containing 200 mM DTT. After centrifugation, 5× SDS sample buffer was added to the supernatant, which was then analyzed by IB with anti-FLAG and anti-Ub antibodies. The chemiluminescence intensity was quantified by scanning using ImageJ software (NIH).

## TIRF microscopy

For live observation with TIRF microscopy, HeLa cells were grown on a glass-bottomed dish as described previously (Shimada et al, 2007). Cells were observed at 10–12 h after transfection as described previously (Henne et al, 2010). Images were acquired for 5 min at 1 s intervals with a TIRF microscopy system (Olympus), a 100× oil immersion objective NA = 1.45 (Olympus) and MetaMorph Imaging System Software. All comparable images were acquired under identical parameters.

## Liposome preparation

Phospholipids were purchased from commercial sources: phosphatidylcholine (PC, 1,2-dioleoyl-*sn*-glycero-3-phosphocholine), phosphatidylethanolamine (PE, 1,2-dioleoyl-*sn*-glycero-3-phosphoethanolamine), PS (1,2-dioleoyl-*sn*-glycero-3-phospho-L-serine), rhodamine-PE [18:1 L-α-PE-N-(lissamine rhodamine B sulfonyl)], PEG2000-biotin-PE (1,2-distearoyl-*sn*-glycero-3-phosphoethanolamine-N-[biotinyl (polyethyleneglycol)-2000]), porcine brain PS, brain PI(4,5)P$_2$, and porcine brain total lipid extract were obtained from Avanti Polar Lipids; cholesterol and Folch lipids [bovine brain lipids (Folch Fraction 1)] were obtained from Sigma-Aldrich; X-biotin-PE [N-((6-(biotinoyl)amino)hexanoyl)-1,2-dihexadecanoylsn-glycero-3-phosphoethanolamine] was obtained from Thermo Fisher Scientific; and fluorescein-PE [N-(fluorescein-5-thiocarbamoyl)-1,2-dihexadecanoyl-sn-glycero-3-phosphoethanolamine] was obtained from AAT Bioquest Inc.

Liposomes were prepared using brain lipids (Folch lipids, ~50% PS of total lipids), brain lipids (20% PS of total lipids, a mixture of 89.5% porcine brain total lipid extract and 10.5% porcine brain PS), or a synthetic lipid mixture [70% PC, 20% PE, 10% cholesterol (w/w), and varying percentages of PS or PI(4,5)P$_2$ (with a corresponding reduction in PC)]. Lipid concentrations were monitored by including 1% rhodamine-PE (w/w). To prepare mSA-αENaC-associated liposomes, 2% PEG2000-biotin-PE (w/w) was included. Lipids were dried under nitrogen gas, desiccated for 2 h, and resuspended at 1 mg/ml in lipid buffer (50 mM HEPES-NaOH at pH 7.2, 100 mM KCl, 2 mM MgCl$_2$, and 5 mM EGTA), followed by hydration at 37 °C for 1 h. To make liposomes of different sizes, they were subjected to five freeze–thaw cycles and centrifuged at $100,000 \times g$ for 15 min. Precipitated liposomes were then resuspended in the same buffer to make a final concentration of 1 mg/ml. They were subjected to extrusion through polycarbonate filter membranes (Avanti Polar Lipids): liposomes were sequentially passed 9 times through a 0.8-μm filter, 21 times through a 0.4-μm filter, 21 times through a 0.2-μm filter, 21 times through a 0.1-μm filter, 21 times through a 0.05-μm filter, and 21 times through a 0.03-μm filter. Instead of extrusion through a 0.03-μm filter, liposomes with the smallest size were also prepared by sonication three times for 5 s with a Branson Sonifier 25 at power level 1. Liposome diameters were examined by transmission electron microscopy.

## Co-sedimentation assay

A co-sedimentation assay was performed as described (Uezu et al, 2011) with slight modifications. Briefly, 5–10 μg of protein was incubated at 25 °C for 10 min with 0.5 mg/ml liposomes in 100 μl of lipid buffer containing various concentrations of CaCl$_2$ and centrifuged at $165,000 \times g$ at 25 °C for 15 min. Equal amounts of the supernatants and pellets were subjected to SDS-PAGE, followed by Coomassie brilliant blue (CBB) staining. Proteins were quantified by scanning using ImageJ (NIH). The assay with liposomes of different sizes was performed using NeutrAvidin (Thermo Fisher Scientific Inc) as described (Sakamoto et al, 2017). Briefly, brain-lipid (Folch-lipid) liposomes (~50% PS) were supplemented with 1% X-biotin-PE and 1% rhodamine-PE, subjected to extrusion through filter membranes, and then sonicated. Purified Nedd4L protein (10 μg) was incubated with 0.2 mg/ml liposomes in 100 μl of lipid buffer containing 2 mM CaCl$_2$ (0.1 μM Ca$^{2+}$). The sample was then incubated with 2 μg of NeutrAvidin and ultracentrifuged. Both total and supernatant fluorescence were measured. The proportion of precipitated liposomes of each size was ~100%.

The assay with mSA-αENaC-associated liposomes was performed as follows. Liposomes were prepared from a synthetic lipid mixture [various percentages of PS or 5% PI(4,5)P$_2$] or brain lipids supplemented with 2% PEG2000-biotin-PE and 1% rhodamine-PE. The liposomes (10 μg) were incubated with purified mSA-αENaC protein (35 pmol) at 4 °C for 10 min in 25 μl of lipid buffer. The sample was then incubated at 25 °C for 10 min with 0.07 μM (8 μg/ml) Nedd4L and/or 1.4 μM BAR domains in 50 μl of ubiquitination buffer (50 mM HEPES-NaOH at pH 7.2, 100 mM KCl, 2 mM MgCl$_2$, 5 mM EGTA, and 0.1 mM DTT) containing 4 mM CaCl$_2$ (0.7 μM Ca$^{2+}$). Subsequently, the sample was subjected to ultracentrifugation. Equal amounts of the total samples and pellets were subjected to SDS-PAGE, followed by CBB staining and IB. The

chemiluminescence intensity was quantified by scanning using ImageJ (NIH).

The assay with mSA-αENaC-associated liposomes (0.05 μm pore-size) was done as follows. Liposomes were prepared from brain lipids (20% PS) supplemented with 2% PEG2000-biotin-PE, 1% rhodamine-PE, and 0.5% fluorescein-PE (w/w) and extruded through membrane filters. To precipitate liposomes, the sample (50 μl) was first incubated with 1 μg of anti-fluorescein goat antibody (Novus Biologicals) at room temperature for 1 h and then incubated with 0.5 μg of protein A/G/L (BioVision) at room temperature for 1 h. Subsequently, the sample was centrifuged at 165,000 × g at 25 °C for 15 min. Equal amounts of the supernatants and pellets were subjected to SDS-PAGE, followed by CBB staining and IB. The percentage of precipitated liposomes was ~100%.

## Calculation of free $Ca^{2+}$ concentrations

At 1.0, 2.0, 3.0, 3.5, 4.0, 4.5, and 4.9 mM $CaCl_2$ in lipid buffer, free $Ca^{2+}$ concentrations were calculated to be 0.04, 0.1, 0.3, 0.4, 0.7, 1.6, and 8.1 μM, respectively, as determined by the Ca-Mg-ATP-EGTA Calculator v1.0 using constants from the NIST database #46 v8. At 1.0, 2.0, 3.0, 3.5, 4.0, 4.5, and 4.9 mM $CaCl_2$ in ubiquitination buffer, free $Ca^{2+}$ concentrations were calculated to be 0.04, 0.1, 0.2, 0.4, 0.7, 1.4, and 6.7 μM, respectively.

## In vitro tubulation assay

An in vitro tubulation assay was performed as described (Uezu et al, 2011; Tsujita et al, 2006) with slight modifications. Briefly, liposomes were prepared from brain lipids (Folch lipids) supplemented with 1% rhodamine-PE. The liposomes (0.2 mg/ml liposomes) were incubated with the Nedd4L C2 domain (4.5 μM) or BAR domains (1.4 μM) at 25 °C for 10 min in lipid buffer containing 3 mM $CaCl_2$ (0.3 μM $Ca^{2+}$). The sample was examined by fluorescence or electron microscopy. The tubulation assay with mSA-αENaC-associated liposomes was performed using the sample mixture utilized in the co-sedimentation assay in which mSA-αENaC-associated liposomes (brain-lipid liposomes) were used.

## Negative-staining electron microscopy

Negative-staining transmission electron microscopy was performed as described (Shimada et al, 2007). Box and whisker plots were used to represent the distribution of liposome diameters.

## In vivo ubiquitination assay

HEK293 cells were used, because they formed membrane tubules upon FCHO2 expression more efficiently than did COS7 cells. HEK293 cells at approximately 90% confluence on a 6-well plate were transfected with 0.8 μg of pCMV-Myc-Nedd4L, 1.6 μg of pEGFP-BAR domain, and 1.6 μg of pCGN-HA-Ub. In some experiments, cells were transfected with various doses (0–1.5 μg) of pCMV-Myc-Nedd4L, 1.5 μg of pEGFP-FCHO2 BAR domain, 1 μg of pCGN-HA-Ub, and 0.5 μg of pCAGI-Hygro-FLAG-αENaC. Total transfected cDNA amount was held constant using empty pCMV-Myc vector. Amiloride (40 μM) was added to the medium when αENaC cDNA was transfected. At 21 h after transfection,

cells were washed with PBS and incubated with serum-free DMEM containing 1% BSA. After 3 h incubation, 10 μM MG132 was added to the medium. After further 4 h incubation, cells were solubilized in 150 μl of lysis buffer C (50 mM Tris-HCl at pH 7.5, 150 mM NaCl, 5 mM EDTA, 10 mM N-ethylmaleimide, 1 mM PMSF, 10 μg/ml leupeptin, and 1 μg/ml pepstatin A) containing 1% SDS. After boiling, the sample was sonicated and then centrifuged at 20,000 × g for 15 min. The supernatant was diluted 10-fold with lysis buffer C containing 1% Triton X-100 and then incubated at 4 °C for 3 h with either anti-Myc or anti-FLAG M2 antibody. Protein G-Sepharose beads were added to the sample, which was then incubated at 4 °C for 3 h. After the beads were thoroughly washed with lysis buffer C containing 1% Triton X-100, bound proteins were eluted by boiling in SDS sample buffer for 5 min. These proteins were then subjected to SDS-PAGE, followed by IB.

## In vitro ubiquitination assay

mSA-αENaC-associated liposomes were made by incubating liposomes (10 μg) with mSA-αENaC (35 pmol) in 25 μl of lipid buffer, as described for the co-sedimentation assay. They were then incubated at 25 °C for 10 min with 0.07 μM Nedd4L, 0.04 μM E1 (Ube1), 0.7 μM E2 (UbcH7), 7 μM Ub, and various concentrations of BAR domains in 50 μl of ubiquitination buffer containing 4 mM $CaCl_2$ (0.7 μM $Ca^{2+}$) and 2 mM ATP. Where indicated, various concentrations of $CaCl_2$ were used. The ubiquitination reaction was started by adding ATP and stopped by adding 10 μl of 6× SDS sample. The sample was then boiled, subjected to SDS-PAGE, and immunoblotted with anti-Ub and anti-αENaC antibodies. The chemiluminescence intensity was quantified by scanning using ImageJ (NIH).

## Other procedures

Stripping immunoblots for re-probing was carried out with the WB Stripping Solution Strong (Nacalai Tesque) according to the manufacturer's protocol. Protein concentrations were quantified with BSA as a reference protein by the Bradford method (Bio-Rad) or the BCA method (Thermo Fisher Scientific). SDS-PAGE was performed as described (Laemmli, 1970).

## Statistical analysis

All experiments were performed at least three times (biological replication) and representative results are shown. Data are shown as the mean ± SEM. Student's tests, one-way analysis of variance test, and two-way analysis of variance test were used to evaluate statistical significances between different treatment groups.

# Data availability

This study includes no data deposited in external repositories.

The source data of this paper are collected in the following database record: biostudies:S-SCDT-10_1038-S44318-024-00268-1.

## Peer review information

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

## Acknowledgements

We thank Drs. H Kai and M Nakao for kindly providing us with ENaC and Ub cDNAs, respectively. We are also grateful to Dr. T Takenawa (Kobe University, Kobe, Japan) for helpful discussions. This work was partly carried out at the Institute of Molecular Embryology and Genetics, the Gene Technology Centre, and Research Facilities of the School of Medicine, Kumamoto University. This study was supported by JSPS KAKENHI Grant Numbers 19K06643 (to HN), 19K06544 (to YS), and Setsuro Fujii Memorial, The Osaka Foundation for Promotion of Fundamental Medical Research (to YS) and by grants from the Mochida Memorial Foundation, the Takeda Science Foundation, and the Uehara Memorial Foundation (to KK).

## Author contributions

**Yasuhisa Sakamoto**: Investigation; Writing—original draft; Writing—review and editing. **Akiyoshi Uezu**: Investigation. **Koji Kikuchi**: Investigation. **Jangmi Kang**: Investigation. **Eiko Fujii**: Investigation. **Toshiro Moroishi**: Investigation. **Shiro Suetsugu**: Resources; Investigation. **Hiroyuki Nakanishi**: Conceptualization; Supervision; Funding acquisition; Investigation; Methodology; Writing—original draft; Project administration; Writing—review and editing.

Source data underlying figure panels in this paper may have individual authorship assigned. Where available, figure panel/source data authorship is listed in the following database record: biostudies:S-SCDT-10_1038-S44318-024-00268-1.

## Disclosure and competing interests statement

The authors declare no competing interests.

# Expanded View Figures

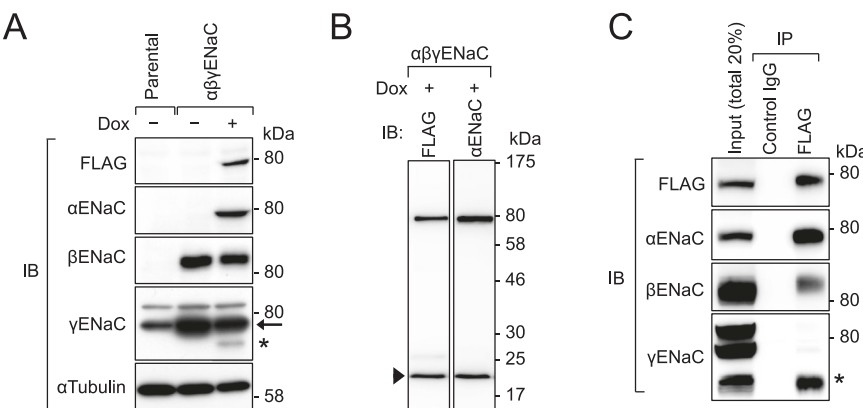

**Figure EV1.  Expression of ENaC subunits in αβγENaC-HeLa cells.**

(A, B) Doxycycline (Dox)-induced expression of αENaC. αβγENaC-HeLa cells were cultured overnight in the presence or absence of Dox. The lysates of parental HeLa and αβγENaC-HeLa cells were subjected to IB. (A) 7.5% gel. (B) Gradient gel (5-20%). Anti-γENaC antibody cross-reacted with two bands of endogenous proteins in parental HeLa cells, the lower band of which overlapped with γENaC (arrow) in αβγENaC-HeLa cells. Upon Dox-induced αENaC expression, an additional 70-kDa γENaC band (asterisk) was detected. It has been shown that co-expression of all three subunits induces ENaC maturation, including proteolytic cleavage of α- and γENaC (Hughey et al, 2003). The 70-kDa γENaC band is likely a cleavage product comprising the C-terminal region. The 20-kDa αENaC band (arrowhead) is likely a cleavage product comprising the N-terminal region. (C) Association of α-, β-, and γENaC. When αENaC was immunoprecipitated with anti-FLAG antibody from αβγENaC-HeLa cells treated with Dox, β- and γENaC were co-precipitated. Asterisk, 70-kDa γENaC band. Source data are available online for this figure.

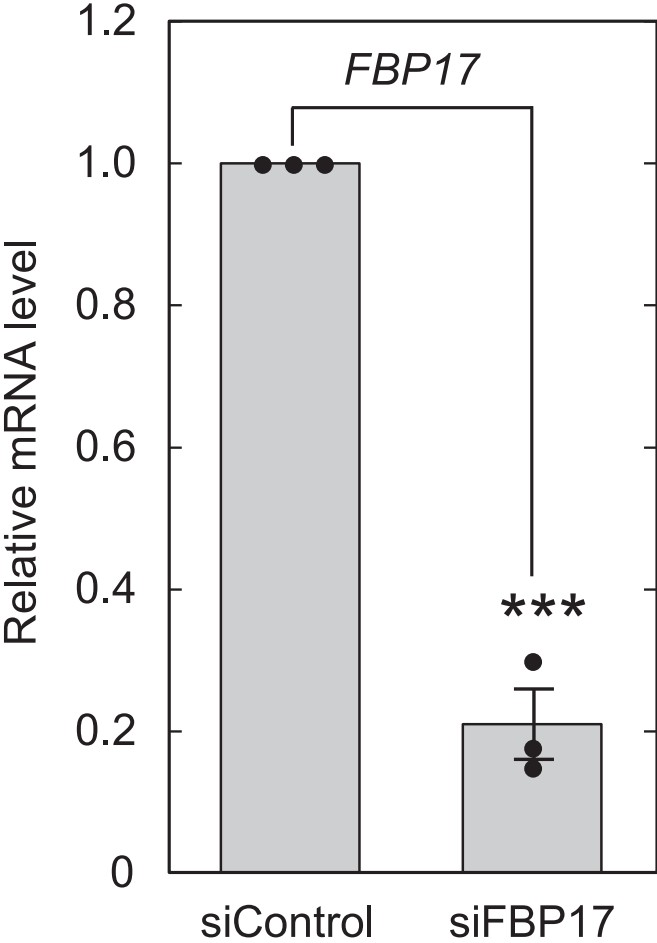

**Figure EV2. Knockdown of FBP17 by siRNA.**

αβγENaC-HeLa cells were treated with each siRNA, and FBP17 mRNA levels were quantified using real-time PCR. The expression of FBP17 was normalized to GAPDH mRNA levels. Data are shown as the mean ± SEM of three independent experiments. ***$P < 0.001$ (Student's $t$ test). Source data are available online for this figure.

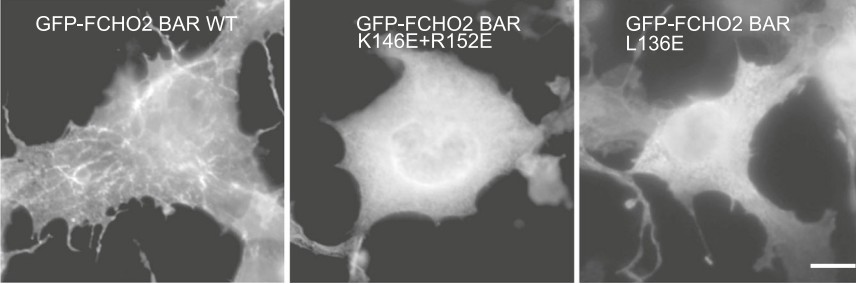

**Figure EV3. Inability of FCHO2 mutants to generate membrane tubules.**

GFP-FCHO2 BAR domain [wild type (WT) or mutant] was expressed in COS7 cells. Cells were then subjected to immunofluorescence microscopy. Scale bar, 10 μm. Source data are available online for this figure.

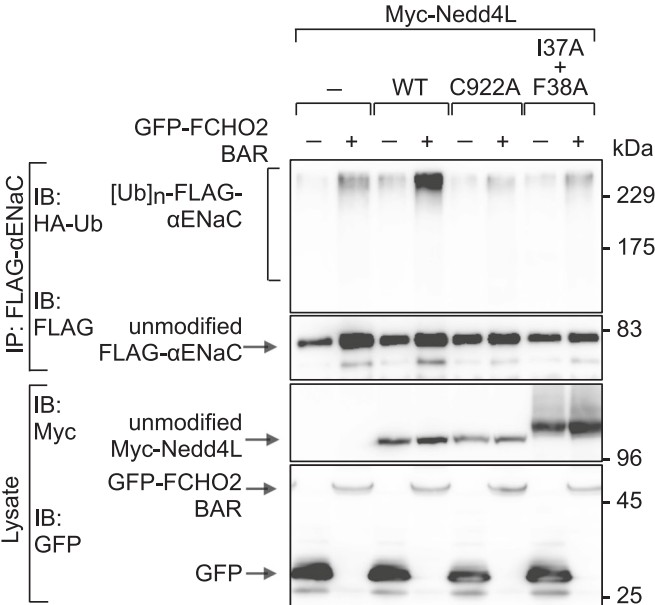

**Figure EV4.   Inability of Nedd4L mutants to ubiquitinate αENaC.**

An in vivo ubiquitination assay was performed with various Nedd4L constructs (each 0.5 μg) using FLAG-αENaC as a substrate in the presence or absence of GFP-FCHO2 BAR domain. Cell lysates were subjected to IP. Samples were analyzed by IB. Source data are available online for this figure.

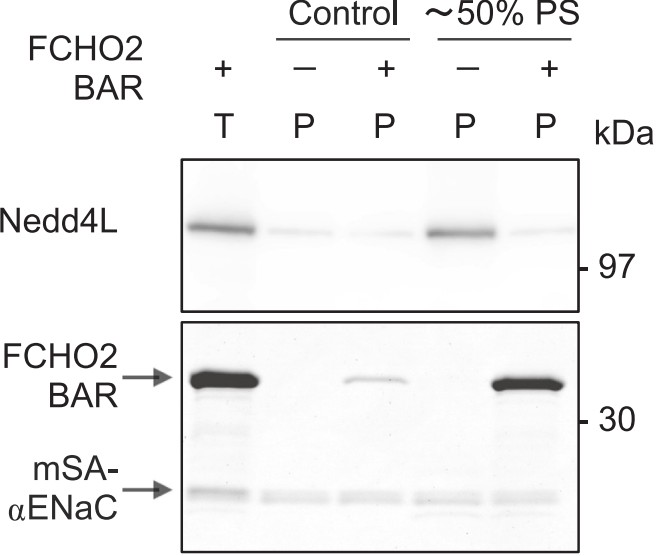

**Figure EV5.   Inhibition of the liposome binding of Nedd4L by the FCHO2 BAR domain.**

A co-sedimentation assay was performed at 0.7 μM Ca²⁺ with control liposomes (0% PS) or brain-lipid liposomes (~50% PS) in the presence or absence of the FCHO2 BAR domain. The total sample (T) and pellets (P) were subjected to SDS-PAGE, followed by IB (upper panel) and CBB staining (lower panel). Source data are available online for this figure.

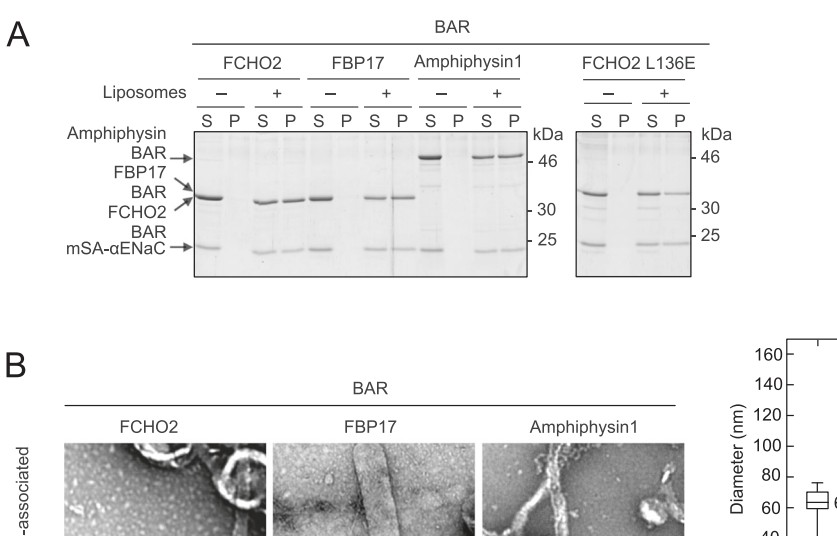

**Figure EV6.  Membrane binding and curvature generation of BAR domains and an FCHO2 mutant.**

Co-sedimentation (**A**) and in vitro tubulation (**B**) assays were performed at 0.7 μM $Ca^{2+}$ with the indicated BAR domains using brain-lipid liposomes (20% PS) that were associated with mSA-αENaC. (**A**) Membrane binding. The supernatants (S) and pellets (P) were subjected to SDS-PAGE followed by CBB staining. (**B**) Curvature generation. Left panel, electron microscopic image. Scale bar, 100 nm. Right panel, distribution of tubule diameters shown by boxplots (number of observations per protein = 22–26). The center line inside the box corresponds to the median, the bounds of the box encompass data points between the first and third quartiles, and the whiskers extend to the minimum and maximum values including outliers. Source data are available online for this figure.

