## [Peer Review File · The EMBO Journal]

The Nedd4L ubiquitin ligase is activated by FCHO2-generated membrane curvature

Yasuhisa Sakamoto, Akiyoshi Uezu, Koji Kikuchi, Jangmi Kang, Eiko Fujii, Toshiro MOROISHI, Shiro Suetsugu, and Hiroyuki Nakanishi

Corresponding author(s): Hiroyuki Nakanishi (hnakanis@gpo.kumamoto-u.ac.jp)

Review Timeline:

Transfer from Review Commons:	6th Jun 23
Editorial Decision:	15th Jun 23
Revision Received:	22nd Dec 23
Editorial Decision:	2nd Feb 24
Revision Received:	2nd Aug 24
Editorial Decision:	4th Sep 24
Revision Received:	17th Sep 24
Accepted:	27th Sep 24

Editor: Cornelius Schneider

Transaction Report:

Review #1

1. Evidence, reproducibility and clarity:

Evidence, reproducibility and clarity (Required)

I enjoyed reading the paper by Sakamoto and colleagues, where they show that Nedd4L ubiquitin ligase activity is stimulated by membranes and in particular positive membrane curvature. This paper is a conceptual advance that hopefully will be extended by many other groups where membranes topology participates in the activation of associated enzymes, giving rise to added complexity but also specificity and further compartmentalization. It is an important paper for all cell biologists to understand.

My comments are all relatively minor and I hope can improve the readability of the paper, but will not alter the overall conclusion as this is well backed up. In general I would like to see more/better statistics/quantitation and better figure legends. I found that often one had to read the paper to understand a figure where reading the figure legend should suffice.

This paper reminds me of a paper from Gilbert Di Paolo's lab on the activation of synaptojanin PIP2 hydrolysis by high membrane curvature. One would expect that there may be many such proteins whose activities will be dependent on their membrane environment. I find it conceptually rather likely that a protein which interacts with membranes via a C2 domain (which has membrane insertions and will thus likely be curvature sensitive) will likely show some positive curvature sensitivity. Can I suggest this paper is referenced and discussed in the light of the discussion statement "Thus, our findings provide a new concept of signal transduction in which a specific degree of membrane curvature serves as a signal for activation of an enzyme that regulates a number of substrates."

Where the paper could be improved (or I have not understood fully)

In figure 1 there is a robust endocytosis of ENaC that is FCHO2 and Nedd4L sensitive. There is a rescue for FCHO2 in a fluorescence image (unquantified), so it would be good to have the more quantitative approach of rescue with both FCHO2 and Nedd4L in the biochemical assay.

In figure 2 there is nice co-localisation between clathrin/FCHO2 and ENaC but not with Nedd4L. It would be good to have some quantitation of the co-localisation. But also one should use a Nedd4L mutant or a mutant of ENaC and so be able to visualise co-localisation between receptor and ubiquitin ligase. I find it strange that there is no (or much less) Nedd4L-GFP visible in the cells overexpressing ENaC... Is there an explanation? Does overexpression of ENaC lead to more auto-ubiquitination of Nedd4L. Also the Nedd4L-GFP signal in other cells is punctate, while in the next figure Myc-Nedd4L is not.

In figure 3 it appears to me that there is co-localization between ENaC and amphiphysin. Is this not a positive piece of information? I am not sure that FBP17 is a good F-BAR domain to use given its oligomerization may well prevent membrane association of Nedd4L. Minor comment: I don't see tubules for amphiphysin in panel B.

Figure 5: The affinity of Nedd4 C2 domain for calcium is quite high given we normally assume a cytosolic concentration of 100nM (approximate). The authors have rightly buffered the calcium with EGTA. Normally we would check that the buffering is sufficient by varying the protein concentration and making sure the affinity is still the same, so can I suggest the authors use 3 or 4 times the amount of C2 domain and make sure the curve does not change (provided liposomes are not limiting). Minor comment: How many experiments and what are error bars (SD?).

Figure 6: Controls have been performed to ensure that liposomes are pelleted, according to methods. In Figure 6B can the authors show that there is the same amount of liposomes in each sample by showing more of the coomassie gel so that the reader can see the Neutraavidin band is the same in each sample. Also I believe a student t-test should not be used in this experiment (but perhaps an Anova test), and in panel D there does not appear to be a description of statistics.

Figure 11: In panel B I note that the FCHo2 BAR domain on small liposomes appears to inhibit Ubiquitination. Is this consistent with the BAR domain not preventing Nedd4L binding?

2. Significance:

Significance (Required)

I enjoyed reading the paper by Sakamoto and colleagues, where they show that Nedd4L ubiquitin ligase activity is stimulated by membranes and in particular positive membrane curvature. This paper is a conceptual advance that hopefully will be extended by many other groups where membranes topology participates in the activation of associated enzymes, giving rise to added complexity but also specificity and further compartmentalization. It is an important paper for all cell biologists to understand.

3. How much time do you estimate the authors will need to complete the suggested revisions:

Estimated time to Complete Revisions (Required)

(Decision Recommendation)

Less than 1 month

4. Review Commons values the work of reviewers and encourages them to get credit for their work. Select 'Yes' below to register your reviewing activity at Web of Science Reviewer Recognition Service (formerly Publons); note that the content of your review will not be visible on Web of Science.

Yes

Review #2

1. Evidence, reproducibility and clarity:

Evidence, reproducibility and clarity (Required)

The authors have reported the involvement of the BAR domain-containing protein FCHO2 in the Nedd4L-mediated endocytosis of ENaC. They propose a model in which the membrane curvature induced by the BAR domain-FCHO2 relieves the auto-inhibition of E3 ligase causing its activation and recruitment. The paper describes a series of in vitro reconstituted experiments that are interesting but not fully connected with the mechanism of ENaC endocytosis. Additional experiments are needed to fully support the authors' conclusions.

Major comments:

1. Although the data reported by the authors regarding FCHO2 and Nedd4L involvement in ENaC endocytosis are convincing, it is suggested that the authors perform the same ENaC endocytosis assay presented in Fig.1B under conditions of FBP17 and amphiphysin1 siRNA to formally prove the selective involvement of FCHO2 in the process among other BAR-containing proteins.
2. According to the previous point, it will be interesting to see not only a snapshot image of the internalisation assay performed by immunofluorescence (Fig.1C) but a more quantitative analysis of the different time points (as in Fig.1B) in condition of FCHO2 siRNA and eventually FBP17 and amphiphysin1 siRNA.
3. In Fig.2B, overexpression of the catalytically inactive version of Nedd4L (Nedd4L C922A) would help to see Nedd4L-ENaC co-localization.
4. In Fig.4D, the authors need to analyse ENaC ubiquitination in the same experimental setting as Fig. 4A instead of transfecting cells with increasing amounts of Nedd4L in the presence or absence of FCHO2 BAR. It is also recommended to include Nedd4L C922A as an additional control.
5. While discussing the role of hydrophobic residues in Nedd4L C2 domain, the authors never mentioned the publication by Escobedo et al., Structure 2014 (DOI:10.1016/j.str.2014.08.016), which highlighted how I37 and L38 are directly involved in Ca²⁺ binding. This aspect should be discussed since the authors show the importance of Ca²⁺ for PS binding in the sedimentation assay.
6. As stated by the authors those two residues I37 and L38 are also involved in E3 enzyme activation by relieving C2-HECT interaction. It is important to further demonstrate the effect of these mutations on ENaC substrate.
7. There are some concerns regarding the in vitro ubiquitination assay performed in Fig.8 and following figures. The Nedd4L proteins used during the assay has been produced as His tagged at the C-terminus, it was reported (Maspero et al, Nat Struct Mol Biol 2013 DOI: 10.1038/nsmb.2566), at least for the isolated HECT domain, that modification of the C-terminal residue of the protein affects its activity. It would be important to judge the activity of the purified proteins used in the assay.

Moreover, as additional control it is suggested the introduction of a mSA-ENaC PY mutant protein. The authors claimed the importance of membrane localized PY motif for recruitment and activation of Nedd4L, it would be informative to perform the experiment in presence of PY mutated ENaC.

8. It is not clear why increasing the concentration of PS (from 20% to 50%) the presence of BAR domain doesn't allow ENaC ubiquitination (Fig.9C), is Nedd4L not recruited to the pellet? It would be interesting to see the sedimentation experiment of Fig.9A done in presence of 50% PS.
9. This reviewer is not an expert of lipids biology, thus the explanations related to the effect of FCHO2 BAR in presence of PI(4,5)P2 (Fig. 10) or 0.05 pore-size liposomes (Fig.11) were not clear. Does FCHO2 BAR have a different effect in inducing membrane tubulation in these two conditions? Is this parameter measurable by tubulation assay?

****Minor Comments****

1. It would be appreciated if a nuclei staining panel is included in all immunofluorescence images, as it would help to identify the number of cells in the field of view (e.g., Fig. 1C, Fig. 2B).
2. It would be recommended to include colocalization analysis, such as Pearson's correlation coefficient or Manders coefficient in immunofluorescence images.
3. It is not clear how the quantitation of mSA-ENaC ubiquitination in Fig. 8D, 8C, and 9B was performed. Did the authors normalise the detected Ub signal over the amount of unmodified mSA-ENaC?

****Referee cross-commenting****

I agree with the comments of other two reviewers.

2. Significance:

Significance (Required)

Unfortunately do to limited knowledge of the reviewer on the lipids biology field it is difficult to judge strengths and limitations of the last part of the manuscript.

3. How much time do you estimate the authors will need to complete the suggested revisions:

Estimated time to Complete Revisions (Required)

(Decision Recommendation)

Between 3 and 6 months

4. Review Commons values the work of reviewers and encourages them to get credit for their work. Select 'Yes' below to register your reviewing activity at Web of Science Reviewer Recognition Service (formerly Publons); note that the content of your review will not be visible on Web of Science.

No

Review #3

1. Evidence, reproducibility and clarity:

Evidence, reproducibility and clarity (Required)

****Summary****

The manuscript by Sakamoto et al. describes how the ubiquitin ligase Nedd4L is activated by membrane curvature generated by the endocytic protein FCHO2. For their experiments, the authors use the epithelial sodium channel (ENaC) as a model Nedd4L target and CME cargo. The authors start their manuscript by showing in cells the importance of FCHO2 and Nedd4L in ENaC internalization. Using a combination of experiments in cells and biochemistry, the authors show that Nedd4L binds preferentially to membranes with the same curvature generated by FCHO2. Next, the authors show that a combination of membrane composition (PS), calcium concentration, PY domain presence and membrane curvature all act in concert to recruit Nedd4L to membranes and fully release its ubiquitination activity. Crucially, the authors show that role of FCHO2 in Nedd4L recruitment is not direct, with FCHO2 simply generating an optimal membrane curvature for Nedd4L binding. Taken together, the authors suggest a mechanism by which the curvature of early clathrin coated pits, generated by FCHO1/2 define an optimal environment for the recruitment and activation of the ubiquitin ligase Nedd4L.

The manuscript convincingly shows the membrane curvature-dependent mechanism of Nedd4L activation. The biochemistry experiments in the manuscript are well designed and the results are of clear. The quality of these experiments is very high. The experiments in cells are, however, not of the same level of quality.

****Major comments****

1. The results do not show convincingly that Nedd4L is recruited to CCPs. There is plenty of indirect evidence, but to support the model shown in the last figure, authors need to show more than the staining in figure 2C. Live-cell imaging showing the post-FCHO2 recruitment of Nedd4L would be required. I understand that the recruitment would possibly occur in a fraction of events and may be difficult to catch. The `cmeAnalysis` script from the `danuser` lab(<https://doi.org/10.1016/j.devcel.2013.06.019>) can facilitate the identification of these events.
2. What happens to ENaC in Nedd4L and FCHO2 knockdown cells? One would expect accumulation of the receptor on the surface.
3. In the experiments in figure 1, it would be important to use a standard CME cargo as an internal control (transferrin). This will serve as a functional confirmation of FCHO2 knockdown and help the reader to put the Nedd4L knockdown experiments into the context of CME.
4. Quantification for the rescue experiment is required (figure 1C). if not possible, at least a picture where the reader can see transfected and non-transfected cells side-by-side is necessary.

****Minor comments****

1. The experiments in figure 3 must be presented in order as they are in the text. For example, figure 3E is cited in the text into the context of figure 7. It is very confusing.
2. A better explanation of the assay in 1C would facilitate its understanding for the non-specialist reader. The reader needs to read the methods section to understand how it was done.

To end on a positive note - I applaud the authors for experiment 6A. It is critical to show that liposome extrusion beyond 0.2um does not guarantee liposomes at that size.

****Referee cross-commenting****

I also agree with the other comments. Nothing to add.

2. Significance:

Significance (Required)

The manuscript convincingly describes a novel mechanism for the activation of the ubiquitin ligase Nedd4L. From a biochemical point of view, the manuscript is solid. However, to be able to put this mechanism in the context of a CME event, the authors need stronger evidence in cells. To be clear, I think that the results presented do suggest a CME link. However, one could argue, for example, that the results could also be explained by ubiquitination of ENaC post CME, in an endosomal compartment with similar curvature.

Expertise of the reviewer: F-BAR proteins, endocytosis, cell biology and biochemistry.

3. How much time do you estimate the authors will need to complete the suggested revisions:

Estimated time to Complete Revisions (Required)

(Decision Recommendation)

Between 1 and 3 months

Yes

Revision Plan

Manuscript number: RC-2023-01906

Corresponding author(s): Hiroyuki, Nakanishi

[The “revision plan” should delineate the revisions that authors intend to carry out in response to the points raised by the referees. It also provides the authors with the opportunity to explain their view of the paper and of the referee reports.]

The document is important for the editors of affiliate journals when they make a first decision on the transferred manuscript. It will also be useful to readers of the reprint and help them to obtain a balanced view of the paper.

*If you wish to submit a full revision, please use our "Full Revision" template. **It is important to use the appropriate template to clearly inform the editors of your intentions.**]*

1. General Statements [optional]

Thank you for your letter dated on May 5, 2023 concerning our manuscript (MS# RC-2023-01906) entitled “Activation of Nedd4L Ubiquitin Ligase by FCHO2-generated Membrane Curvature.”

We thank the reviewers for their constructive comments and suggestions. We have considered all reviewers’ comments and plan to revise our manuscript accordingly.

We believe that our revision plan will greatly improve the quality of our manuscript.

2. Description of the planned revisions

Reviewer #1

I enjoyed reading the paper by Sakamoto and colleagues, where they show that Nedd4L ubiquitin ligase activity is stimulated by membranes and in particular positive membrane curvature. This paper is a conceptual advance that hopefully will be extended by many other groups where membranes topology participates in the activation of associated enzymes, giving rise to added complexity but also specificity and further compartmentalization. It is an important paper for all cell biologists to understand.

1. My comments are all relatively minor and I hope can improve the readability of the paper, but will not alter the overall conclusion as this is well backed up. In general I would like to see

Revision Plan

more/better statistics/quantitation and better figure legends. I found that often one had to read the paper to understand a figure where reading the figure legend should suffice.

Reply: According to the reviewer's comment, we will quantify the experiments (Fig. 1C, Fig. 2, Fig. 9B, and Fig. 10B) and add descriptions of statistics (Fig. 5, Fig. 6, B and D, and Fig. 7C). We will also write better figure legends to enable the readers to easily understand experiments.

2. This paper reminds me of a paper from Gilbert Di Paolo's lab on the activation of synaptojanin PIP2 hydrolysis by high membrane curvature. One would expect that there may be many such proteins whose activities will be dependent on their membrane environment. I find it conceptually rather likely that a protein which interacts with membranes via a C2 domain (which has membrane insertions and will thus likely be curvature sensitive) will likely show some positive curvature sensitivity. Can I suggest this paper is referenced and discussed in the light of the discussion statement "Thus, our findings provide a new concept of signal transduction in which a specific degree of membrane curvature serves as a signal for activation of an enzyme that regulates a number of substrates."

Reply: According to the reviewer's comment, we will cite the paper entitled "synaptojanin-1-mediated PI(4,5)P₂ hydrolysis is modulated by membrane curvature and facilitates membrane fission" by Chang-Ileto et al. (Dev. Cell **20**, 206–18, 2011). We will also discuss this paper in the light of the discussion statement.

3. Where the paper could be improved (or I have not understood fully). In figure 1 there is a robust endocytosis of ENaC that is FCHO2 and Nedd4L sensitive. There is a rescue for FCHO2 in a fluorescence image (unquantified), so it would be good to have the more quantitative approach of rescue with both FCHO2 and Nedd4L in the biochemical assay.

Reply: Although the reviewer suggests a rescue experiment in the biochemical assay, the experiment is difficult because the transfection efficiency is low (about 50%). On the other hand, we agree with the reviewer that a quantitative approach is required in the rescue experiment (Fig. 1C). Therefore, we plan to quantify the rescue experiment for FCHO2 in the immunofluorescence assay. The reviewer also suggests a rescue experiment for Nedd4L as well as FCHO2. However, since the involvement of Nedd4L in ENaC endocytosis is well established, we do not think that the rescue experiment for Nedd4L is further required.

4. In figure 2 there is nice co-localisation between clathrin/FCHO2 and ENaC but not with Nedd4L. It would be good to have some quantitation of the co-localisation. But also one should use a Nedd4L mutant or a mutant of ENaC and so be able to visualise co-localisation between receptor and ub-ligase. I find it strange that there is no (or much less) Nedd4L-GFP visible in the cells overexpressing ENaC... Is there an explanation? Does overexpression of ENaC lead to

Revision Plan

more auto-ubiquitination of Nedd4L. Also the Nedd4L-GFP signal in other cells is punctate, while in the next figure Myc-Nedd4L is not.

Reply: According to the reviewer's comment, we will perform quantitative colocalization analysis in Fig. 2.

We have found that a catalytically inactive Nedd4L mutant, C922A, co-localizes with cell-surface α ENaC and FCHO2 in $\alpha\beta$ ENaC-HeLa cells (see the figure below). According to the reviewer's comment, these data will be added in the revised manuscript.

In Fig. 2C, Nedd4L was transiently transfected in cells stably expressing ENaC. In Nedd4L-transfected cells, overexpression of Nedd4L stimulated ENaC internalization, resulting in the disappearance of ENaC at the cell surface. On the other hand, in non-transfected cells, cell-surface ENaC was detected. Thus, Nedd4L-negative cells are non-transfected cells (cell-surface ENaC positive cells). This explanation will be added in the revised manuscript.

The staining pattern of Nedd4L depends on what section of the cell a confocal microscope was focused on. Nedd4L-GFP signals were punctate at the bottom section of the cell in Fig. 2, whereas Myc-Nedd4L was diffusely distributed at the upper section (cytoplasm) of the cell (Fig. 3). Thus, Nedd4L shows distribution throughout the cytoplasm and punctate staining at the bottom (cell surface). The staining pattern of Nedd4L is also affected by the expression amount of Nedd4L in cells. When Nedd4L was highly expressed in COS7 and HEK293 cells in Fig. 3, the punctate staining was hardly detected. This localization pattern of Nedd4L will be clearly described in the revised manuscript.

Revision Plan

5. In figure 3 it appears to me that there is co-localization between ENaC and amphiphysin. Is this not a positive piece of information? I am not sure that FBP17 is a good F-BAR domain to use given its oligomerization may well prevent membrane association of Nedd4L. Minor comment: I don't see tubules for amphiphysin in panel B.

Reply: The reviewer states that there is co-localization between Nedd4L and amphiphysin1 (Fig. 3A). However, Nedd4L was not recruited to membrane tubules generated by amphiphysin1. We will clearly show that there is no colocalization between Nedd4L and amphiphysin1.

The reviewer states that FBP17 may not be a good F-BAR domain to use because its oligomerization may well prevent membrane association of Nedd4L. However, we have shown that FCHO2 as well as FBP17 forms oligomer (Uezu et al. Genes Cells, 16, 868-878, 2011). Furthermore, we have found that FCHO2 inhibits the membrane binding and catalytic activity of Nedd4L when the PS percentage in liposomes is elevated (unpublished data and Fig. 9C). Thus, since FBP17 and FCHO2 probably have similar properties, we presume that FBP17 is a good F-BAR domain to use.

As the reviewer pointed out, membrane tubules generated by amphiphysin1 were hardly detected in HEK293 cells (Fig. 3B). It showed punctate staining, but did not co-localized with Nedd4L. This description will be added in the revised manuscript.

6. Figure 5: The affinity of Nedd4 C2 domain for calcium is quite high given we normally assume a cytosolic concentration of 100nM (approximate). The authors have rightly buffered the calcium with EGTA. Normally we would check that the buffering is sufficient by varying the protein concentration and making sure the affinity is still the same, so can I suggest the authors use 3 or 4 times the amount of C2 domain and make sure the curve does not change (provided liposomes are not limiting). Minor comment: How many experiments and what are error bars (SD?).

Reply: According to the reviewer's comment, we will check that the buffering is sufficient by varying the protein concentration (Fig. 5). We will also add a description of statistics to the legend to Fig. 5.

7. Figure 6: Controls have been performed to ensure that liposomes are pelleted, according to methods. In Figure 6B can the authors show that there is the same amount of liposomes in each sample by showing more of the coomassie gel so that the reader can see the Neutravidin band is the same in each sample. Also I believe a student t-test should not be used in this experiment (but perhaps an Anova test), and in panel D there does not appear to be a description of statistics.

Revision Plan

Reply: To ensure that the same amounts of liposomes were pelleted, the reviewer suggests that we show more of the Coomassie gel to present the neutravidin bands in Fig. 6B. However, as the molecular weight of neutravidin is about 15 kDa, neutravidin run out of the gel (7% SDS-PAGE gel) where Nedd4L (<100 kDa) was analyzed. Instead, we will describe in the figure legend that the percentage of precipitated liposomes of each size was estimated to be ~100% by measuring both total and supernatant fluorescence.

As the reviewer pointed out, we will use an Anova test in Fig. 6B. We will also add a description of statistics in Fig. 6D.

8. *Figure 11: In panel B I note that the FCHO2 BAR domain on small liposomes appears to inhibit Ubiquitination. Is this consistent with the BAR domain not preventing Nedd4L binding?*

Reply: The FCHO2 BAR domain enhances the liposome binding and catalytic activity of Nedd4L when the strength of interaction of Nedd4L with liposomes (20% PS) is weak. In contrast, we have also found that the FCHO2 BAR domain inhibits the membrane binding and catalytic activity of Nedd4L when the interaction of Nedd4L with liposomes is increased by elevating the PS percentage in liposomes (unpublished data and Fig. 9C).

The reason for the different effects of FCHO2 on Nedd4L is considered as follows: When liposomes (20% PS) are used (the interaction of Nedd4L with PS in liposomes is weak), Nedd4L binds to liposomes mainly through ENaC (Fig. 8F). The liposome binding is hardly mediated by PS. Addition of the FCHO2 BAR domain increases the strength of interaction Nedd4L with PS by generating membrane curvature. Consequently, the FCHO2 BAR domain newly induces the PS-mediated liposome binding of Nedd4L, resulting in the enhancement of liposome binding and catalytic activity of Nedd4L. On the other hand, when the interaction of Nedd4L with PS in liposomes is increased by elevating the PS percentage in liposomes (50% PS), the liposome binding of Nedd4L is mainly mediated by PS. Addition of the FCHO2 BAR domain inhibits the PS-mediated liposome binding of Nedd4L. Since both FCHO2 and Nedd4L are PS-binding proteins, they compete with each other to bind to PS in liposomes. Therefore, the results in Fig. 11B are consistent, because the interaction of Nedd4L with PS is increased by 0.05 μ m pore-size liposomes.

This explanation will be added in the revised manuscript.

Reviewer #2

The authors have reported the involvement of the BAR domain-containing protein FCHO2 in the Nedd4L-mediated endocytosis of ENaC. They propose a model in which the membrane curvature induced by the BAR domain-FCHO2 relieves the auto-inhibition of E3 ligase causing its activation and recruitment. The paper describes a series of in vitro reconstituted experiments that are interesting but not fully connected with the mechanism of ENaC endocytosis. Additional experiments are needed to fully support the authors' conclusions.

Revision Plan

Major comments:

1. *Although the data reported by the authors regarding FCHO2 and Nedd4L involvement in ENaC endocytosis are convincing, it is suggested that the authors perform the same ENaC endocytosis assay presented in Fig.1B under conditions of FBP17 and amphiphysin1 siRNA to formally prove the selective involvement of FCHO2 in the process among other BAR-containing proteins.*

Reply: The reviewer suggests the same ENaC endocytosis assay presented in Fig. 1B under conditions of FBP17 and amphiphysin1 siRNA to prove the selective involvement of FCHO2 in ENaC endocytosis. There seems to be a misunderstanding. Similar to FCHO2, FBP17 and amphiphysin are well known to be involved in clathrin-mediated endocytosis. As ENaC is internalized through clathrin-mediated endocytosis, FBP17 and amphiphysin siRNA presumably inhibit ENaC endocytosis. We cannot understand the significance of FBP17 and amphiphysin1 siRNA in the ENaC endocytosis assay.

2. *According to the previous point, it will be interesting to see not only a snapshot image of the internalisation assay performed by immunofluorescence (Fig.1C) but a more quantitative analysis of the different time points (as in Fig.1B) in condition of FCHO2 siRNA and eventually FBP17 and amphiphysin1 siRNA.*

Reply: According to the reviewer's comment, we will perform a quantitative analysis in Fig. 1C. The reviewer also suggests the immunofluorescence assay at the different time point in Fig. 1C. However, we show the time course of ENaC internalization in Fig. 1B. We do not think that the time course in the immunofluorescence assay is further required. As for FBP17 and amphiphysin siRNA, our response is the same as that to the comment 1 of this reviewer.

3. *In Fig.2B, overexpression of the catalytically inactive version of Nedd4L (Nedd4L C922A) would help to see Nedd4L-ENaC co-localization.*

Reply: This comment is the same as the comment 4 of the reviewer#1.

4. *In Fig.4D, the authors need to analyse ENaC ubiquitination in the same experimental setting as Fig. 4A instead of transfecting cells with increasing amounts of Nedd4L in the presence or absence of FCHO2 BAR. It is also recommended to include Nedd4L C922A as an additional control.*

Reply: The reviewer requests us to analyse ENaC ubiquitination in the same setting as Fig. 4A. However, an in vivo autoubiquitination assay is widely used to determine the catalytic activity of

Revision Plan

E3 Ub ligase, because the E3 activity is typically reflected in their autoubiquitination. Therefore, the autoubiquitination assay is sufficient to show that Nedd4L is specifically activated by membrane tubules generated by FCHO2 in cells. Furthermore, we have found it very difficult to compare ENaC ubiquitination among many GFP-BAR proteins (GFP alone, GFP-FCHO2, GFP-FBP17, amphiphysin1-GFP, GFP-FCHO2 mutant) in the same experimental setting as Fig. 4A. In Fig. 4A, three types of cDNAs (HA-Ub, Myc-Nedd4L, and GFP-BAR protein) were transfected in cells. The expression amounts of Myc-Nedd4L were similar among the GFP-BAR proteins. On the other hand, in Fig. 4D, four types of cDNA (HA-Ub, Myc-Nedd4L, GFP-BAR protein, and FLAG- α ENaC) were transfected in cells. Under these conditions, it is very difficult to adjust the expression amounts of Nedd4L and α ENaC among many GFP-BAR proteins. Even when comparing two GFP-BAR proteins (GFP alone and GFP-FCHO2), it was necessary to assess the expression amounts of Nedd4L by transfection with various cDNA amounts of Nedd4L (Fig. 4D).

Moreover, as shown in Fig. 4D, enhancement of ENaC ubiquitination by FCHO2 is decreased at higher expression of Nedd4L (1.0 and 1.5 μ g DNA), although the reason is unknown. Therefore, we are not sure that we will be able to accurately analyse ENaC ubiquitination in the same setting as Fig. 4A instead of transfecting cells with increasing amounts of Nedd4L.

According to the reviewer's comment, we will examine the effect of Nedd4L C922A on ENaC ubiquitination.

5. While discussing the role of hydrophobic residues in Nedd4L C2 domain, the authors never mentioned the publication by Escobedo et al., Structure 2014 (DOI:10.1016/j.str.2014.08.016), which highlighted how I37 and L38 are directly involved in Ca²⁺ binding. This aspect should be discussed since the authors show the importance of Ca²⁺ for PS binding in the sedimentation assay.

Reply: According to the reviewer's comment, we will cite the reference (Escobedo et al.) and discuss the aspect (I37 and L38 are directly involved in Ca²⁺ binding).

6. As stated by the authors those two residues I37 and L38 are also involved in E3 enzyme activation by relieving C2-HECT interaction. It is important to further demonstrate the effect of these mutations on ENaC substrate.

Reply: To prove that the I37 and F38 residues are involved in E3 enzyme activation by relieving C2-HECT interaction, the reviewer requests us to further demonstrate the effect of Nedd4L I37A+F38A on ENaC ubiquitination. However, these two residues are critical not only for Nedd4L activation but also for membrane binding and curvature sensing of Nedd4L. We also show that membrane binding of Nedd4L is critical for ENaC ubiquitination. Actually, we have found that Nedd4L I37A+F38A mutant, which loses membrane binding, shows little ENaC ubiquitination (unpublished data), whereas it enhances autoubiquitination (Fig. 4C). Thus, the

Revision Plan

effect of the I37A+F38A mutant on ENaC ubiquitination is not appropriate to prove that the two residues are involved in E3 enzyme activation.

7. There are some concerns regarding the in vitro ubiquitination assay performed in Fig.8 and following figures. The Nedd4L proteins used during the assay has been produced as His tagged at the C-terminus, it was reported (Maspero et al, Nat Struct Mol Biol 2013 DOI: 10.1038/nsmb.2566), at least for the isolated HECT domain, that modification of the C-terminal residue of the protein affects its activity. It would be important to judge the activity of the purified proteins used in the assay.

Moreover, as additional control it is suggested the introduction of a mSA-ENaC PY mutant protein. The authors claimed the importance of membrane localized PY motif for recruitment and activation of Nedd4L, it would be informative to perform the experiment in presence of PY mutated ENaC.

Reply: The reviewer states that there are some concerns regarding His-tagged Nedd4L proteins. We have prepared Nedd4L that has no tag at its N- or C-terminus. N-terminal GST-tagged, C-terminal untagged Nedd4L was expressed in E. coli and purified by Glutathione-Sepharose column chromatography. The GST tag was cleaved off and Nedd4L was further purified by Mono Q anion-exchange column chromatography. Using this purified sample, we have examined the catalytic activity of untagged Nedd4L. We have found that concerning Ca²⁺-dependency, PS-dependency, and curvature-sensing, the properties of untagged Nedd4L are similar to those of C-terminal His-tagged Nedd4L (unpublished data).

According to the reviewer's comment, we will perform the experiment in the presence of PY-mutated ENaC.

8. It is not clear why increasing the concentration of PS (from 20% to 50%) the presence of BAR domain doesn't allow ENaC ubiquitination (Fig.9C), is Nedd4L not recruited to the pellet? It would be interesting to see the sedimentation experiment of Fig.9A done in presence of 50% PS.

Reply: This comment is essentially the same as the comment 8 of the reviewer#1. We have found that FCHO2 BAR domain inhibits the membrane binding of Nedd4L when the PS percentage in liposomes is elevated (~50%) (unpublished data). According to the reviewer's comment, these data will be added in the revised manuscript.

9. This reviewer is not an expert of lipids biology, thus the explanations related to the effect of FCHO2 BAR in presence of PI(4,5)P2 (Fig. 10) or 0.05 pore-size liposomes (Fig.11) were not clear. Does FCHO2 BAR have a different effect in inducing membrane tubulation in these two conditions? Is this parameter measurable by tubulation assay?

Revision Plan

Reply: According to the reviewer's comment, we will write more clearly the explanation related to the effect of FCHO2 BAR domain in the presence of PI(4,5)P₂ or 0.05 μm pore-size liposomes.

Minor Comments

1. It would be appreciated if a nuclei staining panel is included in all immunofluorescence images, as it would help to identify the number of cells in the field of view (e.g., Fig. 1C, Fig. 2B).

Reply: According to the reviewer's comment, we will show immunofluorescence images to identify the number of cells in Fig. 1C and Fig. 2B.

2. It would be recommended to include colocalization analysis, such as Pearson's correlation coefficient or Manders coefficient in immunofluorescence images.

Reply: According to the reviewer comment, we plan to perform quantitative colocalization analysis in Fig. 2.

3. It is not clear how the quantitation of mSA-ENaC ubiquitination in Fig.8D, 8C, and 9B was performed. Did the authors normalise the detected Ub signal over the amount of unmodified mSA-ENaC?

Reply: We did not normalize the detected Ub signals over the amount of unmodified mSA-ENaC, because the same amount of mSA-ENaC was added in each assay. The chemiluminescence intensity of Ub signals was quantified by scanning using ImageJ. According to the reviewer' comment, we will clearly describe how the quantification of mSA-ENaC ubiquitination was performed.

Reviewer #3

--- Summary ---

The manuscript by Sakamoto et al. describes how the ubiquitin ligase Nedd4L is activated by membrane curvature generated by the endocytic protein FCHO2. For their experiments, the authors use the epithelial sodium channel (ENaC) as a model Nedd4L target and CME cargo. The authors start their manuscript by showing in cells the importance of FCHO2 and Nedd4L in ENaC internalization. Using a combination of experiments in cells and biochemistry, the authors show that Nedd4L binds preferentially to membranes with the same curvature generated by

Revision Plan

FCHO2. Next, the authors show that a combination of membrane composition (PS), calcium concentration, PY domain presence and membrane curvature all act in concert to recruit Nedd4L to membranes and fully release its ubiquitination activity. Crucially, the authors show that role of FCHO2 in Nedd4L recruitment is not direct, with FCHO2 simply generating an optimal membrane curvature for Nedd4L binding. Taken together, the authors suggest a mechanism by which the curvature of early clathrin coated pits, generated by FCHO1/2 define an optimal environment for the recruitment and activation of the ubiquitin ligase Nedd4L.

The manuscript convincingly shows the membrane curvature-dependent mechanism of Nedd4L activation. The biochemistry experiments in the manuscript are well designed and the results are of clear. The quality of these experiments is very high. The experiments in cells are, however, not of the same level of quality.

--- Major comments ---

*1) The results do not show convincingly that Nedd4L is recruited to CCPs. There is plenty of indirect evidence, but to support the model shown in the last figure, authors need to show more than the staining in figure 2C. Live-cell imaging showing the post-FCHO2 recruitment of Nedd4L would be required. I understand that the recruitment would possibly occur in a fraction of events and may be difficult to catch. The *cmeAnalysis* script from the *danuser lab* (<https://doi.org/10.1016/j.devcel.2013.06.019>) can facilitate the identification of these events.*

Reply: According to the reviewer comment, we plan to examine by live-cell TIRF microscopy that Nedd4L is recruited to CCPs.

2) What happens to ENaC in Nedd4L and FCHO2 knockdown cells? One would expect accumulation of the receptor on the surface.

Reply: We have found that upon Nedd4L or FCHO2 knockdown, α ENaC accumulates at the cell surface in $\alpha\beta\gamma$ ENaC-HeLa cells (see the figure below). According to the reviewer's comment, we will show these data in the revised manuscript.

In $\alpha\beta\gamma$ ENaC-HeLa cells treated with each siRNA, cell-surface α ENaC was biotinylated and sequentially precipitated with anti-FLAG antibody and then avidin beads. Samples were analyzed with immunoblotting with anti-FLAG antibody. Data are shown as the mean \pm S.E.M of three independent experiments. *, $P < 0.05$ (the Student's t-test).

3) *In the experiments in figure 1, it would be important to use a standard CME cargo as an internal control (transferrin). This will serve as a functional confirmation of FCHO2 knockdown and help the reader to put the Need4L knockdown experiments into the context of CME.*

Reply: According to the reviewer's comment, we will use a standard CME cargo as an internal control (transferrin).

4) *Quantification for the rescue experiment is required (figure 1C). if not possible, at least a picture where the reader can see transfected and non-transfected cells side-by-side is necessary.*

Reply: This comment is the same as those of the reviewer#1 (comment 3) and reviewer#2 (comment 2). According to the reviewer's comment, we plan to quantify the rescue experiment (Fig. 1C).

--- *Minor comments* ---

1) *The experiments in figure 3 must be presented in order as they are in the text. For example, figure 3E is cited in the text into the context of figure 7. It is very confusing.*

Reply: According to the reviewer's comment, we will present the experiments in Fig. 3 in order they are in the text.

Revision Plan

2) *A better explanation of the assay in 1C would facilitate its understanding for the non-specialist reader. The reader needs to read the methods section to understand how it was done.*

Reply: According to the reviewer' comment, we will write a better explanation of the assay in the Fig. 1C legend to enable the readers to understand how it was done.

3. Description of the revisions that have already been incorporated in the transferred manuscript

Please insert a point-by-point reply describing the revisions that were already carried out and included in the transferred manuscript. If no revisions have been carried out yet, please leave this section empty.

4. Description of analyses that authors prefer not to carry out

Please include a point-by-point response explaining why some of the requested data or additional analyses might not be necessary or cannot be provided within the scope of a revision. This can be due to time or resource limitations or in case of disagreement about the necessity of such additional data given the scope of the study. Please leave empty if not applicable.

Dear Prof. Nakanishi,

Thank you for transferring your manuscript with Review Commons referee reports and responses to The EMBO Journal. I have now carefully read your manuscript, the referee comments as well as your revision plan. I agree with the reviewers that the proposed modulation of binding affinity and activity of Nedd4L by membrane curvature is of interest and that the experiments are rigorous and convincingly support the conclusions.

I also agree with you that the experiments proposed in the "revision plan" letter are appropriate to address the issues raised by the reviewers. We think it would be helpful in the response to comment 6 of referee 2 if you could provide the unpublished evidence that Nedd4L I37A shows little ENaC ubiquitination.

I therefore invite you to submit a revised version of the manuscript upon completion of these proposed experiments.

I would be happy to discuss the revision in more detail via email or phone/videoconferencing if you have any additional questions.

I should also add that it is The EMBO Journal policy to allow only a single major round of revision and that it is therefore important to resolve the main concerns at this stage.

We generally allow three months as standard revision time, which can be extended to six months in the case of major revisions. As a matter of policy, competing manuscripts published during this period will not negatively impact on our assessment of the conceptual advance presented by your study. However, please contact me as soon as possible upon publication of any related work to discuss the appropriate course of action. Should you foresee a problem in meeting this deadline, please let us know in advance to discuss an extension.

When preparing your letter of response to the referees' comments, please bear in mind that this will form part of the Review Process File and will therefore be available online to the community. For more details on our Transparent Editorial Process, please visit our website: <https://www.embopress.org/page/journal/14602075/authorguide#transparentprocess>. Please also see the attached instructions for further guidelines on preparation of the revised manuscript.

Please feel free to contact me if you have any further questions regarding the revision. Thank you for the opportunity to consider your work for publication. I look forward to discussing your revision.

Yours sincerely,

Cornelius Schneider

Cornelius Schneider, PhD
Editor
The EMBO Journal
c.schneider@embojournal.org

- a point-by-point response to the referees' comments, with a detailed description of the changes made (as a word file).
- a word file of the manuscript text.

- individual production quality figure files (one file per figure)
 - a complete author checklist, which you can download from our author guidelines (<https://www.embopress.org/page/journal/14602075/authorguide>).
 - Expanded View files (replacing Supplementary Information)
- Please see out instructions to authors
<https://www.embopress.org/page/journal/14602075/authorguide#expandedview>

We realize that it is difficult to revise to a specific deadline. In the interest of protecting the conceptual advance provided by the work, we recommend a revision within 3 months (12th Sep 2023). Please discuss the revision progress ahead of this time with the editor if you require more time to complete the revisions. Use the link below to submit your revision:

Link Not Available

Rev_Com_number: RC-2023-01906

New_manu_number: EMBOJ-2023-114687

Corr_author: Nakanishi

Title: Activation of Ned4L Ubiquitin Ligase by FCHO2-generated Membrane Curvature

Our responses to reviewers' comments and changes in the revised version (EMBOJ-2023-114687)

Title: Activation of Nedd4L ubiquitin ligase by FCHO2-generated membrane curvature

Authors: Y. Sakamoto, A. Uezu, K. Kikuchi, T. Moroishi, S. Suetsugu, and H. Nakanishi

Revisions have been made as follows according to the reviewers' comments

Responses to Reviewer #1

1. *My comments are all relatively minor and I hope can improve the readability of the paper, but will not alter the overall conclusion as this is well backed up. In general I would like to see more/better statistics/quantitation and better figure legends. I found that often one had to read the paper to understand a figure where reading the figure legend should suffice.*

Reply: Thank you for your valuable comment. According to the reviewer's comment, we quantified the experiments (Fig 1C and 2 in the unrevised manuscript). These results are shown in Fig 1D and Fig 2A–C. The statistics are described in the figure legends (page 42, lines 9–11; page 43 lines 3 and 4; and page 43, lines 10 and 11). We also added descriptions of statistics to the figure legends of Fig. 6A, 6B, 7A, 7B, 7D, 8C, 9C, 9E, 9F, 10C, 11B, and EV5: page 45, lines 4 and 5 (Fig 6 A and B); page 45, line 13 (Fig 7A); page 45, line 23 to page 46, line 2 (Fig 7B); page 46, lines 7 and 8 (Fig 7D); page 47, lines 1 and 2 (Fig 8C); page 47, lines 18 and 19 (Fig 9C); page 48, lines 1 and 2 (Fig 9F); page 48, lines 6–8 (Fig 9E); page 49, lines 4 and 5 (Fig 10C); page 49, lines 17 and 18 (Fig 11B) ; and page 52, lines 22 and 23 (Fig EV5).

Furthermore, additional details have been added to the figure legends of Fig 1B, 1D, 1E, 1F, and 7B: page 41, lines 7–16 (Fig 1B); page 42, lines 1–11 (Fig 1D); page 42, lines 12–18 (Fig. 1 E and F); and page 45, line 17 to page 46, line 2 (Fig 7B).

2. *This paper reminds me of a paper from Gilbert Di Paolo's lab on the activation of synaptojanin PIP2 hydrolysis by high membrane curvature. One would expect that there may be many such proteins whose activities will be dependent on their membrane environment. I find it conceptually rather likely that a protein which*

interacts with membranes via a C2 domain (which has membrane insertions and will thus likely be curvature sensitive) will likely show some positive curvature sensitivity. Can I suggest this paper is referenced and discussed in the light of the discussion statement "Thus, our findings provide a new concept of signal transduction in which a specific degree of membrane curvature serves as a signal for activation of an enzyme that regulates a number of substrates."

Reply: Thank you for providing this valuable suggestion. Based on the reviewer's comment, we have cited and discussed the paper by Chang-Ileto *et al.* (Dev. Cell **20**, 206–218, 2011) (page 15, lines 6–9).

3. *Where the paper could be improved (or I have not understood fully). In figure 1 there is a robust endocytosis of ENaC that is FCHO2 and Nedd4L sensitive. There is a rescue for FCHO2 in a fluorescence image (unquantified), so it would be good to have the more quantitative approach of rescue with both FCHO2 and Nedd4L in the biochemical assay.*

Reply: Thank you for this valuable suggestion. While the reviewer suggests a rescue experiment in the biochemical assay, it poses challenges due to the low transfection efficiency (approximately 50%). However, we acknowledge the importance of a quantitative approach in the rescue experiment. Consequently, we have quantified the rescue experiment for FCHO2 in the immunofluorescence assay, as depicted in Fig 1D. The reviewer also recommends a rescue experiment for Nedd4L, in addition to FCHO2. However, given the well-established role of Nedd4L in ENaC endocytosis, we believe that a rescue experiment for Nedd4L is not deemed necessary.

4. *In figure 2 there is nice co-localisation between clathrin/FCHO2 and ENaC but not with Nedd4L. It would be good to have some quantitation of the co-localisation. But also one should use a Nedd4L mutant or a mutant of ENaC and so be able to visualise co-localisation between receptor and ub-ligase. I find it strange that there is no (or much less) Nedd4L-GFP visible in the cells overexpressing ENaC... Is there an explanation? Does overexpression of ENaC lead to more auto-ubiquitination of Nedd4L. Also the Nedd4L-GFP signal in other cells is punctate, while in the next figure Myc-Nedd4L is not.*

Reply: Thank you for this insightful comment. According to the reviewer's comment, we conducted quantitative colocalization analysis, as illustrated in Fig. 2A–C. Additionally, we investigated the localization of a catalytically inactive Nedd4L mutant, C922A, which co-localized with cell-surface α ENaC and FCHO2 in $\alpha\beta\gamma$ ENaC-HeLa cells, as depicted in Fig 2B and detailed in the manuscript (page 6, lines 14–18). The explanation of Nedd4L staining in Appendix Fig S2A (Fig. 2B in the unrevised manuscript), Fig 2C, and Fig 4 (Fig 2C and 3 in the unrevised manuscript) has been provided in the text (page 6, lines 14 and 15; page 7, lines 16–18).

5. *In figure 3 it appears to me that there is co-localization between ENaC and amphiphysin. Is this not a positive piece of information? I am not sure that FBP17 is a good F-BAR domain to use given its oligomerization may well prevent membrane association of Nedd4L. Minor comment: I don't see tubules for amphiphysin in panel B.*

Reply: Thank you for raising this concern. Regarding the reviewer's observation of co-localization between Nedd4L and amphiphysin1 [Fig. 4A (Fig. 3A in the unrevised manuscript)], we want to clarify that Nedd4L was not recruited to membrane tubules generated by amphiphysin1. To provide a clearer representation of the localization of amphiphysin1 and Nedd4L, we have updated the image in Fig. 4A (bottom panel)

In response to concerns about the FBP17 F-BAR domain, we want to emphasize that FCHO2, as well as FBP17, forms oligomers (Uezu *et al.* Genes Cells, 16, 868–878, 2011). Furthermore, we have demonstrated that FCHO2 inhibits the membrane binding and catalytic activity of Nedd4L when the PS percentage in liposomes is elevated, as shown in Fig. 10C and EV4, and described in the text (page 13, lines 6–8). Considering these similarities, we maintain that FBP17 is a suitable F-BAR domain for use.

Addressing concerns about membrane tubules generated by amphiphysin1 in HEK293 cells [Fig. 4B (Fig. 3B in the unrevised manuscript)], we note that these tubules were hardly detected, showing punctate staining and no co-localization with Nedd4L. This information has been added to the text (page 8, lines 2–4).

6. *Figure 5: The affinity of Nedd4 C2 domain for calcium is quite high given we normally assume a cytosolic concentration of 100nM (approximate). The authors have rightly buffered the calcium with EGTA. Normally we would check that the buffering is sufficient by varying the protein concentration and making sure the affinity is still the*

same, so can I suggest the authors use 3 or 4 times the amount of C2 domain and make sure the curve does not change (provided liposomes are not limiting). Minor comment: How many experiments and what are error bars (SD?).

Reply: Thank you for this valuable suggestion. In response to the reviewer's comment about examining the sufficiency of buffering by varying the protein concentration, we have performed experiments using three times the amount of the Nedd4L C2 domain. The results, demonstrating that the liposome binding did not change, are attached below.

In the co-sedimentation assay with the Nedd4L C2 domain, we used 5 μg (upper panel) or 15 μg (lower panel) of the C2 domain with brain-lipid liposomes (~50% PS) at various Ca^{2+} concentrations. The supernatants (S) and pellets (P) were subjected to SDS-PAGE, followed by Coomassie brilliant blue staining. (-) for $[Ca^{2+}]$ indicates EGTA alone.

In Fig. 5A and B (unrevised manuscript), data was shown as the mean \pm S.D. of three independent experiments. In the revised manuscript, error bars have been changed from S.D. to S.E.M., as indicated in Fig. 6.

7. Figure 6: Controls have been performed to ensure that liposomes are pelleted, according to methods. In Figure 6B can the authors show that there is the same amount of liposomes in each sample by showing more of the coomassie gel so that the reader can see the Neutravidin band is the same in each sample. Also I believe a student t-test should not be used in this experiment (but perhaps an Anova test), and in panel D there does not appear to be a description of statistics.

Reply: Thank you for these valuable suggestions. Regarding the reviewer's suggestion to

show more of the Coomassie gel to present the Neutravidin bands in Fig 7B (Fig 6B in the unrevised manuscript), we acknowledge the concern about the molecular weight of Neutravidin and its potential to run out of the gel. Instead of presenting additional gel images, we have clarified in the figure legend of Fig 7B (page 45, lines 17–21) that the proportion of precipitated liposomes of each size was calculated to be approximately 100% by measuring both total and supernatant fluorescence.

In response to the reviewer's point about using one-way analysis of variance with Tukey's post hoc test, this information has been added to the figure legend of Fig 7B (page 45, line 23 to page 46, line 2). Additionally, a description of statistics has been included in the figure legend of Fig 7D (Fig 6D in the unrevised manuscript) (page 46, lines 7 and 8).

8. *Figure 11: In panel B I note that the FCHO2 BAR domain on small liposomes appears to inhibit Ubiquitination. Is this consistent with the BAR domain not preventing Nedd4L binding?*

Reply: Thank you for raising this query. The FCHO2 BAR domain exhibits contrasting effects on the liposome binding and catalytic activity of Nedd4L, depending on the strength of Nedd4L's interaction with liposomes. The FCHO2 BAR domain enhances both the liposome binding and catalytic activity of Nedd4L, when the interaction strength of Nedd4L with liposomes is weak (20% PS) (Fig 10A and B). Conversely, the BAR domain inhibits the binding and activity of Nedd4L when the interaction of Nedd4L with liposomes is increased by elevating the PS percentage in liposomes (~50% PS) (Fig 10C and Fig EV4). The mechanism underlying these dual actions of the FCHO2 BAR domain on Nedd4L is as follows: When using liposomes (20% PS), Nedd4L primarily binds to liposomes through mSA- α ENaC, albeit weakly (Fig. 9F and 10A). This liposome binding is minimally mediated by PS. The addition of the FCHO2 BAR domain enhances the strength of Nedd4L's interaction with PS by inducing membrane curvature. Furthermore, this interaction with PS is synergistically potentiated by mSA- α ENaC on liposomes. Consequently, Nedd4L gains a new high-affinity binding to liposomes through PS, which augments its catalytic activity. In contrast, when using liposomes (~50% PS), the liposome binding of Nedd4L is predominantly mediated by PS. The addition of the FCHO2 BAR domain hinders the PS-mediated liposome binding of Nedd4L. Considering that both FCHO2 and Nedd4L are PS-binding proteins, they compete for PS binding in liposomes. This is consistent with the results obtained with 0.05 μ m pore-size liposomes (Fig 12B). In this case, the interaction of Nedd4L

with liposomes through PS is increased by membrane curvature. This detailed explanation has been incorporated into the manuscript (page 17, lines 6–23), and the discussion has been revised accordingly (page 17, line 24 to page 18, line 7).

Responses to Reviewer #2

1. *Although the data reported by the authors regarding FCHO2 and Nedd4L involvement in ENaC endocytosis are convincing, it is suggested that the authors perform the same ENaC endocytosis assay presented in Fig.1B under conditions of FBP17 and amphiphysin1 siRNA to formally prove the selective involvement of FCHO2 in the process among other BAR-containing proteins.*

Reply: Thank you for your valuable suggestion. The reviewer suggests repeating the ENaC endocytosis assay presented in Fig. 1B under conditions of FBP17 and amphiphysin1 siRNA to demonstrate the selective involvement of FCHO2 in ENaC endocytosis. There seems to be a misunderstanding. Similar to FCHO2, FBP17 and amphiphysin are well-known participants in clathrin-mediated endocytosis. As ENaC is internalized through clathrin-mediated endocytosis, FBP17 and amphiphysin siRNA would presumably inhibit ENaC endocytosis. We are unable to comprehend the significance of FBP17 and amphiphysin1 siRNA in the ENaC endocytosis assay.

2. *According to the previous point, it will be interesting to see not only a snapshot image of the internalisation assay performed by immunofluorescence (Fig.1C) but a more quantitative analysis of the different time points (as in Fig.1B) in condition of FCHO2 siRNA and eventually FBP17 and amphiphysin1 siRNA.*

Reply: Thank you for this valuable suggestion. The reviewer suggests a quantitative analysis in Fig 1D (Fig 1C in the unrevised manuscript). However, we have presented the time course of ENaC internalization in Fig 1B, making the time course in the immunofluorescence assay redundant. Regarding FBP17 and amphiphysin siRNA, our response remains the same as in response to the comment 1 of this reviewer.

3. *In Fig.2B, overexpression of the catalytically inactive version of Nedd4L (Nedd4L C922A) would help to see Nedd4L-ENaC co-localization.*

Reply: Thank you for this valuable comment. This comment is the same as the comment 4 of the reviewer #1.

4. *In Fig.4D, the authors need to analyse ENaC ubiquitination in the same experimental setting as Fig. 4A instead of transfecting cells with increasing amounts of Nedd4L in the presence or absence of FCHO2 BAR. It is also recommended to include Nedd4L C922A as an additional control.*

Reply: Thank you for these valuable suggestions. The reviewer requests an analysis of ENaC ubiquitination in the same setting as Fig 5A (Fig 4A in the unrevised manuscript). However, an in vivo autoubiquitination assay is widely used to determine the catalytic activity of E3 Ub ligase, as the E3 activity is typically reflected in their autoubiquitination. Therefore, the autoubiquitination assay is sufficient to show that Nedd4L is specifically activated by membrane tubules generated by FCHO2 in cells. Furthermore, comparing ENaC ubiquitination among many GFP-BAR proteins in the same experimental setting as Fig 5A is challenging due to the complex conditions of multiple transfected cDNAs. We have found it difficult to adjust the expression amounts of Nedd4L and α ENaC among many GFP-BAR proteins. Even when comparing two GFP-BAR proteins (GFP alone and GFP-FCHO2), assessing the expression amounts of Nedd4L by transfecting various cDNA amounts of Nedd4L was necessary (Fig 5F). Enhancement of ENaC ubiquitination by FCHO2 is decreased at higher expression of Nedd4L (1.0 and 1.5 μ g DNA), although the reason is unknown. We could not accurately analyze ENaC ubiquitination in the same setting as Fig. 5A.

According to the reviewer's comment, we have examined the effect of Nedd4L C922A on ENaC ubiquitination. These results are shown in Fig. EV3 and described in the text (page 11, line 21 to page 12, line 1).

5. *While discussing the role of hydrophobic residues in Nedd4L C2 domain, the authors never mentioned the publication by Escobedo et al., Structure 2014 (DOI:10.1016/j.str.2014.08.016), which highlighted how I37 and L38 are directly involved in Ca²⁺ binding. This aspect should be discussed since the authors show the importance of Ca²⁺ for PS binding in the sedimentation assay.*

Reply: Thank you for pointing this out. According to the reviewer's comment, we have cited and discussed the reference (Escobedo *et al.*) in the text (page 11, lines 10–12).

6. *As stated by the authors those two residues I37 and L38 are also involved in E3 enzyme activation by relieving C2-HECT interaction. It is important to further demonstrate the effect of these mutations on ENaC substrate.*

Reply: Thank you for this valuable suggestion. According to the reviewer's comment, we have investigated the effect of the Nedd4L I37A+F38A mutant on α ENaC ubiquitination. These results are presented in Fig EV3 and described in the text (page 11, line 21 to page 12, line 1).

7. *There are some concerns regarding the in vitro ubiquitination assay performed in Fig.8 and following figures. The Nedd4L proteins used during the assay has been produced as His tagged at the C-terminus, it was reported (Maspero et al, Nat Struct Mol Biol 2013 DOI: 10.1038/nsmb.2566), at least for the isolated HECT domain, that modification of the C-terminal residue of the protein affects its activity. It would be important to judge the activity of the purified proteins used in the assay. Moreover, as additional control it is suggested the introduction of a mSA-ENaC PY mutant protein. The authors claimed the importance of membrane localized PY motif for recruitment and activation of Nedd4L, it would be informative to perform the experiment in presence of PY mutated ENaC.*

Reply: Thank you for raising these concerns. The reviewer expresses concerns about His-tagged Nedd4L proteins. We generated Nedd4L without tags at its N- or C-terminus. N-terminal GST-tagged, C-terminal untagged Nedd4L was expressed in *E. coli* and purified through Glutathione-Sepharose column chromatography. The GST tag was removed, and Nedd4L was further purified via Mono Q anion-exchange column chromatography. Using this purified untagged Nedd4L sample, we examined its catalytic activity. The properties of untagged Nedd4L, concerning Ca^{2+} dependency, PS dependency, and curvature sensing, were found to be similar to those of C-terminal His-tagged Nedd4L (unpublished data).

In response to the reviewer's comment, we have conducted experiments in the presence of PY-mutated ENaC. These results are displayed in Appendix Fig S3 and described in the text (page 12, lines 12–15).

8. *It is not clear why increasing the concentration of PS (from 20% to 50%) the*

presence of BAR domain doesn't allow ENaC ubiquitination (Fig.9C), is Nedd4L not recruited to the pellet? It would be interesting to see the sedimentation experiment of Fig.9A done in presence of 50% PS.

Reply: Thank you for raising this query and providing this insightful suggestion. This comment mirrors the comment 8 of the reviewer #1. We observed that the FCHO2 BAR domain inhibits the membrane binding of Nedd4L when the PS percentage in liposomes is elevated (approximately 50%). These data are presented in Fig EV4 and described in the text (page 13, lines 6–8).

9. *This reviewer is not an expert of lipids biology, thus the explanations related to the effect of FCHO2 BAR in presence of PI(4,5)P₂ (Fig. 10) or 0.05 pore-size liposomes (Fig.11) were not clear. Does FCHO2 BAR have a different effect in inducing membrane tubulation in these two conditions? Is this parameter measurable by tubulation assay?*

Reply: Thank you for raising these questions. According to the reviewer's comment, we have revised the explanation related to the effect of the FCHO2 BAR domain in the presence of PI(4,5)P₂ or 0.05 μm pore-size liposomes (page 13, line 18 to page 14, line 18). The FCHO2 BAR domain has been shown to induce the tubulation of PI(4,5)P₂ liposomes (Henne et al, 2010). It is presumed that the FCHO2 BAR domain has the same effect between PS and PI(4,5)P₂ liposomes in inducing membrane tubulation. Because the BAR domain binds to negatively charged phospholipids, such as PS and PI(4,5)P₂, and forces membranes to bend according to the intrinsic curvature, thereby inducing the formation of membrane tubules with a specific diameters (curvature). On the other hand, the diameter of 0.05 μm pore-size liposomes is consistent with that of membrane tubules generated by the FCHO2 BAR domain. This domain is, thus, expected not to induce the tubulation of 0.05 μm pore-size liposomes. We confirmed that when a tubulation assay (immunofluorescence) was performed, liposome tubulation was not observed.

Minor comments

1. *It would be appreciated if a nuclei staining panel is included in all immunofluorescence images, as it would help to identify the number of cells in the field of view (e.g., Fig. 1C, Fig. 2B).*

Reply: Thank you for this valuable suggestion. According to the reviewer's comment, we have outlined cell borders with solid lines to identify the number of cells in Fig 1D and Appendix S2A (Figs 1C and 2B, respectively, in the unrevised manuscript). Additionally, we have added original immunofluorescence pictures of Fig. 1D, where solid boxes correspond to the images of Fig 1D. These data are presented in Fig Appendix S1.

2. *It would be recommended to include colocalization analysis, such as Pearson's correlation coefficient or Manders coefficient in immunofluorescence images.*

Reply: Thank you for this recommendation. This comment is the same as the comment 4 of the reviewer #1.

3. *It is not clear how the quantitation of mSA-ENaC ubiquitination in Fig.8D, 8C, and 9B was performed. Did the authors normalise the detected Ub signal over the amount of unmodified mSA-ENaC?*

Reply: Thank you for raising this concern. We did not normalize the detected Ub signals over the amount of unmodified mSA-ENaC because the same amount of mSA-ENaC was added in each assay. The chemiluminescence intensity of Ub signals was quantified by scanning using ImageJ software. To address the reviewer's comment, this method has been added to the figure legends of Figs 9C, D, and 10B (Figs 8C, D, and 9B, respectively, in the unrevised manuscript) (page 47, lines 18 and 19; page 47, line 23; and page 49, lines 2 and 3).

Responses to Reviewer #3

Major comments

1. *The results do not show convincingly that Nedd4L is recruited to CCPs. There is plenty of indirect evidence, but to support the model shown in the last figure, authors need to show more than the staining in figure 2C. Live-cell imaging showing the post-FCho2 recruitment of Nedd4L would be required. I understand that the recruitment would possibly occur in a fraction of events and may be difficult to*

catch. The cmeAnalysis script from the danuser lab can facilitate the identification of these events.

Reply: Thank you for this insightful comment. According to the reviewer's comment, we have examined whether Nedd4L is recruited to CCPs by live-cell TIRF microscopy. These results are presented in Fig 3 and Appendix S2B, and described in the text (page 6, line 21 to page 7, line 15).

2. *What happens to ENaC in Nedd4L and FCHO2 knockdown cells? One would expect accumulation of the receptor on the surface.*

Reply: Thank you for raising this query. We observed that upon Nedd4L or FCHO2 knockdown, α ENaC accumulates at the cell surface in $\alpha\beta\gamma$ ENaC-HeLa cells. Following the reviewer's comment, these results have been shown in Fig. 1F, and described in the text (page 6, lines 4 and 5).

3. *In the experiments in figure 1, it would be important to use a standard CME cargo as an internal control (transferrin). This will serve as a functional confirmation of FCHO2 knockdown and help the reader to put the Nedd4L knockdown experiments into the context of CME.*

Reply: Thank you for this valuable suggestion. To address the reviewer's comment, we have used a standard CME cargo as an internal control (transferrin). These results are shown in Fig 1C, and described in the text (page 5, line 23 to page 6, line 2).

4. *Quantification for the rescue experiment is required (figure 1C). if not possible, at least a picture where the reader can see transfected and non-transfected cells side-by-side is necessary.*

Reply: Thank you for this valuable suggestion. This comment is the same as those of the reviewer #1 (comment 3) and reviewer #2 (comment 2). In response to the reviewer's comment, we have quantified the rescue experiment in Fig 1D and Appendix S1.

Minor comments

1. *The experiments in figure 3 must be presented in order as they are in the text. For example, figure 3E is cited in the text into the context of figure 7. It is very confusing.*

Reply: Thank you for pointing out this error. According to the reviewer's comment, we have moved Fig 3C–E (unrevised manuscript) to Fig 5C and E (revised manuscript) to maintain their order in the text.

2. *A better explanation of the assay in 1C would facilitate its understanding for the non-specialist reader. The reader needs to read the methods section to understand how it was done.*

Reply: Thank you for this valuable suggestion. To address the reviewer's comment, we have provided a clearer explanation of the assay in the figure legend of Fig 1D (Fig 1C in the unrevised manuscript) (page 42, lines 1–11).

Other changes

1. Toshio Moroishi has been added as an author.
2. According to the submission guideline for the EMBO Journal, the abstract has been revised so as to not exceed 175 words.
3. Six references (Aguet *et al*, 2013; Chang-Ileto *et al*, 2011; Escobedo *et al*, 2014; Roberts *et al*, 2001; Saffarians *et al*, 2009; and Zaccai *et al*, 2022) have been added.
4. As new figures were added, the figure numbering has been changed: 1C→1D; 1D→1E; 2B→Appendix S2A; 3A→4A; 3B→4B; 3C→5C; 3D→5E; 3E→5E; 4A→5A; 4B→5B; 4C→5D; 4D→5F; 5→6; 6→7; 7→8; 8→9; 9→10; 10→11; 11→12; 12→13; S1→EV1; S2→EV2; and S3→EV5.
5. When revising our manuscript, we found that the images in Fig. 2C (unrevised manuscript) showed Myc-FCHO2, but not endogenous FCHO2. To correct this mistake, the images in Fig. 2C have been changed to new images that show endogenous FCHO2.

6. We found minor mistakes in Fig 7A in the right panel (Fig 6A right panel in the unrevised manuscript), Fig 7B (Fig 6B), Fig 9E right panel (Fig 8E right panel), and Fig 9F right panel (Fig 8 right panel). These figures have been corrected. The new figures are essentially the same as the unrevised figures.
7. According to the submission guideline for the EMBO Journal, actual individual data from each experiment have been plotted in Fig. 1B, 1C, 1D, 1F, 2A, 2B, 2C, 7B, and 9E.
8. In the unrevised manuscript, the percentage of cells forming membrane tubules among cells expressing the FCHO2 BAR domain was described in the text. These data were shown as the mean \pm S.D. In the revised manuscript, error bars have been changed from S.D. to S.E.M. These results and statistics are described in the text (page 8, lines 8 and 9).

Dear Prof. Nakanishi,

Thank you for submitting your manuscript for consideration by the EMBO Journal. It has now been seen by two of the original referees. whose comments are shown below.

While both referees agree that the manuscript is significantly improved, referee #1 asks for additional minor text improvements and referee #2 thinks that there are important concerns that were not sufficiently addressed in the revision. While we normally do not allow for a second round of revisions I have the impression here that these additional experiments requested are reasonable and addressable and I therefore invite you to address these concerns in a second round of revisions. Do not hesitate to contact me if you have additional questions.

Thank you for the opportunity to consider your work for publication. I look forward to your revision.

Yours sincerely,

Cornelius Schneider, PhD
Editor
The EMBO Journal
c.schneider@embojournal.org

Please remember: Digital image enhancement is acceptable practice, as long as it accurately represents the original data and conforms to community standards. If a figure has been subjected to significant electronic manipulation, this must be noted in the figure legend or in the 'Materials and Methods' section. The editors reserve the right to request original versions of figures and

the original images that were used to assemble the figure.

We realize that it is difficult to revise to a specific deadline. In the interest of protecting the conceptual advance provided by the work, we recommend a revision within 3 months (2nd May 2024). Please discuss the revision progress ahead of this time with the editor if you require more time to complete the revisions. Use the link below to submit your revision:

Referee #1:

The manuscript by Sakamoto et al. describes the activation mechanism of the ubiquitin ligase Nedd4L by membrane curvature at clathrin mediated endocytosis (CME) sites. The authors show that the curvature generated by the CME nucleator protein FCHO2 generates the ideal curvature for the recruitment of Nedd4L to endocytic sites which, in turn, lead to the ubiquitination and internalization of the sodium channel ENaC.

In this revised version, the authors have addressed all my previous concerns and, in my opinion, present a convincing and relevant story demonstrating an interesting and novel mechanism: How the specific curvature generated at curvature of clathrin coated pits serve as a signal for the activation of an enzyme essential for the trafficking of ENaC, and potentially other receptors.

Minor concerns still to be addressed:

- 1) I appreciate the effort of the authors to show that Nedd4L localises to clathrin coated pits. Moreover, I understand their difficulties to colocalise FCHO2 and Nedd4L using overexpression. As correctly pointed out by the authors, many CME proteins require very narrow levels of expression to correctly localise to CCPs. As FCHO2 is a feature of the overwhelming majority of CCPs and this data has been shown multiple times in literature (Henne, 2010; Lehmann, 2019 and Zaccai, 2022, to name a few), I think it is not relevant to show that the data in figure 3A. In my opinion, it deviates from the main point of your story. Therefore, if Nedd4L localises to clathrin puncta, there is no reason for readers to believe that FCHO2 will not be present.
- 2) The term "closed off" used to explain figure 3 is, I believe, not appropriate, especially for the non-specialist reader. "Excised", "internalised", or even "pinched off" are more appropriate terms. Crucially, many CCPs are abortive and without appropriate controls (i.e. dynamin) or a thorough quantification and classification of CCP lifetimes, it is unadvisable to say that vesicles were excised, or these are abortive events.
- 3) I also believe that the data on clathrin coated plaques is misleading and irrelevant to the manuscript. Here, the authors say that Nedd4L "closes off" from plaques. This is clearly not appropriate without showing any other evidence such as dynamin recruitment or reduction of clathrin fluorescence. Furthermore, the statement gives the impression that Nedd4L can, on its own, cause vesicle scission. From the data presented, it seems that Nedd4L is simply transiently recruited. Above all, clathrin plaques (or flat clathrin lattices) have recently been established as structures with a specialised function in controlling cellular adhesions (Lock, JCB 2019; Zuidema, JCS 2018; Zuidema, JCS 2022 and; Hakanpää, JCB 2023). This function seems to be independent of their endocytic function.

Referee #2:

The points raised by Review Commons during the initial evaluation were not adequately addressed by the authors. I find the changes and additional data presented, as well as the justifications provided in the response, to be unsatisfactory. The text flow is challenging to read, and the content seems too preliminary to meet the standards for publication. At its current stage, I do not believe the manuscript is suitable for publication in EMBO.

Specifically, the concerns regarding the validation of FCHO2's specific involvement in ENaC endocytosis require further attention. According to the authors, FCHO2 is essential for Nedd4-L-mediated ubiquitination and endocytosis of ENaC due to its role in generating a specific degree of membrane curvature crucial for Nedd4-L activation.

It is crucial to establish a clear dependency of ENaC ubiquitination solely in conditions of FCHO2 depletion, excluding scenarios where other BAR domain-containing proteins are depleted. If conducting ENaC endocytosis experiments is deemed too labor-intensive, it is important at least, incorporate into the new Figure 1E the impact of FBP17 and amphiphysin1 siRNA on ENaC ubiquitination. This addition would help convince the reviewer that FCHO2 has a distinct physiological effect, supporting the manuscript's credibility.

The authors' explanation regarding the "dual role" of the FCHO2 domain based on the percentage of PS in liposomes is unclear. It remains uncertain how this discrepancy can be reconciled with its impact on ENaC endocytosis. Additionally, the authors should address whether there are specific physiological contexts or conditions that could account for the dual role played by the FCHO2 domain. A more detailed and comprehensive explanation of these aspects is necessary to enhance the clarity and coherence of the manuscript.

Our responses to reviewers' comments and changes in the revised version (EMBOJ-2023-114687R1)

Title: Activation of Nedd4L ubiquitin ligase by FCHO2-generated membrane curvature

Authors: Y. Sakamoto, A. Uezu, K. Kikuchi, J. Kang, E. Fujii, T. Moroishi, S. Suetsugu, and H. Nakanishi

Revisions have been made as follows according to the reviewers' comments

Responses to Reviewer #1

1. *I appreciate the effort of the authors to show that Nedd4L localises to clathrin coated pits. Moreover, I understand their difficulties to colocalise FCHO2 and Nedd4L using overexpression. As correctly pointed out by the authors, many CME proteins require very narrow levels of expression to correctly localise to CCPs. As FCHO2 is a feature of the overwhelming majority of CCPs and this data has been shown multiple times in literature (Henne, 2010; Lehmann, 2019 and Zaccari, 2022, to name a few), I think it is not relevant to show that the data in figure 3A. In my opinion, it deviates from the main point of your story. Therefore, if Nedd4L localises to clathrin puncta, there is no reason for readers to believe that FCHO2 will not be present.*

Reply: Thank you for your valuable comment. According to the reviewer's comment, we have deleted Fig. 3A and Movie EV1.

2. *The term "closed off" used to explain figure 3 is, I believe, not appropriate, especially for the non-specialist reader. "Excised", "internalised", or even "pinched off" are more appropriate terms. Crucially, many CCPs are abortive and without appropriate controls (i.e. dynamin) or a thorough quantification and classification of CCP lifetimes, it is unadvisable to say that vesicles were excised, or these are abortive events.*

Reply: Thank you for providing this valuable suggestion. Based on the reviewer's comment, we have deleted the term "closed off".

3. *I also believe that the data on clathrin coated plaques is misleading and irrelevant to the manuscript. Here, the authors say that Neddl4 "closes off" from plaques. This is clearly not appropriate without showing any other evidence such as dynamin recruitment or reduction of clathrin fluorescence. Furthermore, the statement gives the impression that Neddl4 can, on its own, cause vesicle scission. From the data presented, it seems that Neddl4 is simply transiently recruited. Above all, clathrin plaques (or flat clathrin lattices) have recently been established as structures with a specialised function in controlling cellular adhesions (Lock, JCB 2019; Zuidema, JCS 2018; Zuidema, JCS 2022 and; Hakanpää, JCB 2023). This function seems to be independent of their endocytic function.*

Reply: Thank you for this valuable comment. Based on the reviewer's comment, we have deleted the data (Fig. 3B and Movie EV2). We have also deleted the description about clathrin-coated plaques from the text.

Responses to Reviewer #2

1. *It is crucial to establish a clear dependency of ENaC ubiquitination solely in conditions of FCHO2 depletion, excluding scenarios where other BAR domain-containing proteins are depleted. If conducting ENaC endocytosis experiments is deemed too labor-intensive, it is important at least, incorporate into the new Figure 1E the impact of FBP17 and amphiphysin1 siRNA on ENaC ubiquitination. This addition would help convince the reviewer that FCHO2 has a distinct physiological effect, supporting the manuscript's credibility.*

Reply: Thank you for your comment. In response to the reviewer's comment, we examined the effects of FBP17 and amphiphysin2 knockdown on ENaC ubiquitination. Instead of amphiphysin1, we analyzed amphiphysin2, because amphiphysin1 is not expressed in HeLa cells, according to the Protein Atlas (www.proteinatlas.org). Amphiphysin2 RNAi remarkably inhibited ENaC expression at the cell surface. These results are attached below. Therefore, we could not accurately evaluate the effect of amphiphysin2 knockdown on ENaC ubiquitination. Amphiphysin2 has been reported to function in endocytic recycling. Amphiphysin2 RNAi may inhibit ENaC transport to the cell surface.

On the other hand, we found that FBP17 RNAi did not reduce ENaC ubiquitination at the cell surface. These results are shown in Fig 1E and Fig EV2 and are described in the text (page 6, lines 4 and 5).

2. The authors' explanation regarding the "dual role" of the FCHO2 domain based on the percentage of PS in liposomes is unclear. It remains uncertain how this discrepancy can be reconciled with its impact on ENaC endocytosis. Additionally, the authors should address whether there are specific physiological contexts or conditions that could account for the dual role played by the FCHO2 domain. A more detailed and comprehensive explanation of these aspects is necessary to enhance the clarity and coherence of the manuscript.

Reply: Thank you for these valuable comments. Accordingly, we have rewritten the explanation of the stimulatory and inhibitory effects of the FCHO2 BAR domain on Nedd4L (page 17, lines 3-21). Whether the effect is stimulatory or inhibitory depends on the PS percentage in liposomes. However, given that 20% PS is comparable to the

inner leaflet of the plasma membranae, it is plausible that FCHO2 exhibits only a stimulatory effect on Nedd4L in cells. This has been described in the revised text (page 17, lines 5-7). We believe that there are no specific physiological conditions where FCHO2 shows an inhibitory effect on Nedd4L in cells.

Other changes

1. Jangmi Kang and Eiko Fujii have been added as authors.
2. The experimental procedure for FBP17 RNAi has been described in the text (page 24, lines 11 and 12; and page 24, line 21 to page 25, line 6).
3. As a new figure (Fig EV2) has been added, the figure numbering has been updated: EV2→EV3; EV3→EV4; EV4→EV5; and EV5→EV6.
4. As Movies EV 1 and 2 have been deleted, the movie numbering has been updated: EV3→EV1
5. A reference (Lehmann *et al.*, 2019) has been added.

Dear Dr Nakanishi

Thank you for submitting a revised version of your manuscript. Unfortunately, referee #2 was not able to re-review the manuscript. We think that your responses to the concerns raised by this referee are reasonable and have therefore decided to not require any further experiments. There remain only a few mainly editorial points that have to be addressed before I can extend formal acceptance of the manuscript:

- On the abstract page of the manuscript, please include 4-5 general keyword terms to enhance searchability.
- Please rename the Conflict-of-Interest section into "Disclosure and Competing Interests Statement", in accordance with our updated Guide to Authors
- As we are switching from a free-text author contribution statement towards a more formal statement based on Contributor Role Taxonomy (CRediT) terms, please remove the present Author Contribution section and instead specify each author's contribution(s) directly in the Author Information page of our submission system during upload of the final manuscript. See <https://casrai.org/credit/> for more information.
- Please make sure that the author checklist is filled out. Please select the response and mark in which section the info is available
- Please upload the figures as individual, hi-res figure files
- Please ZIP the SOURCE DATA for EV and/or appendix figures together into a single archive.
- Please provide the legends for figures 7d-f in a sequential manner (legend for figure 7f is provided before legend of figure 7e).
- Please define the box plots in terms of minima, maxima, center, and whiskers in the legends of figures 7a, d; EV 5b.
- Although 'n' is provided, please describe the nature of entity for 'n' in the legends of figures 7a, d; EV 5b."
- Please note that the scale bar needs to be defined for figure 2c.
- Please define the asterisk "*" in the legend of figure EV 1c.
- Please correct the section order which should be: title page with complete author information, abstract, keywords, introduction, results, discussion, methods, data availability section, acknowledgements, disclosure and competing interests statement, references, main figure legends, tables, expanded figure legends.
- In our standard source data check, we have noted unexplained numerical duplications in the source data. I have attached the corresponding files with the detected duplications labelled in color. Please take a look and correct as needed. A brief explanation would be very helpful.

With best regards,

Cornelius Schneider

Cornelius Schneider, PhD
Editor | The EMBO Journal
c.schneider@embojournal.org

Our responses to editorial points and changes in the revised version (EMBOJ-2023-114687R2)

Title: Activation of Ned4L ubiquitin ligase by FCHO2-generated membrane curvature

Authors: Y. Sakamoto, A. Uezu, K. Kikuchi, J. Kang, E. Fujii, T. Moroishi, S. Suetsugu, and H. Nakanishi

Revisions have been made as follows according to the editorial points

- On the abstract page of the manuscript, please include 4-5 general keyword terms to enhance searchability.

Reply: According to the point, we have added five keywords on the abstract page (page 2, line 19).

- Please rename the Conflict-of-Interest section into "Disclosure and Competing Interests Statement", in accordance with our updated Guide to Authors

Reply: Based on the point, we have changed "Competing Interests " to "Disclosure and Competing Interests Statement" (page 34, line 17).

- As we are switching from a free-text author contribution statement towards a more formal statement based on Contributor Role Taxonomy (CRediT) terms, please remove the present Author Contribution section and instead specify each author's contribution(s) directly in the Author Information page of our submission system during upload of the final manuscript. See <https://casrai.org/credit/> for more information.

Reply: According to this point, we have deleted the Author Contribution section from the text.

- Please make sure that the author checklist is filled out. Please select the response and mark in which section the info is available

Reply: According to this editorial point, we have filled out the author checklist.

- Please upload the figures as individual, hi-res figure files

Reply: According to this point, we have uploaded the figures as individual high-resolution figure files (EPS format).

- Please ZIP the SOURCE DATA for EV and/or appendix figures together into a single archive.

Reply: Based on this point, we have ZIPPED the SOURCE DATA into a single archive.

- Please provide the legends for Figures 7d-f in a sequential manner (legend for figure 7f is provided before legend of figure 7e).

Reply: We think that Figures 7D-F should be Figures 9D-F. According to the editorial point, we have provided the legends for Figures 9D-F in a sequential manner (page 47, line 18 to page 48, line 10).

- Please define the box plots in terms of minima, maxima, center, and whiskers in the legends of figures 7a, d; EV 5b.

Reply: We think that Figure EV 5B should be Figure EV 6B. According to this editorial point, we have defined the box plots in the legend of Figure 7A (page 45, lines 5-9), 7D (page 46, lines 3-6), and EV 6B (page 53, lines 8-11).

- Although 'n' is provided, please describe the nature of entity for 'n' in the legends of figures 7a, d; EV 5b."

Reply: 'n' represents the number of observations per group in boxplots. This is described in the legend of Figures 7A (page 45, line 6), 7D (page 46, lines 3 and 4) and EV6B (page 53, lines 8 and 9).

- Please note that the scale bar needs to be defined for figure 2c.

Reply: According to this point, we have defined the scale bar in Figure 2C (page 43, line 10).

- Please define the asterisk "*" in the legend of figure EV 1c.

Reply: According to the point, we have defined the asterisk in the legend of Figure EV 1C (page 51, line 23). In addition, we found that there was no arrowhead in Figure EV1, although it is defined in the legend. To correct this, we have changed the arrow to the arrowhead in Figure EV1B.

- Please correct the section order which should be: title page with complete author information, abstract, keywords, introduction, results, discussion, methods, data availability section, acknowledgements, disclosure and competing interests statement, references, main figure legends, tables, expanded figure legends.

Reply: According to the point, we have corrected the section order.

- In our standard source data check, we have noted unexplained numerical duplications in the source data. I have attached the corresponding files with the detected duplications labelled in color. Please take a look and correct as needed. A brief explanation would be very helpful.

Reply: Thank you very much for pointing out the numerical duplications in the source data of Figures, 7A, 7D, 9E, and 9F. In Figures 7A and 7E, the duplicate values resulted from the calculation [measured value (two significant digits) x scale factor (ten

significant digits)]. For example, 80 mm (measured value) x 7.431666250 nm/mm (scale factor) = 594.5333 nm. To avoid duplications, the values of liposome diameters have been measured more accurately (to three decimal places). The source data of Figures 7A and 7E have been corrected. Accordingly, these figures have also been corrected. The new figures are essentially the same as the unrevised figures.

In Figures 9E and 9F, the duplications resulted from our mistakes. We have re-quantified the chemiluminescence intensities by re-scanning. The source data of these figures have been corrected. Accordingly, these figures have also been corrected. In the source data of Figure 9E, the value at 0.05 is set to 1 because the figure is represented as folds relative to the value at 0.05. In the legend of Figure 9E, "*** P < 0.01" has been deleted. The new figures are essentially the same as the unrevised figures.

Dear Prof. Nakanishi,

I am pleased to inform you that your manuscript has been accepted for publication in the EMBO Journal.

Yours sincerely,

Cornelius Schneider, PhD
Editor
The EMBO Journal
c.schneider@embojournal.org
